# Greedier is Better: Selecting Multiple Neighbors per Iteration for Sparse Subspace Clustering

**Jwo-Yuh Wu**                                                    *jywu@nycu.edu.tw*
*Institute of Communications Engineering*
*National Yang Ming Chiao Tung University*

**Liang-Chi Huang**                                              *lchuang@nycu.edu.tw*
*Institute of Communications Engineering*
*National Yang Ming Chiao Tung University*

**Wen-Hsuan Li**                                                 *vincent@nycu.edu.tw*
*Institute of Communications Engineering*
*National Yang Ming Chiao Tung University*

**Chun-Hung Liu**                                                *chliu@ece.msstate.edu*
*Department of Electrical and Computer Engineering*
*Mississippi State University*

**Rung-Hung Gau**                                               *gaurunghung@nycu.edu.tw*
*Institute of Communications Engineering*
*National Yang Ming Chiao Tung University*

**Reviewed on OpenReview:** *https://openreview.net/forum?id=djD8IbSvgm*

## Abstract

Sparse subspace clustering (SSC) using greedy-based neighbor selection, such as orthogonal matching pursuit (OMP), has been known as a popular computationally-efficient alternative to the standard $\ell_1$-minimization based methods. However, existing stopping rules of OMP to halt neighbor search needs additional offline work to estimate some ground truths, e.g., subspace dimension and/or noise strength. This paper proposes a new SSC scheme using generalized OMP (GOMP), a soup-up of OMP whereby multiple, say $p(\geq 1)$, neighbors are identified per iteration to further speed up neighbor acquisition, along with a new stopping rule requiring nothing more than a knowledge of the ambient signal dimension and the number $p$ of identified neighbors in each iteration. Compared to conventional OMP (i.e., $p = 1$), the proposed GOMP method involves fewer iterations, thereby enjoying lower algorithmic complexity. Under the semi-random model, analytic performance guarantees are provided. It is shown that, with a high probability, (i) GOMP can retrieve more true neighbors than OMP, consequently yielding higher data clustering accuracy, and (ii) the proposed stopping rule terminates neighbor search once the number of recovered neighbors is close to the subspace dimension. Issues about selecting $p$ for practical implementation are also discussed. Computer simulations using both synthetic and real data are provided to demonstrate the effectiveness of the proposed approach and validate our analytic study.

## 1 Introduction

### 1.1 Motivation

Subspace clustering (Vidal, 2011; Yang et al., 2008; Goh & Vidal, 2007) is a key enabling technique in modern unsupervised machine learning and its principles can be recapitulated as follows. Consider a noisy dataset

---

**Algorithm 1** SSC-LASSO algorithm

---

**Input**: Observed dataset $\mathcal{Y} = \{\mathbf{y}_1, \mathbf{y}_2, ..., \mathbf{y}_N\}$, data matrix $\mathbf{Y} = [\mathbf{y}_1...\mathbf{y}_N]$, regularization parameter $\lambda > 0$
   **for** $i = 1$ **to** $N$ **do**
      1) $\mathbf{c}_i^* = \arg\min \lambda\|\mathbf{c}_i\|_1 + \frac{1}{2}\|\mathbf{y}_i - \mathbf{Y}\mathbf{c}_i\|_2^2$ s.t. $c_{i,i} = 0$.
      2) Normalize $\mathbf{c}_i^*$ and let $\overline{\mathbf{c}}_i^* = [c_{i,1}^*...c_{i,i-1}^*\ 0\ c_{i,i+1}^*...c_{i,N}^*]^T \in \mathbb{R}^N$.
   **end for**
   3) Set $\mathbf{C} = [c_{i,j}] = [\overline{\mathbf{c}}_1^* \cdots \overline{\mathbf{c}}_N^*]$, and $\mathbf{G} = [g_{i,j}]$, where $g_{i,j} = |c_{i,j}| + |c_{j,i}|$.
   4) Form an $N$-node similarity graph in which the edge between nodes $i$ and $j$ has edge weight $g_{i,j}$.
   5) Apply spectral clustering to the similarity graph.
**Output**: Partition $\mathcal{Y} = \widehat{\mathcal{Y}}_1 \cup ... \cup \widehat{\mathcal{Y}}_{\widehat{L}}$.

---

$\mathcal{Y} = \{\mathbf{y}_1, \mathbf{y}_2, ..., \mathbf{y}_N\} \subset \mathbb{R}^n$ whose ground truth obeys a disjoint union as

$$\mathcal{Y} = \mathcal{Y}_1 \cup \mathcal{Y}_2 \cup ... \cup \mathcal{Y}_L, \tag{1}$$

where cluster $\mathcal{Y}_k \subset \mathbb{R}^n$ consists of $|\mathcal{Y}_k| > 0$ noisy data points coming from a $d_k$-dimensional subspace $\mathcal{S}_k$, and $|\mathcal{Y}_1| + ... + |\mathcal{Y}_L| = N$. A partition of $\mathcal{Y}$ into the form (1) is widely known as the union-of-subspaces model (Vidal, 2011), which underpins a panoply of practical data clusters ranging from human face images, hand-written digits, to trajectories of moving objects in videos. Given $\mathcal{Y}$ with unknown $L$ and $d_k$, $1 \leq k \leq L$, the task of subspace clustering is to uncover the partition (1). Among existing solutions to this problem, sparse subspace clustering (SSC) (Liu et al., 2013; Li et al., 2017; Lu et al., 2019; Elhamifar & Vidal, 2013), catalyzed by the witnessed success of compressive sensing (CS) and sparse representation (Baraniuk, 2007; Candès & Wakin, 2008; Davenport et al., 2011; Elad, 2010), has gained much attention because of its compelling experimental performance and provable performance guarantees. A key ingredient of SSC is to identify for each data point $\mathbf{y}_i$ a neighbor group by using the sparse representation technique. Formally, we collect all data points in $\mathcal{Y}$ to form the matrix $\mathbf{Y} = [\mathbf{y}_1\ \mathbf{y}_2...\mathbf{y}_N] \in \mathbb{R}^{n \times N}$ and try to express $\mathbf{y}_i$ as a linear combination of columns of $\mathbf{Y}$ except $\mathbf{y}_i$, say, $\mathbf{y}_i = \mathbf{Y}\mathbf{c}_i$, where $\mathbf{c}_i = [c_{i,1}\ c_{i,2}...c_{i,N}]^T \in \mathbb{R}^N$ subject to $c_{i,i} = 0$. SSC aims to find an optimal *sparse* solution $\mathbf{c}_i^*$ so that the columns of $\mathbf{Y}$ indexed by the support of $\mathbf{c}_i^*$ are correct neighbors of $\mathbf{y}_i$ (i.e., from the same cluster). Computing $\mathbf{c}_i^*$ is typically done by solving the following $\ell_1$-minimization problem (a.k.a. the Lasso regressor (Hastie et al., 2015))

$$\mathbf{c}_i^* = \arg\min \lambda\|\mathbf{c}_i\|_1 + \frac{1}{2}\|\mathbf{y}_i - \mathbf{Y}\mathbf{c}_i\|_2^2 \ \text{ s.t. } c_{i,i} = 0, \tag{2}$$

where $0 < \lambda < 1$ is a regularization factor. Once $\mathbf{c}_i^*$ is obtained, SSC accordingly constructs a similarity graph with the edge connecting $\mathbf{y}_i$ and $\mathbf{y}_j$ with weight $g_{i,j} = |c_{i,j}| + |c_{j,i}|$ (see step 3 of Algorithm 1), followed by spectral clustering (von Luxburg, 2007) for final data segmentation (see Algorithms 1 and 2 for outlines of SSC-LASSO and spectral clustering algorithms, respectively). Solving problem (2) is computationally demanding, especially for large-scale high-dimensional datasets. Therefore, low-complexity alternatives using greedy-based neighbor selection, e.g., orthogonal matching pursuit (OMP) (Davenport et al., 2011; Elad, 2010), were proposed to perform on par with $\ell_1$-minimization in many cases (see Algorithm 3 for an outline of OMP-based SSC algorithm). SSC-OMP conducts neighbor identification by computing a sequence of orthogonal projections. Specifically, for each $\mathbf{y}_i$ OMP iteratively identifies one neighbor each time as the data point yields the peak absolute inner product when paired with the residual vector (the initial residual vector $\mathbf{r}_0^{(i)} = \mathbf{y}_i$). As a new neighbor is identified, its identity is added to the "already-detected" neighbor index subset $\Lambda_m$, and the new residual vector $\mathbf{r}_m^{(i)}$ is updated as the orthogonal projection $\mathbf{r}_m^{(i)} = (\mathbf{I} - \mathbf{Y}_{\Lambda_m}(\mathbf{Y}_{\Lambda_m}^T \mathbf{Y}_{\Lambda_m})^{-1}\mathbf{Y}_{\Lambda_m}^T)\mathbf{r}_{m-1}^{(i)}$, in which $\mathbf{Y}_{\Lambda_m}$ consists of the columns of $\mathbf{Y}$ indexed by $\Lambda_m$. Once the neighbor identification process ends, the sparse representation vector $\mathbf{c}_i^*$ is then obtained by solving a least-squares problem (see step 4 of Algorithm 3). Notably, existing stopping rules of OMP halt neighbor search if either the number of iterations reaches a predesignated maximum $M$ or the residual power is below a threshold $\tau$ (Tschannen & Bölcskei, 2018). While the number $M$ is closely related to the ground truth subspace dimension, the threshold $\tau$ is determined by the background noise strength. Hence, offline estimating the subspace dimension and noise strength is necessary.

---

**Algorithm 2** Spectral clustering algorithm

---

**Input**: Weighted adjacency matrix $\mathbf{G} = [g_{i,j}] \in \mathbb{R}^{N \times N}$

  1) Let $\mathbf{L} = \mathbf{I} - \mathbf{A}^{-1/2}\mathbf{G}\mathbf{A}^{-1/2}$, where $\mathbf{A} = diag\{a_1, ..., a_N\}$ and $a_i = \sum_{j=1}^{N} g_{i,j}$.

  2) Estimate the number of clusters $\widehat{L} = \min_k |\lambda_{k+1} - \lambda_k|$, where $\lambda_k$ is the $k$th smallest eigenvalue of $\mathbf{L}$.

  3) Find $\mathbf{v}_1, \mathbf{v}_2, ..., \mathbf{v}_{\widehat{L}}$, the eigenvectors of $\mathbf{L}$ associated with the $\widehat{L}$ smallest eigenvalues of $\mathbf{L}$, and set

    $\mathbf{V} = \left[ \mathbf{v}_1/\|\mathbf{v}_1\|_2 \ \ \mathbf{v}_2/\|\mathbf{v}_2\|_2 ... \mathbf{v}_{\widehat{L}}/\|\mathbf{v}_{\widehat{L}}\|_2 \right] \in \mathbb{R}^{N \times \widehat{L}}$.

  4) Segment the $N$ rows of $\mathbf{V}$ into $\widehat{L}$ clusters using the K-means algorithm.

  5) Declare $\mathbf{y}_i \in \widehat{\mathcal{Y}}_l$ if the $i$th row of $\mathbf{V}$ is assigned to the $l$th cluster.

**Output**: Partition $\mathcal{Y} = \widehat{\mathcal{Y}}_1 \cup ... \cup \widehat{\mathcal{Y}}_{\widehat{L}}$.

---

---

**Algorithm 3** SSC-OMP algorithm

---

**Input**: Observed dataset $\mathcal{Y} = \{\mathbf{y}_1, \mathbf{y}_2, ..., \mathbf{y}_N\}$, data matrix $\mathbf{Y} = [\mathbf{y}_1...\mathbf{y}_N]$, maximum number of iterations
      $M$, residual vector norm threshold $\tau$

  **for** $i = 1$ **to** $N$ **do**

    Let $m = 0$, $\mathbf{r}_0^{(i)} = \mathbf{y}_i$, $\Lambda_0 = \phi$.

    **if** $(m < M)$ and $(\|\mathbf{r}_m^{(i)}\|_2 > \tau)$ **then**

      1) $m \leftarrow m + 1$.

      2) $\Lambda_m = \Lambda_{m-1} \cup j^*$, where $j^* = \underset{1 \le j \ne i \le N}{\arg\max} |\langle \mathbf{y}_j, \mathbf{r}_{m-1}^{(i)} \rangle|$.

      3) $\mathbf{r}_m^{(i)} = (\mathbf{I} - \mathbf{Y}_{\Lambda_m}(\mathbf{Y}_{\Lambda_m}^T \mathbf{Y}_{\Lambda_m})^{-1}\mathbf{Y}_{\Lambda_m}^T)\mathbf{r}_{m-1}^{(i)}$.

    **end if**

    4) When the above procedure terminates with $M^{(i)}$ iterations, compute $\mathbf{c}_i^* = \underset{\mathbf{c}:\text{supp}(\mathbf{c}) \subset \Lambda_{M^{(i)}}}{\arg\min} \|\mathbf{y}_i - \mathbf{Y}\mathbf{c}\|_2$.

    5) Normalize $\mathbf{c}_i^*$ and let $\bar{\mathbf{c}}_i^* = [c_{i,1}^*...c_{i,i-1}^* \ 0 \ c_{i,i+1}^*...c_{i,N}^*]^T \in \mathbb{R}^N$.

  **end for**

  6) Set $\mathbf{C} = [c_{i,j}] = [\bar{\mathbf{c}}_1^* \cdots \bar{\mathbf{c}}_N^*]$, and $\mathbf{G} = [g_{i,j}]$, where $g_{i,j} = |c_{i,j}| + |c_{j,i}|$.

  7) Form an $N$-point similarity graph in which the edge between nodes $i$ and $j$ has edge weight $g_{i,j}$.

  8) Apply spectral clustering to the similarity graph.

**Output**: Partition $\mathcal{Y} = \widehat{\mathcal{Y}}_1 \cup ... \cup \widehat{\mathcal{Y}}_{\widehat{L}}$.

---

In addition to algorithm development, investigating the mathematical performance guarantees of SSC using fruitful analytical tools from CS also received considerable attention. The vast majority of related works focused on investigating sufficient conditions ensuring the so-called subspace detection property (SDP) (Soltanolkotabi & Candès, 2012), that is, neighbor identification is correct in the way that the coefficient $c_{i,j}^* \ne 0$ only if $\mathbf{y}_i$ and $\mathbf{y}_j$ are in the same cluster; see (Soltanolkotabi & Candès, 2012; Soltanolkotabi et al., 2014; Wu et al., 2021; Wang & Xu, 2016; Wang et al., 2019) regarding the $\ell_1$-minimization solutions, and (Dyer et al., 2013; Heckel & Bölcskei, 2015; You et al., 2016b; Tschannen & Bölcskei, 2018) pertaining to greedy search. However, SDP is neither necessary nor sufficient for perfect data segmentation. Indeed, as reported in many studies (Ng et al., 2001; Vershynin, 2018), a known type of similarity graph effectuating successful clustering is one configured with many intra-cluster and few inter-cluster edges. Evidently, this arises when the sparse regression misidentifies few neighbors, hence violating SDP. The downside of few falsely directed edges from cluster to cluster can be effectively compensated by spectral clustering (see (von Luxburg, 2007) for more discussions on this issue). Fulfillment of SDP does not guarantee correct clustering, especially when accompanied by meager neighbors. The reason is that, despite no inter-cluster edges, the resultant similarity graph is cut into excessively many isolated pieces; poor graph connectivity in this way tends to cause over-estimation of the number $L$ of clusters, leading to a large data clustering error[1] (Soltanolkotabi et al., 2014; Wu et al., 2021). The above facts altogether shed further light on the study of sparse regression for SSC. On the aspect of algorithm design, the efforts shall be particularly geared towards

---

[1] Once a similarity graph is constructed, e.g., using either $\ell_1$-minimization or greedy methods, one can further employ pruning schemes such as Yang et al. (2020); Qin et al. (2023) to obtain an updated graph with improved connectivity.

fast acquisition of plentiful neighbors so that the similarity graph can be fleshed out in a right configuration. As to neighbor recovery performance guarantees, much remains to be explored in search of new analysis criteria that are not so stringent as SDP, especially able to reflect neighbor recovery error. In the framework of two-step weighted $\ell_1$-minimization (Wu et al., 2021), the neighbor recovery rate was analyzed, and specifically the probability that the sparse regressor produces at least $k_t(>0)$ correct and at most $k_f(\geq 0)$ incorrect neighbors was found. Such a probabilistic characterization is intuitive and quite flexible in that it directly takes account of the general case with neighbor misidentification. For the special case of $k_f = 0$, i.e., error-free neighbor identification, it can reveal how much chance SDP stands with no less than $k_t$ recovered neighbors.

## 1.2 Paper Contributions

This paper aims at tackling the aforesaid challenges by revamping the OMP, considering its up-to-par performance, reduced computational complexity, and, most importantly, the inherent flexibility to boost neighbor acquisition. Pivoted on the Generalized OMP (GOMP), a prominent variant of OMP that is first introduced in the literature of CS (Wang et al., 2012) and allowed to identify multiple neighbors per iteration, we propose a new sparse regression scheme for SSC, together with an in-depth analytic study of its neighbor recovery performance guarantee. Specific technical contributions of this paper are summarized as follows.

- We propose to employ GOMP as an effective alternative to OMP for fast neighbor identification. In particular, we first point out that the deviation (caused by noise) of the residual vector from the desired ground truth subspace, pinned down by the Angle of Deviation (AoD), plays a pivotal role in neighbor identification. According to this fact, we then argue that GOMP, while digging out more neighbors per iteration, enjoys a smaller AoD, which makes it more resilient to noise corruption and able to achieve higher neighbor identification accuracy than OMP.

- Efficient stopping rules are crucial for greedy-based neighbor selection. If the algorithm stops early, we would end up with scant correct neighbors, yet if late, with overly many false ones; either case is apt to cause poor graph connectivity and eventual erroneous data clustering. Alongside the proposed GOMP we devise a new stopping rule geared toward fulfilling the dimension-aware property, that is, the algorithm is halted once the number of recovered neighbors is fairly close to the subspace dimension (in general this is the right moment to leave off neighbor search since the residual thereafter is typically dominated by noise). Mathematically, the proposed stopping rule judges the ratio of residual norms over consecutive two iterations against a threshold dependent on the ambient space dimension $n$, which is known once the dataset is given, and the number $p$ of neighbors identified per iteration that is at the designer's disposal. Advantageously, this dispenses with an extra offline estimation of the subspace dimension or noise strength as required in the existing solutions (Soltanolkotabi & Candès, 2012; Soltanolkotabi et al., 2014; Wu et al., 2021; Wang & Xu, 2016; Wang et al., 2019; Dyer et al., 2013; Heckel & Bölcskei, 2015; You et al., 2016b; Tschannen & Bölcskei, 2018).

- Capitalized on the work (Wu et al., 2021) and under the semi-random model (Soltanolkotabi & Candès, 2012; Soltanolkotabi et al., 2014), we conduct recovery rate analysis to bear out the claimed merits of the proposed GOMP scheme. Supposing that the ground truth subspaces are well-separated from each other, we first derive an analytic probability lower bound for the event that at least $k_m$ neighbors ($0 \leq k_m \leq p$) are correct (i.e., from the ground truth subspace) in the $m$th iteration. Such a local iteration-wise recovery rate result is then exploited to obtain the global recovery rate, namely, the probability lower bound for the event that at least $k_t$ correct neighbors in total are identified throughout. The obtained analytic formula shows that, for a large data size $N$ and small noise power, GOMP enjoys a higher correct neighbor recovery rate than the conventional OMP, thereby confirming GOMP can facilitate fast acquisition of many correct neighbors. Finally, we show that, with a high probability, the proposed stopping rule possesses the dimension-aware property.

- To implement the proposed GOMP method, the number $p$ of neighbors identified per iteration should be set beforehand. By further analyzing the obtained recovery rate formulae, the impact

of $p$ on the recovery rate is first discussed. For real-world datasets, oftentimes unable to meet the assumptions required by the semi-random model, recovery rate analysis is rather difficult to carry through. We, therefore, conduct numerical simulations to investigate the selection of $p$ aimed at fulfilling the dimension-aware property. Interestingly, both the recovery rate analysis for the semi-random model and our simulation study for real-world datasets indicate that a large $p$ is preferred when (i) the ground truth subspaces are well-separated from each other, or (ii) the data size $N$ is large. Some guidelines for selecting $p$ are suggested accordingly.

### 1.3 Connection to Previous Works

Efficient sparse regression for neighbor identification has played a pivotal role in the success of SSC. The standard $\ell_1$-minimization based method was first introduced in the landmark paper (Elhamifar & Vidal, 2013), and many related solutions have been proposed since then. In (Soltanolkotabi et al., 2014; Wu et al., 2021), iterative re-weighted $\ell_1$-minimization was adopted to further improve neighbor identification accuracy at the expense of higher algorithmic complexity. In (You et al., 2018; Peng et al., 2013; Matsushima & Brbic, 2019), computationally-efficient solutions under the framework of $\ell_1$-minimization were then proposed; the basic idea therein was to pre-process a small amount of data points to acquire side information about the ground truth subspaces, based on which a pruned dataset can then be used to reduce computations. Notably, (You et al., 2016a; Panagakis & Kotropoulos, 2014) utilized mixed-norm regularization to further enhance connectivity of the similarity graph. In contrast to $\ell_1$-minimization, greedy search such as OMP is one widely considered low-complexity neighbor identification scheme (Dyer et al., 2013; Heckel & Bölcskei, 2015; You et al., 2016b; Tschannen & Bölcskei, 2018). Recently in (Chen et al., 2018; Zhu et al., 2019), modified OMP algorithms aiming at improving network connectivity have also been proposed; the methods therein utilized the already established neighbor connections to narrow down the candidate neighbor list, overall promoting neighbor recovery and consequently better network connectivity. It is worth noting that all the existing OMP-based solutions identify one neighbor per iteration, and employ stopping rules calling for a knowledge of the subspace dimension or noise strength. Boosting neighbor recovery via multi-neighbor identification per iteration as well as the development of efficient stopping rules free from aforementioned side information is not yet addressed in the literature of SSC.

Regarding the study of mathematical performance guarantees, sufficient conditions for SDP under the $\ell_1$-minimization framework have been investigated in, say, (Elhamifar & Vidal, 2013; Soltanolkotabi & Candès, 2012) for the noiseless case, and (Soltanolkotabi et al., 2014; Wang & Xu, 2016; Wang et al., 2019) for the noisy case. Elhamifar & Vidal (2013) utilized convex geometry techniques to derive sufficient conditions tailored for specialized subspace orientations (e.g., disjoint or independent subspaces), while Soltanolkotabi & Candès (2012) dealt with the generalization to subspaces with a non-trivial intersection. Under noise corruption, Soltanolkotabi et al. (2014) leveraged certain approximation of the LASSO functional and the restricted isometry property of the noisy data matrix to estimate the probability that SDP holds; for LASSO sparse regression, Wang & Xu (2016) further investigated sufficient conditions that the regularization parameter must satisfy in order to guarantee SDP. Recently, Wu et al. (2021) extended the study in (Soltanolkotabi et al., 2014) to provide recovery rate analyses for general neighbor recovery events. For SSC employing greedy neighbor identification, Dyer et al. (2013) and You et al. (2016b) considered the noiseless scenario and derived sufficient conditions for SDP using convex geometry analysis; Tschannen & Bölcskei (2018) then extended the results in (Dyer et al., 2013) and (You et al., 2016b) to the Gaussian-noise setting, and derived probability lower bounds for the event the SDP holds. As far as we can see, all existing studies of performance guarantees for OMP-based SSC revolved around the fulfillment of SDP; the general case when neighbor misidentification occurs is left unaddressed.

To sum up, while GOMP has been investigated in CS (Wang et al., 2012), its application and potential impacts on SSC remain yet to be explored. This paper is a first step toward this goal. Thanks to multi-neighbor identification per iteration, the proposed GOMP method boosts neighbor recovery at lower algorithmic complexity as compared to conventional OMP. In addition, our newly developed stopping rule enjoys the dimension-aware property, thereby free from off-line subspace dimension estimation. Moreover, we leverage recovery rate analysis to derive mathematical performance guarantees for general neighbor re-

covery events. In view of the above achievements, our study of SSC with GOMP can contribute to more well-rounded literature on SSC under the framework of greedy neighbor selection.

The rest of this paper is organized as follows. Section 2 first explains why GOMP can outperform conventional OMP, and then introduces the foundations behind the proposed stopping rule. Afterwards, the algorithmic complexity of OMP and the proposed GOMP are analyzed. Section 3 analyzes the recovery rate and discusses the issue of selecting the number $p$ of recovered neighbors per iteration. Section 4 provides numerical simulations to verify our theoretical findings and discussions in Section 3. Section 5 presents the proofs of the main mathematical results. Finally, Section 6 concludes this paper. To ease reading, some detailed technical proofs are relegated to the appendix.

## 2 Proposed SSC-GOMP

This section introduces the proposed SSC-GOMP scheme. We first brief in Section 2.1 the reason why multiple neighbor recovery in each iteration is favored, in an attempt to motivate our GOMP proposal. In Section 2.2, we then encapsulate the foundations behind the proposed stopping rule. Finally, in Section 2.3 we provide algorithmic complexity analysis to justify the computational efficiency of the proposed GOMP as compared to OMP.

### 2.1 Why Multiple Neighbor Recovery per Iteration?

Recall that OMP iteratively identifies a neighbor each time as the data point when paired with the residual vector yields peak absolute inner product (see step 2 of Algorithm 3). Hence, the orientation of the residual vector in each iteration, in particular, the degree to which it deviates from the ground truth subspace, is important for accurate neighbor identification. To formalize this notion, assume that we are to build a neighbor list for the data point $\mathbf{y}_i$ coming from the cluster $\mathcal{Y}_k$ whose ground truth subspace is $\mathcal{S}_k$. Impaired by noise, the residual vector $\mathbf{r}_m^{(i)}$ computed in the $m$th iteration ($m \geq 1$) is perturbed outwards $\mathcal{S}_k$. If we write $\mathbf{r}_m^{(i)} = \mathbf{r}_{m,\parallel}^{(i)} + \mathbf{r}_{m,\perp}^{(i)}$, where $\mathbf{r}_{m,\parallel}^{(i)} \in \mathcal{S}_k$ and $\mathbf{r}_{m,\perp}^{(i)} \in \mathcal{S}_k^{\perp}$ (the orthogonal complement of $\mathcal{S}_k$), such perturbation can be pinned down by the angle of deviation (AoD)

$$\phi_m^i \triangleq \tan^{-1}(\|\mathbf{r}_{m,\perp}^{(i)}\|_2/\|\mathbf{r}_{m,\parallel}^{(i)}\|_2) \tag{3}$$

whereby a large $\phi_m^i$ means $\mathbf{r}_m^{(i)}$ severely deviates from $\mathcal{S}_k$. As the OMP algorithm iterates, perturbation of the residual would become increasingly severe. This is mainly because, owing to orthogonal projection (see step 3 of Algorithm 1), the current residual $\mathbf{r}_m^{(i)}$ is obtained by removing from the previous $\mathbf{r}_{m-1}^{(i)}$ the component lying in the subspace spanned by the already-selected neighbors, most of which are likely correct. Consequently, the magnitude $\|\mathbf{r}_{m,\parallel}^{(i)}\|_2$ of the component $\mathbf{r}_{m,\parallel}^{(i)} \in \mathcal{S}_k$ diminishes from iteration to iteration; instead, the term $\|\mathbf{r}_{m,\perp}^{(i)}\|_2$, which reflects the strength of projected misidentified neighbors (if any) plus noise onto $\mathcal{S}_k^{\perp}$, is typically non-decreasing with $m$. Put together, the cascade effect is, therefore, an increase in $\phi_m^i$ with $m$, rendering the residual $\mathbf{r}_m^{(i)}$ more and more prone to neighbor misidentification as the algorithm iterates. In this regard, a simple remedy for securing enough correct neighbors in few iterations (a "small $m$" is favored) is therefore to identify multiple neighbors per iteration, say, the $p$ data points ($p > 1$) corresponding to the largest $p$ absolute inner products. GOMP is therefore a potential solution to meet this goal. Using synthetic data, Fig. 1-(a) clearly demonstrates GOMP yields smaller average AoD than OMP thanks to fewer iterations[2]; this accordingly brings about higher neighbor identification accuracy, as illustrated in Fig. 1-(b) (this issue will be elaborated in Section 3). The reduction in the number of iterations can moreover reduce algorithmic complexity, which is potentially appealing in real-time applications. Indeed, the major computational bottleneck of the OMP algorithm is the orthogonal projection operation (step 3 in Algorithm 3). To recover $p(> 1)$ neighbors, conventional OMP requires $p$ iterations, hence $p$ orthogonal projections. Instead, GOMP calls for just one iteration, so one orthogonal projection only, and therefore is more computationally efficient (detailed algorithmic complexity comparison of OMP and GOMP is given at

---

[2]An analytic study of average AoD for GOMP/OMP is rather challenging, and is one of our future works.

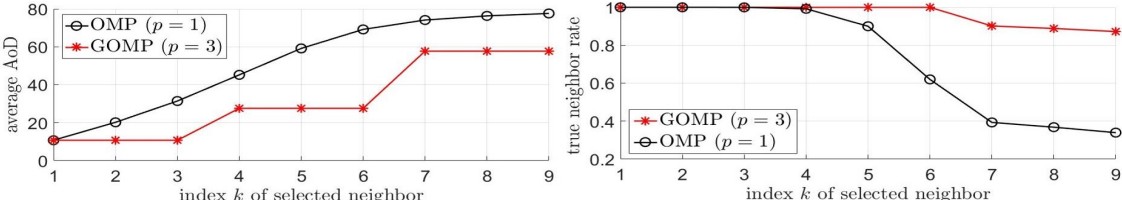

Figure 1: Comparison of GOMP and OMP in terms of average AoD and empirical recovery rate. We consider a synthetic data set of 135 vectors drawn from $L = 3$ orthogonal subspaces, each of a dimension 9, in an ambient domain $\mathbb{R}^{100}$; 45 data points per cluster. The data vectors are sampled uniformly from the intersection of the unit-sphere in $\mathbb{R}^{100}$ with the ground truth subspace and are corrupted by zero mean Gaussian noise with variance 0.04. For GOMP, $p = 3$ neighbors are picked per iteration as those when matched to the residual yielding the largest three absolute inner products. A total number of 9 neighbors are recovered using both OMP and GOMP. (a) Left: plot of average AoD upon detection of the $k$th neighbor, $1 \leq k \leq 9$. For GOMP, every three neighbors are detected in each iteration based on the same residual, leading to a staircase AoD curve. Clearly, GOMP results in smaller AoD thanks to fewer iterations. (b) Right: plot of the true neighbor rate, i.e., the fraction of true neighbors recovered, versus the index $k$ of the detected neighbor. Benefiting from smaller AoD, GOMP is seen to improve neighbor identification accuracy.

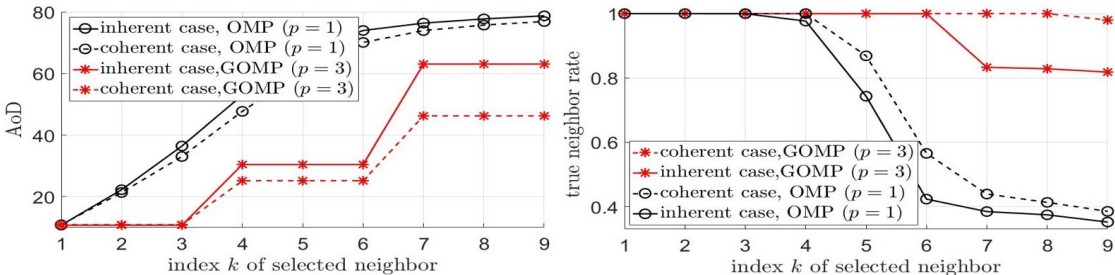

Figure 2: An illustration of AoD of GOMP and OMP in the presence of data coherence. We consider a synthetic dataset of 273 vectors drawn from $L = 3$ orthogonal subspaces, each of a dimension 9, in an ambient domain $\mathbb{R}^{100}$; 91 data points per cluster. To generate coherent data, in each cluster the first 46 data points are uniformly drawn from the intersection of the unit-sphere in $\mathbb{R}^{100}$ and the ground truth subspace and are corrupted by zero-mean Gaussian noise with variance 0.04, while the remaining 47th~91th points are "small perturbations" of the 2nd~46th points, by an additive zero mean Gaussian noise with variance $10^{-4}$. The inherent case, wherein all 91 data vectors per cluster are uniformly generated as above, is included as the baseline. We set $p = 3$ for GOMP. (a) Plots of AoD when identifying neighbors for the 1st data point. Since the neighboring points are pair-wise strongly coherent, it is highly likely a coherent pair is picked in each iteration. For both OMP and GOMP, data coherence even reduces AoD as compared to the inherent case. (b) The true neighbor recovery rates are higher, thanks to smaller AoD.

the end of this section). Considering all the above facts, we thus propose to adopt GOMP in place of OMP for neighbor identification.

**Remark:** Finally, we would like to comment on the the case when coherent data points are present. Assume that two correct neighbors of $\mathbf{y}_i$ are coherent (so that they are aligned toward the same direction), and both are selected by GOMP in the same iteration. As such, the dimension of the subspace identified throughout this iteration is $p - 1$, rather than $p$ as in the inherent case (i.e., all data points are sufficiently uncorrelated with each other). To obtain the new residual vector $\mathbf{r}_m^{(i)}$, the current residual $\mathbf{r}_{m-1}^{(i)}$ is projected onto the orthogonal complement of the span of the "already-selected" neighbors, which is of a higher dimension (one more) than the inherent case. In this way, the resultant signal component $\mathbf{r}_{m,\parallel}^{(i)}$ is better retained, leading to a larger $\|\mathbf{r}_{m,\parallel}^{(i)}\|_2$ and consequently a smaller AoD $= \tan^{-1}(\|\mathbf{r}_{m,\perp}^{(i)}\|_2 / \|\mathbf{r}_{m,\parallel}^{(i)}\|_2)$ (since the strength $\|\mathbf{r}_{m,\perp}^{(i)}\|_2$ of the projected misidentified neighbors (if any) plus noise onto $\mathcal{S}_k^{\perp}$, is roughly the same in both cases). This is confirmed by our experimental study as illustrated in Fig. 2. The above results indicate that GOMP can work well even in the presence of coherent data points.

## 2.2 Halt When There Are About as Many Neighbors as Subspace Dimension

Since the data point $\mathbf{y}_i$ comes from the $d_k$-dimensional ground truth subspace $\mathcal{S}_k$, a group of around $d_k$ true neighbors would reach a "critical mass" to well explain $\mathbf{y}_i$ and, if so, the residual from then on is highly apt to be dominated by noise, standing very little chance to uncover more true neighbors (this will be born out by our mathematical analysis in Section 3). Grounded on this fact, the algorithm is expected to be halted once $d_k$ neighbors or so are available. We shall recall the stopping rule widely adopted in the literature (You et al., 2016b; Tschannen & Bölcskei, 2018), which terminates neighbor search when either the number of iterations reaches a pre-set maximum $M$, or the residual becomes so small that $\|\mathbf{r}_m^{(i)}\|_2 \leq \tau$ for some threshold $\tau > 0$. Though implicit, this assumes the availability of prior knowledge about $M$ and $\tau$; the former is arguably all about the subspace dimension $d_k$ and the latter is closely related to the background noise strength, both of which can only be acquired through extra off-line estimation process. Considering that the dedicated overhead of parameter estimation could be costly, below we develop a new stopping rule which is per se aware of the subspace dimension without the need of knowing $M$ or $\tau$.

To introduce the proposed approach, let us write the data point under consideration as $\mathbf{y}_i = \mathbf{x}_i + \mathbf{e}_i$, where $\mathbf{x}_i$ is the noiseless signal point and $\mathbf{e}_i$ is the additive noise. The residual vector $\mathbf{r}_m^{(i)}$, which can be obtained from $\mathbf{y}_i$ through a sequence of $m$ orthogonal projections (step 3 of Algorithm 3), can be expressed as

$$\mathbf{r}_m^{(i)} = \prod_{l=1}^{m} \mathbf{P}_l \mathbf{y}_i = \prod_{l=1}^{m} \mathbf{P}_l(\mathbf{x}_i + \mathbf{e}_i) = \prod_{l=1}^{m} \mathbf{P}_l \mathbf{x}_i + \prod_{l=1}^{m} \mathbf{P}_l \mathbf{e}_i, \tag{4}$$

where $\mathbf{P}_l$ is the orthogonal projection onto the orthogonal complement of the subspace spanned by the already-selected neighbors up to the $l$th iteration. Assume that most of the recovered neighbors up to the $m$ iterations are from the correct subspace $\mathcal{S}_k$ so that the projected signal is very small and the residual $\mathbf{r}_m^{(i)}$ is strongly dominated by the projected noise, i.e., $\mathbf{r}_m^{(i)} \approx \prod_{l=1}^{m} \mathbf{P}_l \mathbf{e}_i$. This is typically the case once about as many neighbors as the subspace dimension are recovered. If the noise $\mathbf{e}_i$ is Gaussian, so is the residual $\mathbf{r}_m^{(i)}$, which, being nearly isotropic, tends to distribute its power evenly over all the dimensions (about $n - d_k$) of the orthogonal complement of the subspace spanned by all the already-selected neighbors; that is to say, each dimension shares a factor $1/(n - d_k)$ of the total power $\|\mathbf{r}_m^{(i)}\|_2^2$. During the $(m+1)$th iteration, $\mathbf{r}_{m+1}^{(i)}$ is then obtained from $\mathbf{r}_m^{(i)}$ by removing from it the components along the $p$ newly selected neighbors, implying that $\|\mathbf{r}_m^{(i)} - \mathbf{r}_{m+1}^{(i)}\|_2^2$ is close to $p \times \|\mathbf{r}_m^{(i)}\|_2^2/(n - d_k) \approx p \times \|\mathbf{r}_m^{(i)}\|_2^2/n$, in which the approximation makes sense since the subspace dimension $d_k$ is in general very small in comparison with the ambient space dimension $n$. Taking the square root and using the triangle inequality, we then obtain the following condition to halt the neighbor search

$$\|\mathbf{r}_m^{(i)}\|_2 - \|\mathbf{r}_{m+1}^{(i)}\|_2 \leq (\|\mathbf{r}_m^{(i)}\|_2 \sqrt{p})/\sqrt{n}, \tag{5}$$

or equivalently,

$$\frac{\|\mathbf{r}_{m+1}^{(i)}\|_2}{\|\mathbf{r}_m^{(i)}\|_2} \geq 1 - \sqrt{p/n}. \tag{6}$$

The proposed halting rule (6) is dimension-aware because, for most cases, it is triggered once the residual $\mathbf{r}_m^{(i)}$ is dominated by noise owing to the recovery of $d_k$ neighbors or thereabouts. Notably, the left-hand-side of (6) is a random variable and, thus, there is no way of ensuring (6) always holds. Instead, under Gaussian noise assumption it is shown in Section 5.3 that, with $m = \lceil d_L/p \rceil$, inequality (6) holds with a probability higher than $1 - 2pe^{-\sqrt{n/p}}$. As a result, as $p/n$ is close to zero (thus, $n/p$ is very large), the proposed stopping rule (6) can highly likely be triggered.[3] The proposed halting rule (6) is dimension-aware because, for most cases, it is triggered once the residual $\mathbf{r}_m^{(i)}$ is dominated by noise owing to the recovery of $d_k$ neighbors or thereabouts. We should moreover note that the left-hand-side of (6) admits the form of a residual norm ratio, which advantageously rids off the knowledge of noise strength. Indeed, since $\mathbf{r}_m^{(i)} \approx \prod_{l=1}^{m} \mathbf{P}_l \mathbf{e}_i$, we have

$$\|\mathbf{r}_m^{(i)}\|_2 \approx \|\prod_{l=1}^{m} \mathbf{P}_l \mathbf{e}_i\|_2 = \alpha_m \|\mathbf{e}_i\|_2, \text{ for some } 0 < \alpha_m < 1. \tag{7}$$

---

[3]Such a probabilistic interpretation of stopping criteria is not uncommon, e.g., the widely considered stopping rule $\|\mathbf{r}_m^{(i)}\|_2 < \tau$ (Tschannen & Bölcskei, 2018) cannot be guaranteed to meet deterministically, as $\|\mathbf{r}_m^{(i)}\|_2$ is a random variable.

---

**Algorithm 4** SSC-GOMP algorithm with the proposed data-dependent stopping criterion

---

**Input**: Observed data set $\mathcal{Y} = \{\mathbf{y}_1, \mathbf{y}_2, ..., \mathbf{y}_N\}$, data matrix $\mathbf{Y} = [\mathbf{y}_1...\mathbf{y}_N]$

    **for** $i = 1$ **to** $N$ **do**

        Let $m = 0$, $\mathbf{r}_0^{(i)} = \mathbf{y}_i$, $\mathbf{r}_{-1}^{(i)} = 2\mathbf{y}_i$, $\Lambda_0 = \phi$.

        **if** $(1 - \|\mathbf{r}_m^{(i)}\|_2/\|\mathbf{r}_{m-1}^{(i)}\|_2 \geq \sqrt{p/n})$ **then**

            1) $m \leftarrow m + 1$.

            2) $\Lambda_m = \Lambda_{m-1} \cup \mathcal{T}_m$, where $\mathcal{T}_m$ is the set of cardinality $p$ such that
$$|\langle \mathbf{y}_j, \mathbf{r}_{m-1}^{(i)}\rangle| \geq |\langle \mathbf{y}_q, \mathbf{r}_{m-1}^{(i)}\rangle|, \ \forall j \in \mathcal{T}_m, \ q \notin \mathcal{T}_m.$$

            3) $\mathbf{r}_m^{(i)} = (\mathbf{I} - \mathbf{Y}_{\Lambda_m}(\mathbf{Y}_{\Lambda_m}^T \mathbf{Y}_{\Lambda_m})^{-1}\mathbf{Y}_{\Lambda_m}^T)\mathbf{r}_{m-1}^{(i)}$.

        **end if**

        4) When the above procedure terminates with $M^{(i)}$ iterations, compute $\mathbf{c}_i^* = \underset{\mathbf{c}:\text{supp}(\mathbf{c})\subset\Lambda_{M^{(i)}-1}}{\arg\min} \|\mathbf{y}_i - \mathbf{Y}\mathbf{c}\|_2$.

        5) Normalize the column vector $\mathbf{c}_i^*$ to be unit-norm. Let $\overline{\mathbf{c}}_i^* = [c_{i,1}^*...c_{i,i-1}^* \ 0 \ c_{i,i+1}^*...c_{i,N}^*]^T \in \mathbb{R}^N$.

    **end for**

    6) Set $\mathbf{C} = [c_{i,j}] = [\overline{\mathbf{c}}_1^* \cdots \overline{\mathbf{c}}_N^*]$, and $\mathbf{G} = [g_{i,j}]$, where $g_{i,j} = |c_{i,j}| + |c_{j,i}|$.

    7) Form an $N$-point similarity graph in which the edge between nodes $i$ and $j$ has edge weight $g_{i,j}$.

    8) Apply spectral clustering to the similarity graph.

**Output**: Partition $\mathcal{Y} = \widehat{\mathcal{Y}}_1 \cup ... \cup \widehat{\mathcal{Y}}_{\widehat{L}}$.

---

The condition (7) motivates the stopping rule of the form You et al. (2016b); Tschannen & Bölcskei (2018), whereby the threshold $\tau$ depends on the noise strength $\|\mathbf{e}_i\|_2$. Using (7), the residual norm ratio in the proposed stopping rule (6) reads

$$\frac{\|\mathbf{r}_{m+1}^{(i)}\|_2}{\|\mathbf{r}_m^{(i)}\|_2} \approx \frac{\alpha_{m+1}}{\alpha_m}, \ \text{for some } 0 < \alpha_{m+1} < \alpha_m < 1. \tag{8}$$

which is clearly independent of the noise strength $\|\mathbf{e}_i\|_2$. The decision threshold in (6) assumes nothing more than a knowledge of the ambient dimension $n$, which is always known in advance, and the parameter $p$, which is at the designer's disposal. As such, it is free from the need of an extra offline estimation of the subspace dimension or noise strength, in marked contrast with the existing solutions You et al. (2016b); Tschannen & Bölcskei (2018). We summarize the proposed SSC-GOMP algorithm in Algorithm 4. We note from step 4) of Algorithm 4 that the support of $\mathbf{c}_i^*$ is $\Lambda_{M^{(i)}-1}$, namely, the indexes of "already-detected" neighbors up to the $(M^{(i)} - 1)$th iteration, thereby precluding those identified in the $M^{(i)}$th iteration, i.e., $j \in \mathcal{T}_{M^{(i)}}$. To see the reason behind this, recall from step 3) of Algorithm 4 that the residual in the $M^{(i)}$th iteration reads

$$\mathbf{r}_{M^{(i)}}^{(i)} = \mathbf{r}_{M^{(i)}-1}^{(i)} - \mathbf{Y}_{\Lambda_{M^{(i)}}}(\mathbf{Y}_{\Lambda_{M^{(i)}}}^T \mathbf{Y}_{\Lambda_{M^{(i)}}})^{-1}\mathbf{Y}_{\Lambda_{M^{(i)}}}^T \mathbf{r}_{M^{(i)}-1}^{(i)}. \tag{9}$$

If the stopping condition (6) is triggered, we have $\|\mathbf{r}_{M^{(i)}-1}^{(i)}\|_2 \approx \|\mathbf{r}_{M^{(i)}}^{(i)}\|_2$, which together with (9) implies

$$\mathbf{Y}_{\Lambda_{M^{(i)}}}(\mathbf{Y}_{\Lambda_{M^{(i)}}}^T \mathbf{Y}_{\Lambda_{M^{(i)}}})^{-1}\mathbf{Y}_{\Lambda_{M^{(i)}}}^T \mathbf{r}_{M^{(i)}-1}^{(i)} \approx \mathbf{0}. \tag{10}$$

Since $\Lambda_{M^{(i)}} = \Lambda_{M^{(i)}-1} \cup \mathcal{T}_{M^{(i)}}$, (10) implies that $\mathbf{r}_{M^{(i)}-1}^{(i)}$ is nearly orthogonal to $\mathbf{y}_j$, $j \in \mathcal{T}_{M^{(i)}}$, which are therefore unlikely to be neighbors of $\mathbf{y}_i$.

**Remark:** In case that the projected signal component $\prod_{l=1}^m \mathbf{P}_l\mathbf{x}_i$ is non-zero, our arguments of deriving (6) remains valid. This is because, as long as sufficiently many true neighbors are recovered, the term $\prod_{l=1}^m \mathbf{P}_l\mathbf{x}_i$ actually acts as a noise; thus, the residual $\mathbf{r}_m^{(i)} = \prod_{l=1}^m \mathbf{P}_l\mathbf{x}_i + \prod_{l=1}^m \mathbf{P}_l\mathbf{e}_i$ is indeed "noise-only" irrespective of whether $\prod_{l=1}^m \mathbf{P}_l\mathbf{x}_i$ is zero or not. To see this, let us stack the "already-selected" neighbors up to the $m$th iteration as a matrix $\widetilde{\mathbf{Y}}_m^{(i)} \in \mathbb{R}^{n \times pm}$, which admits the form $\widetilde{\mathbf{Y}}_m^{(i)} = \widetilde{\mathbf{X}}_m^{(i)} + \widetilde{\mathbf{E}}_m^{(i)}$, where $\widetilde{\mathbf{X}}_m^{(i)}$ and $\widetilde{\mathbf{E}}_m^{(i)}$ are the signal point and noise matrices, respectively. In case there are sufficiently many correct neighbors recovered, we have $\mathbf{x}_i = \widetilde{\mathbf{X}}_m^{(i)}\mathbf{c}$ for some $\mathbf{c}$. In this way, the projected signal component reads

Table 1: Comparison of GOMP and OMP in terms of running time ($\times 10^{-5}$ second).

| number of iteration | 1 | 2 | 3 | 4 | 5 | 6 | 7 | 8 | 9 |
|---|---|---|---|---|---|---|---|---|---|
| OMP $(p=1)$ | 3.2 | 3.4 | 3.4 | 3.5 | 3.6 | 3.7 | 3.8 | 3.9 | 4.0 |
| GOMP $(p=3)$ | 3.9 | 4.1 | 4.2 | $\times$ | $\times$ | $\times$ | $\times$ | $\times$ | $\times$ |

$$
\begin{aligned}
\prod_{l=1}^{m} \mathbf{P}_l \mathbf{x}_i &= \prod_{l=1}^{m-1} \mathbf{P}_l \underbrace{\left( \mathbf{I} - \widetilde{\mathbf{Y}}_m^{(i)} (\widetilde{\mathbf{Y}}_m^{(i)T} \widetilde{\mathbf{Y}}_m^{(i)})^{-1} \widetilde{\mathbf{Y}}_m^{(i)T} \right)}_{\mathbf{P}_m} \mathbf{x}_i \\
&= \prod_{l=1}^{m-1} \mathbf{P}_l \left( \mathbf{I} - \widetilde{\mathbf{Y}}_m^{(i)} (\widetilde{\mathbf{Y}}_m^{(i)T} \widetilde{\mathbf{Y}}_m^{(i)})^{-1} \widetilde{\mathbf{Y}}_m^{(i)T} \right) (\mathbf{x}_i - \widetilde{\mathbf{Y}}_m^{(i)} \mathbf{c}) \\
&= \prod_{l=1}^{m-1} \mathbf{P}_l \left( \mathbf{I} - \widetilde{\mathbf{Y}}_m^{(i)} (\widetilde{\mathbf{Y}}_m^{(i)T} \widetilde{\mathbf{Y}}_m^{(i)})^{-1} \widetilde{\mathbf{Y}}_m^{(i)T} \right) (\mathbf{x}_i - \widetilde{\mathbf{X}}_m^{(i)} \mathbf{c} - \widetilde{\mathbf{E}}_m^{(i)} \mathbf{c}) \\
&= \prod_{l=1}^{m-1} \mathbf{P}_l \left( \mathbf{I} - \widetilde{\mathbf{Y}}_m^{(i)} (\widetilde{\mathbf{Y}}_m^{(i)T} \widetilde{\mathbf{Y}}_m^{(i)})^{-1} \widetilde{\mathbf{Y}}_m^{(i)T} \right) (-\widetilde{\mathbf{E}}_m^{(i)} \mathbf{c}) = \prod_{l=1}^{m} \mathbf{P}_l (-\widetilde{\mathbf{E}}_m^{(i)} \mathbf{c}).
\end{aligned}
\tag{11}
$$

The residual vector accordingly becomes $\mathbf{r}_m^{(i)} = \prod_{l=1}^{m} \mathbf{P}_l (-\widetilde{\mathbf{E}}_m^{(i)} \mathbf{c}) + \prod_{l=1}^{m} \mathbf{P}_l \mathbf{e}_i = \prod_{l=1}^{m} \mathbf{P}_l (-\widetilde{\mathbf{E}}_m^{(i)} \mathbf{c} + \mathbf{e}_i)$, which is again a projected noise. Hence, the proposed stopping rule (6) makes sense and still works even $\prod_{l=1}^{m} \mathbf{P}_l \mathbf{e}_i$ is non-zero.

### 2.3 Algorithmic Complexity

We end this section by analyzing the algorithmic complexity of the proposed GOMP and OMP. To ease discussion, assume that $pM$ neighbors are to be recovered. Then OMP requires $pM$ iterations, in which the $m$th iteration has complexity $\mathcal{O}(nN + nm + nm^2 + m^3)$, $1 \leq m \leq pM$, respectively (Sturm & Christensen, 2012). Instead, GOMP requires just $M$ iterations, the $m$th iteration with complexity $\mathcal{O}(nN + npm + n(pm)^2 + (pm)^3)$, $1 \leq m \leq M$, respectively. Notice that, for $1 \leq m \leq M$, the $pm$th iteration of OMP has the same complexity as the $m$th iteration of GOMP. Hence, it is clear that the OMP scheme calls for additional $(p-1)M$ iterations, among which the complexity is $\mathcal{O}(nN + nm + nm^2 + m^3)$ for $1 \leq m \leq pM$, $m \neq p, 2p, \ldots, Mp$. To further justify the low-complexity advantage of the proposed GOMP, we consider the data set used in Fig. 1 and compare the running time of the proposed GOMP $(p=3)$ and OMP $(p=1)$; the results are listed in Table 1. From the table, we observe the following.

- Running time increases with $m$, the number of iteration. This is mainly because, as $m$ increases, the number of the identified neighbors, and hence the dimension of the least squares problem, in step 4) increases, leading to longer running time.

- Even though the $m$th iteration of GOMP and the $pm$th iteration of OMP have the same order of complexity, the running time of the former is slightly higher. The reason behind this is that, while OMP computes the maximal absolute inner product, GOMP seeks the first $p$ largest ones: this entails additional sorting efforts and therefore higher running time. Despite this, GOMP involves $(p-1)M$ less iterations, and overall less running time, than OMP. Based on Table 1, the running time of GOMP is about 37.5% of that of OMP.

## 3 Theoretical Results

In this section, we present the recovery rate analysis for the proposed SSC-GOMP. Under the semi-random model assumption, Section 3.1 first derives analytic recovery rate formulae. Section 3.2 then discusses the guidelines for selecting the number $p$ of recovered neighbors per iteration.

### 3.1 Recovery Rate Analysis

To formalize matters, the data vector is assumed to follow the standard additive noise model, that is,

$$\mathbf{y}_i = \mathbf{x}_i + \mathbf{e}_i, 1 \leq i \leq N, \tag{12}$$

where $\mathbf{x}_i \in \mathbb{R}^n$ is the unit-norm noiseless signal vector and $\mathbf{e}_i \in \mathbb{R}^n$ is the noise. The analyses below are built on the popular semi-random model (Soltanolkotabi & Candès, 2012; Soltanolkotabi et al., 2014; Wu et al., 2021; Wang & Xu, 2016), i.e., the ground truth subspaces $\mathcal{S}_1, ..., \mathcal{S}_L$ in the partition (1) are fixed but otherwise unknown, whereas the data vectors and noise are random. Such a model is widely used in the theoretical study of SSC, thanks to its interpretability and amiability to analysis. Similar to the previous works (Wu et al., 2021; Wang et al., 2019; Heckel & Bölcskei, 2015; Tschannen & Bölcskei, 2018), the following assumptions are made in the sequel.

**Assumption 1.** *For each $1 \leq i \leq N$, the signal vector $\mathbf{x}_i \in \mathbb{R}^n$ is uniformly sampled from $\mathcal{B}_k$, where $\mathcal{B}_k \triangleq \{\mathbf{x} \mid \mathbf{x} \in \mathbb{R}^n, \|\mathbf{x}\|_2 = 1\} \cap \mathcal{S}_k$ is the intersection of the unit sphere with the subspace $\mathcal{S}_k$.* □

**Assumption 2.** *For each $1 \leq i \leq N$, the noise $\mathbf{e}_i \in \mathbb{R}^n$, $1 \leq i \leq N$, are i.i.d. Gaussian random vectors with zero mean and covariance matrix $(\sigma^2/n)\mathbf{I}$, and are independent of the signal vectors $\mathbf{x}_i$'s.* □

To gauge the degree to which two subspaces are separated away from each other, we recall the affinity between two distinct subspaces $\mathcal{S}_k$ and $\mathcal{S}_l$ that is defined to be (Soltanolkotabi & Candès, 2012)

$$aff(\mathcal{S}_k, \mathcal{S}_l) \triangleq \frac{\|\mathbf{U}_k^T \mathbf{U}_l\|_F}{\sqrt{\min\{d_k, d_l\}}}, \tag{13}$$

where columns of $\mathbf{U}_k$ ($\mathbf{U}_l$, respectively) form an orthonormal basis for $\mathcal{S}_k$ ($\mathcal{S}_l$, respectively).

**Assumption 3.** *The subspace affinity satisfies*

$$\max_{k, k \neq l} aff(\mathcal{S}_k, \mathcal{S}_l) + \frac{9\sqrt{3}d_L(1 + \sigma)}{(8 - 12\sigma)\sqrt{n - d_L} \log N} \leq \frac{\tau}{4 \log N}, \tag{14}$$

*where $0 < \tau < 1$.* □

Notably, Assumptions 3 guarantees that different subspaces are well separated from each other; affinity conditions akin to (14) are also needed in many existing studies of performance guarantees for SSC (Wu et al., 2021; Wang et al., 2019; Heckel & Bölcskei, 2015; Tschannen & Bölcskei, 2018). Without loss of generality, we assume in the sequel $\mathbf{x}_N \in \mathcal{S}_L$, therefore $\mathbf{y}_N \in \mathcal{Y}_L$, and our goal is to identify a neighbor group of $\mathbf{y}_N$. Under the above three assumptions it can be shown that GOMP stands a high chance to recover many true neighbors in each of the first $\lceil d_L/p \rceil$ iterations. More precisely, we have the following theorem.

**Theorem 1.** *(Iteration-Wise Recovery Rate): Let $\{k_1, k_2, ..., k_M\}$ be a sequence of integers satisfying $0 \leq k_m \leq p$, for $1 \leq m \leq M \leq \lceil d_L/p \rceil$. Under Assumptions 1 to 3, the proposed SSC-GOMP obtains at least $k_m$ true neighbors at the mth iteration, $1 \leq m \leq M$, with a probability exceeding*

$$1 - Ne^{-n/8} - 6\left(\frac{\sigma}{\sqrt{\pi}}\right)^{d_L - p(M-1)} - \sum_{m=1}^{M}\left[\left(\left(\frac{2e(N - |\mathcal{Y}_L|)}{(p - k_m + 1)N^{8 \log N/d_L}}\right)^{p - k_m + 1}\right.\right. \\ \left.\left. + \left(\sqrt{\frac{2}{\pi}}\tau\right)^{|\mathcal{Y}_L| - d_L - k_m}\left(\frac{e(|\mathcal{Y}_L| - 1)}{k_m - 1}\right)^{k_m - 1} + \frac{4 + 2c}{N^2}\right)\mathbf{1}(k_m > 0)\right], \tag{15}$$

*where $\mathbf{1}(\bullet)$ is the indicator function, $c > 0$ is a constant, and $d_L$ is the dimension of the subspace $\mathcal{S}_L$.*

*Proof*: See Section 5.1. □

Further scrutiny reveals the lower bound (15) is high in most practical cases. Indeed, with a large ambient dimension $n$ and small noise level $\sigma$, the second and third terms in (15) are kept small; regarding the last summation, the first and third terms scale like $N^{[1 - (8 \log N/d_L)](p - k_m + 1)}$ and $N^{-2}$, respectively, whereas the

second term decays exponentially fast as $|\mathcal{Y}_L|$ increases, thanks to $0 < \tau < 1$ (see Assumption 3). When specialized to $p = 1$ and $k_m = 1$ for all $1 \leq m \leq M$, i.e., the case with conventional OMP subject to all detected neighbors being true, the lower bound (15) then reads

$$1 - Ne^{-n/8} - 6\left(\frac{\sigma}{\sqrt{\pi}}\right)^{d_L - p(M-1)} - M\left[\frac{2e(N - |\mathcal{Y}_L|)}{N^{8\log N/d_L}} + \left(\sqrt{\frac{2}{\pi}}\tau\right)^{|\mathcal{Y}_L| - d_L - 1} + \frac{4 + 2c}{N^2}\right]. \tag{16}$$

Notably, a probability lower bound akin to (16) was also reported in (Tschannen & Bölcskei, 2018), which addressed sufficient conditions ensuring correct neighbor recovery. Based on the iteration-wise recovery rate result given in Theorem 1, the following corollary further establishes the global recovery rate, namely, the probability of the event that GOMP succeeds in recovering a specified total number of true neighbors throughout the $M \leq \lceil d_L/p \rceil$ iterations.

**Corollary 1.** *(Global Recovery Rate): Let $0 \leq k_t \leq pM$ be an integer and write $k_t = Mq_t + r_t$, where $0 \leq r_t \leq M - 1$. Under the same setup as in Theorem 1, GOMP can recover at least $k_t$ true neighbors in total throughout $M(\leq \lceil d_L/p \rceil)$ iterations with a probability higher than*

$$1 - Ne^{-n/8} - 6\left(\frac{\sigma}{\sqrt{\pi}}\right)^{d_L - p(M-1)}$$

$$- r_t\left[\left(\frac{2e(N - |\mathcal{Y}_L|)}{(p - q_t)N^{8\log N/d_L}}\right)^{p - q_t} + \left(\sqrt{\frac{2}{\pi}}\tau\right)^{|\mathcal{Y}_L| - d_L - q_t - 1}\left(\frac{e(|\mathcal{Y}_L| - 1)}{q_t}\right)^{q_t} + \frac{4 + 2c}{N^2}\right] \tag{17}$$

$$- (M - r_t)\left[\left(\frac{2e(N - |\mathcal{Y}_L|)}{(p - q_t + 1)N^{8\log N/d_L}}\right)^{p - q_t + 1} + \left(\sqrt{\frac{2}{\pi}}\tau\right)^{|\mathcal{Y}_L| - d_L - q_t}\left(\frac{e(|\mathcal{Y}_L| - 1)}{q_t - 1}\right)^{q_t - 1} + \frac{4 + 2c}{N^2}\right]\mathbf{1}(q_t > 0).$$

*Proof* : See Section 5.2. $\qquad\qquad\qquad\qquad\qquad\qquad\qquad\qquad\qquad\qquad\qquad\qquad\qquad\qquad\qquad\square$

By following our examination of (15), it is easy to check the probability lower bound (17) is high in most practical cases. Moreover, with the aid of (17), GOMP is seen to yield a higher true neighbor recovery rate as compared with the conventional OMP. To better illustrate this, we assume without loss of generality that a group of $pM$ neighbors is to be found, while demanding at least $k_t = kM$ true neighbors, for some $1 < k \leq p$ (that is to say, $k/p$ of the total neighbors are true). Hence, GOMP requires $M$ iterations, and OMP $p$ times more. Under this setting, the lower bound (17) for GOMP reads

$$1 - Ne^{-n/8} - 6\left(\frac{\sigma}{\sqrt{\pi}}\right)^{d_L - pM + p} - M\left(\frac{2e(N - |\mathcal{Y}_L|)}{(p - k + 1)N^{8\log N/d_L}}\right)^{p - k + 1}$$

$$- M\left(\sqrt{\frac{2}{\pi}}\tau\right)^{|\mathcal{Y}_L| - d_L - k}\left(\frac{e(|\mathcal{Y}_L| - 1)}{k - 1}\right)^{k - 1} - \frac{(4 + 2c)M}{N^2}, \tag{18}$$

whereas for OMP the bound becomes

$$1 - Ne^{-n/8} - 6\left(\frac{\sigma}{\sqrt{\pi}}\right)^{d_L - pM + 1} - \frac{2kMe(N - |\mathcal{Y}_L|)}{N^{8\log N/d_L}} - kM\left(\sqrt{\frac{2}{\pi}}\tau\right)^{|\mathcal{Y}_L| - d_L - 1} - \frac{(4 + 2c)kM}{N^2}. \tag{19}$$

Note that (18) differs from (19) in the last four terms, among which it is clear that $6\left(\sigma/\sqrt{\pi}\right)^{d_L - pM + p} < 6\left(\sigma/\sqrt{\pi}\right)^{d_L - pM + 1}$ for small noise level $\sigma$, and $\frac{(4 + 2c)M}{N^2} < \frac{(4 + 2c)kM}{N^2}$ because $k > 1$. Since the $4^{\text{th}}$ term $M\left(\frac{2e(N - |\mathcal{Y}_L|)}{(p - k + 1)N^{8\log N/d_L}}\right)^{p - k + 1}$ in (18) and the $4^{\text{th}}$ term $\frac{2kMe(N - |\mathcal{Y}_L|)}{N^{8\log N/d_L}}$ in (19) scale like $N^{[1 - (8\log N/d_L)](p - k + 1)}$ and $N^{[1 - (8\log N/d_L)]}$, respectively, both vanish whenever the data size $N$ is very large. Also, since $\tau < 1$, both the $5^{\text{th}}$ term $M\left(\sqrt{\frac{2}{\pi}}\tau\right)^{|\mathcal{Y}_L| - d_L - k}\left(\frac{e(|\mathcal{Y}_L| - 1)}{k - 1}\right)^{k - 1}$ in (18) and the $5^{\text{th}}$ term $kM\left(\sqrt{\frac{2}{\pi}}\tau\right)^{|\mathcal{Y}_L| - d_L - 1}$ in (19) decay exponentially fast to zero as the cluster size $|\mathcal{Y}_L|$ grows, therefore negligible. Accordingly, when the

subspaces are well separated from each other and the data/cluster size is large enough, GOMP improves the true neighbor recovery rate.

Theorem 1 and Corollary 1 specify the neighbor recovery rate up to the $\lceil d_L/p \rceil$th iteration, upon which GOMP is expected to be halted by the proposed stopping rule (6) thanks to the availability of about $d_L$ recovered neighbors. Such a conjecture is provably true with a high probability, as established in the next theorem.

**Theorem 2.** *Under the same setup as in Theorem 1, GOMP accompanied by the proposed stopping rule (6) halts neighbor search till the $\lceil d_L/p \rceil$th iteration with a probability exceeding*

$$
\begin{aligned}
& 1 - Ne^{-n/8} - 6\left(\frac{\sigma}{\sqrt{\pi}}\right)^{d_L - p(\lceil d_L/p \rceil - 1)} - 2pe^{-\sqrt{n/p}} \\
& - \left\lceil \frac{d_L}{p} \right\rceil \left[ \frac{2e(N - |\mathcal{Y}_L|)}{N^{8 \log N/d_L}} + \left(\sqrt{\frac{2}{\pi}}\tau\right)^{|\mathcal{Y}_L| - d_L - p}\left(\frac{e(|\mathcal{Y}_L| - 1)}{p - 1}\right)^{p-1} + \frac{4 + 2c}{N^2} \right].
\end{aligned}
\tag{20}
$$

*Proof* : See Section 5.3.  $\square$

Under Assumptions 1 to 3, Theorem 2 provides an analytic probability lower bound for the event that the proposed stopping rule (6) is aware of the subspace dimension, i.e., it terminates the GOMP algorithm when the number of recovered neighbors is close to the subspace dimension $d_L$. We further remark that when the noise power increases slightly further, the dimension of the span of data points in $\mathcal{Y}_L$ would exceed $d_L$; accordingly, the algorithm would stop with more than $\lceil d_L/p \rceil$ iterations in order to recover more neighbors commensurate with the increased dimension. As the noise power grows higher, the residual is soon dominated by noise so as to trigger the stopping condition (6), rendering the algorithm terminated in fewer than $\lceil d_L/p \rceil$ iterations with scant neighbors. This phenomenon will be seen in our simulation study.

### 3.2 On Selection of $p$

The performance of the proposed GOMP algorithm depends on the parameter $p$, i.e., the number of identified neighbors per iteration. With the aid of the recovery rate analyses for the semi-random model, below we first discuss the selection of $p$ aimed at improving the recovery rate. The results will shed light on the selection of $p$ for practical datasets.

Recall the assertion of Theorem 2 that, with a high probability, the proposed stopping rule (6) halts the algorithm once $d_L$ neighbors or so are recovered and, if so, the GOMP algorithm yields a high global recovery rate throughout all iterations according to Corollary 1. Hence, a natural criterion for selecting $p$ is to maximize both the probability lower bounds (17) and (20); in this way, the proposed algorithm stands a high chance of recovering many true neighbors and only few false neighbors. However, we should note that both probability lower bounds are rather complicated functions in $p$ and the ground truth parameters (such as the data size $N$, subspace dimension $d_L$, cluster size $|\mathcal{Y}_L|$, noise power $\sigma$, and the subspace affinity $\tau$). An explicit and tractable rule for choosing $p$ toward recovery rate enhancement is therefore very difficult to obtain. Even if such a solution can be found, it necessarily depends on the ground truth parameters, which are nonetheless unknown to the designers. Despite this, by analyzing the lower bounds (17) and (20) we can still summarize certain interesting properties of the recovery rate as the parameter $p$ varies.

For this, let us examine each individual term in the probability lower bounds (17) and (20) to see how they change with $p$. Suppose that a total number $M_t = pM$ of neighbors are recovered, among which at least $k_t(> 0)$ are true. To ease discussions, we assume that $k_t \geq M$ so that $q_t = \lfloor k_t/M \rfloor > 0$, where $\lfloor \bullet \rfloor$ is the floor function; hence, on average at least $q_t$ true neighbors are found in each iteration. We can first observe from (17) the followings:

a) Clearly, the 3$^{\text{rd}}$ term $-6(\sigma/\sqrt{\pi})^{d_L - M_t + p}$ in the lower bound (17) increases with $p$ when the noise level is so small that $\sigma < \sqrt{\pi}$.

b) Following the discussions below (19), the 4$^{\text{th}}$ and 7$^{\text{th}}$ terms of (17) scale like $-N^{[1-(8 \log N/d_L)](p-q_t)}$ and $-N^{[1-(8 \log N/d_L)](p-q_t+1)}$, respectively. Since

$$
p - q_t = p - \lfloor k_t/(M_t/p) \rfloor = p - \lfloor p(k_t/M_t) \rfloor,
\tag{21}
$$

which increases with $p$, we conclude that the 4$^{\text{th}}$ and 7$^{\text{th}}$ terms of (17) also increases with $p$.

c) To check the 5$^{\text{th}}$ and 8$^{\text{th}}$ terms of (17), we first note that (i) the subspace affinity bound $\tau < 1$ (see (14)), thereby $\tau\sqrt{2/\pi} < 1$, and (ii) since the number $k_t$ of recovered true neighbors never exceeds the cluster size $|\mathcal{Y}_L|$, we must have $|\mathcal{Y}_L| > k_t > q_t = \lfloor k_t/M \rfloor$, leading to $e(|\mathcal{Y}_L| - 1)/q_t > 1$. Hence, it can be readily seen that the 5$^{\text{th}}$ term $-r_t\left(\sqrt{\frac{2}{\pi}}\tau\right)^{|\mathcal{Y}_L|-d_L-q_t-1}\left(\frac{e(|\mathcal{Y}_L|-1)}{q_t}\right)^{q_t}$ and the 8$^{\text{th}}$ term $-(M-r_t)\left(\sqrt{\frac{2}{\pi}}\tau\right)^{|\mathcal{Y}_L|-d_L-q_t}\left(\frac{e(|\mathcal{Y}_L|-1)}{q_t-1}\right)^{q_t-1}$ decrease with $q_t$. Notably, since $q_t = \lfloor p(k_t/M_t) \rfloor$ grows as $p$ increases, we therefore conclude that the 5$^{\text{th}}$ and 8$^{\text{th}}$ terms decrease with $p$.

d) Finally, the sum of the 6$^{\text{th}}$ and 9$^{\text{th}}$ terms of (17) equals $\frac{-M(4+2c)}{N^2} = \frac{-M_t(4+2c)}{pN^2}$, which increases with $p$.

Combining (a)$\sim$(d), we know that the 3$^{\text{rd}}$, 4$^{\text{th}}$, 6$^{\text{th}}$, 7$^{\text{th}}$, and 9$^{\text{th}}$ terms of (17) increase with $p$, whereas the 5$^{\text{th}}$ and 8$^{\text{th}}$ terms decrease with $p$. In particular, if the subspace affinity $\tau$ is very small, the 5$^{\text{th}}$ and 8$^{\text{th}}$ terms in (17) can be neglected so that the lower bound (17) increases with $p$. Hence, as long as the ground truth subspaces are well-separated from each other, a large $p$ is preferred. On the other hand, we consider the scenario that the data size $N$ and cluster size $|\mathcal{Y}_L|$ are large; therefore the 4$^{\text{th}}$, 5$^{\text{th}}$, 7$^{\text{th}}$, and 8$^{\text{th}}$ terms in the lower bound (17) are vanishingly small (see also the discussions below (19)). The lower bound (17) is then reduced to

$$1 - Ne^{-n/8} - 6\left(\frac{\sigma}{\sqrt{\pi}}\right)^{d_L-M_t+p} - \frac{M_t(4+2c)}{pN^2}. \tag{22}$$

Clearly, with $M_t$, $N$, $d_L$, and $\sigma$ being fixed, the lower bound (22) increases with $p$. Hence, we can also conclude that, as long as the data size $N$ and cluster size $|\mathcal{Y}_L|$ are large enough, a large $p$ is preferred. Next, we consider the lower bound (20). To ease discussion, we assume that $p$ divides $d_L$. Based on our analyses on the lower bound (17) (see the discussions (a) to (d) above) we note that:

e) The 3$^{\text{rd}}$ term $-6(\sigma/\sqrt{\pi})^{d_L-p(\lceil d_L/p \rceil-1)} = -6(\sigma/\sqrt{\pi})^p$ increases with $p$ when the noise level is so small that $\sigma < \sqrt{\pi}$.

f) The 4$^{\text{th}}$ term $-2pe^{-\sqrt{n/p}}$ is negligible since the ambient dimension $n \gg p$.

g) Clearly, the 5$^{\text{th}}$ term $-\left\lceil\frac{d_L}{p}\right\rceil\frac{2e(N-|\mathcal{Y}_L|)}{N^{8\log N/d_L}}$ increases with $p$.

h) Since $\tau\sqrt{2/\pi} < 1$ and $e(|\mathcal{Y}_L| - 1)/p > 1$ (see discussion (c)), $\left(\sqrt{\frac{2}{\pi}}\tau\right)^{|\mathcal{Y}_L|-d_L-p}\left(\frac{e(|\mathcal{Y}_L|-1)}{p-1}\right)^{p-1}$ increases with $p$; obviously $\lceil d_L/p \rceil$ decreases with $p$. Hence, it is hard to tell whether the 6$^{\text{th}}$ term $-\lceil\frac{d_L}{p}\rceil\left(\sqrt{\frac{2}{\pi}}\tau\right)^{|\mathcal{Y}_L|-d_L-p}\left(\frac{e(|\mathcal{Y}_L|-1)}{p-1}\right)^{p-1}$ of (20) increases with $p$ or not.

i) Finally, the 7$^{\text{th}}$ term $-\lceil\frac{d_L}{p}\rceil\frac{4+2c}{N^2}$ increases with $p$.

By (e), (f), (g), and (i), we know that the 3$^{\text{rd}}$, 4$^{\text{th}}$, 5$^{\text{th}}$, 7$^{\text{th}}$ terms of (20) increase with $p$. Still, if the subspace affinity $\tau$ is very small, the 6$^{\text{th}}$ term of (20) is negligible, leading to the conclusion that (20) increases with $p$. Hence, a large $p$ is favored for well-separated ground truth subspaces. On the other hand, when $N$ and $|\mathcal{Y}_L|$ are large, the 5$^{\text{th}}$ and 6$^{\text{th}}$ terms are vanishingly small (again, see the discussions below (19)) so that (20) increases with $p$. As a result, a large $p$ is preferred whenever the data size $N$ and cluster size $|\mathcal{Y}_L|$ are large. It is worth noting that the inferred tendency of (20) as $p$ varies totally agrees with that of (17). Finally, we should note that the cluster size $|\mathcal{Y}_L|$ is unknown beforehand. For balanced datasets, wherein all clusters are about equally large, $|\mathcal{Y}_L|$ is typically in direct proportion to the data size $N$; in this case, we then would like to increase $p$ for large-scale datasets. We can therefore conclude that, for well-separated subspaces (small affinity) or large-scale balanced datasets (large $N$ and $|\mathcal{Y}_L|$), a large $p$ is preferred (the same tendency is also seen in our simulation study using both synthetic and real-world datasets). This is expected since, in the former case, the data point under consideration is surrounded by many true neighbors, and in the latter case tends to be strongly correlated with many neighbors. Either way, the residual would quickly diminish in

just few iterations, and one should therefore adopt a large $p$ to boost neighbor recovery before the residual is depleted of the component from the desired subspace.

We should note however that our recovery rate analyses are developed under the semi-random model, that is, the signal points are uniform (Assumption 1) and noise is Gaussian (Assumption 2), which are seldom met in practice. Hence, for real-world datasets, performance guarantees similar to the semi-random model can hardly be obtained. Without such analytic metrics, a simple and intuitively reasonable way of judging a good $p$ would be the fulfillment of the dimension-aware property; this is because the availability of about $d_L$ neighbors is likely to begat a similarity graph with good connectivity, i.e., sufficiently many intra-cluster edges and few inter-cluster edges. Based on our simulation study (see Section 4 for more details) it is observed that the dimension-aware property tends to hold for a wide range of $p$ when datasets enjoy a small subspace affinity, such as the Coil-100 (Nene et al., 1996) and MNIST datasets (LeCun et al., 1998), but seems to fail for datasets with medium-to-large subspace affinities, such as the Devanagari (Acharya et al., 2015) and Extended Yale B (Lee et al., 2005) datasets. In particular, the Devanagari dataset is subject to large noise corruption and prefers a large $p$; contrarily, the Extended Yale B is under small noise corruption and prefers a small $p$. We would therefore conclude that the dimension-aware property holds when the subspace affinity is small; this is because there would then exist ample intra-cluster neighbors for the data point under consideration, making it highly likely to find about $d_L$ correct neighbors no matter how many are recovered in each iteration. The above simulation findings also suggest that the selection of $p$ is highly dependent on the affinity and noise power, which are nonetheless known unless additional offline estimation is further conducted. In the absence of such side information, we would then propose to select a large $p$ when the data size $N$ is large; this is justified by our simulation results and also agrees with our previous claim in the semi-random model case. In particular, we would suggest choosing $p = 2, 3$ for small-scale datasets such as Extended Yale B, Coil-100, and NCKU human face (Chen & Lien, 2009) datasets (all with thousands of data points or less), while $p = 5, 6$ for large-scale datasets, e.g. MNIST, Devanagari, and Cifar-10 (Krizhevsky, 2009) datasets (with tens of thousands of data points or more).

**Remark:** In the context of sparse signal recovery via GOMP, the selection of $p$, in general, depends on the sparsity level, but the development of explicit rules for choosing $p$ still remains a difficult open problem (Fu et al., 2022).

## 4   Experimental Results

In this section, numerical simulations based on both synthetic data and real-world data are used to corroborate our theoretical study and illustrate the performance of the proposed GOMP method. Synthetic data generated are similar to (Elhamifar & Vidal, 2013; Wu et al., 2021). The ground truth is a union of three subspaces $\mathcal{S}_1$, $\mathcal{S}_2$ and $\mathcal{S}_3$ of $\mathbb{R}^{350}$, with an equal subspace dimension $d_1 = d_2 = d_3 = 6$; separation between two distinct subspaces is gauged by the subspace affinity defined in (13). Noiseless signal points are uniformly drawn at random from each subspace and are corrupted by Gaussian noise with zero mean and standard deviation $\sigma$. The sampling density is defined to be $\phi_k \triangleq |\mathcal{Y}_k|/d_k$, $1 \le k \le 3$. Regarding the neighbor identification performance metrics, we consider the true neighbor rate (TNR) (Wu et al., 2021), that is,

$$\text{TNR} \triangleq \frac{\sum_{(i,j) \in \mathcal{T}} \mathbf{1}(c_{i,j}^* \neq 0)}{\sum_{1 \le i,j \le N} \mathbf{1}(c_{i,j}^* \neq 0)}, \ \mathcal{T} \triangleq \{(i,j)|\mathbf{y}_i, \mathbf{y}_j \in \mathcal{Y}_l, \text{ for some } 1 \le l \le L\}, \tag{23}$$

where $c_{i,j}^*$ is the $j$th entry of the computed sparse representation vector $\mathbf{c}_i^*$ using GOMP/OMP (see step 4 of Algorithm 3 and Algorithm 4), and the average number of recovered neighbors (ANRN)

$$\text{ANRN} \triangleq \sum_{i=1}^{N} \|\mathbf{c}_i^*\|_0/N, \tag{24}$$

which assesses the ability to boost neighbor acquisition. As in (Soltanolkotabi & Candès, 2012; Soltanolkotabi et al., 2014; Wu et al., 2021; You et al., 2018; Matsushima & Brbic, 2019), the global clustering performance is evaluated based on the correct clustering rate (CCR), defined as

$$\text{CCR} \triangleq (\# \text{ of correctly clustered data points})/N. \tag{25}$$

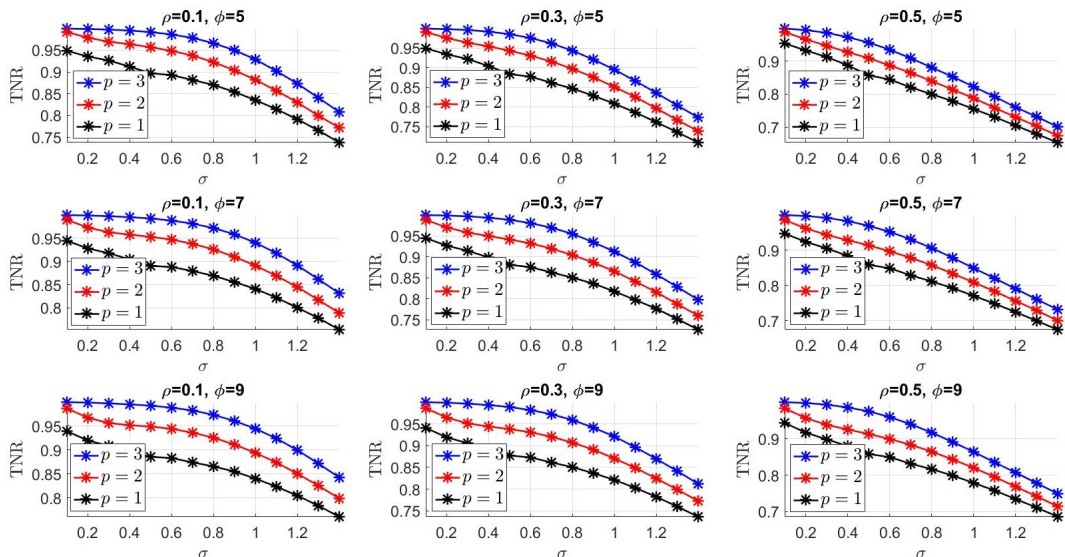

Figure 3: TNR versus noise standard deviation $\sigma$ for nine different pairs of subspace affinity $\rho$ and sample density $\phi$; assume the subspace dimension is known beforehand and the number of iterations is preset to be $M = \lceil 6/p \rceil$ $(p = 1, 2, 3)$.

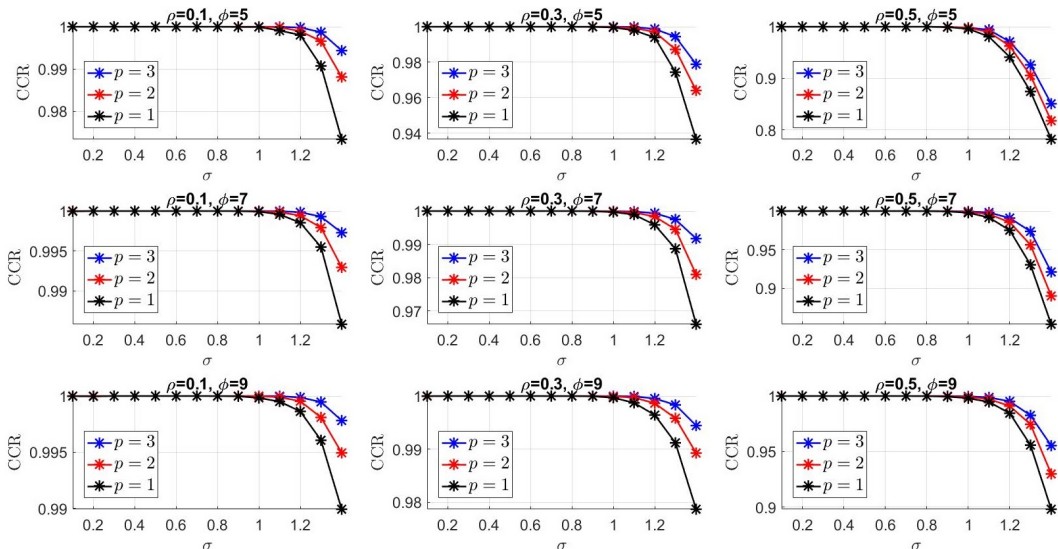

Figure 4: CCR versus noise standard deviation $\sigma$ for nine different pairs of subspace affinity $\rho$ and sample density $\phi$; assume the subspace dimension is known beforehand and the number of iterations is preset to be $M = \lceil 6/p \rceil$ $(p = 1, 2, 3)$.

## 4.1 Synthetic Data

We first use synthetic data to test the performance of the proposed method. To ease illustration, we consider the case that $aff(\mathcal{S}_k, \mathcal{S}_l) = \rho$ for all $1 \leq k \neq l \leq 3$, i.e., the three subspaces are equally separated from each other, and that $\phi_1 = \phi_2 = \phi_3 = \phi$, so with equal sample density. Associated with 9 different pairs $(\rho, \phi)$ of subspace affinity and sample density, Fig. 3 and 4 plot the simulated TNR and CCR, respectively, versus noise standard deviation $\sigma$; sub-figures on the same row (column, respectively) correspond to an identical $\phi$ ($\rho$, respectively), while $\rho$ ($\phi$, respectively) is increased when going from left to right (top to bottom, respectively). In generating Fig. 3 and 4, the total number of iterations is preset to be $M = 6$ for OMP and $M = \lceil 6/p \rceil$ for GOMP, respectively; this represents the ideal situation that subspace dimension (equal to 6) is known. It is first seen from Fig. 3 that, as compared to the conventional OMP $(p = 1)$, the

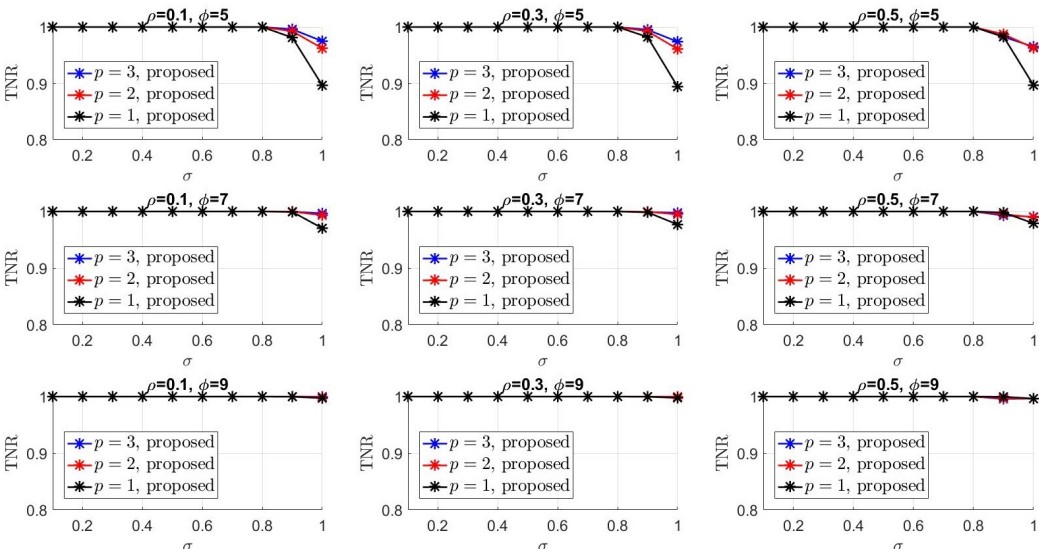

Figure 5: TNR versus noise standard deviation $\sigma$ for nine different pairs of subspace affinity $\rho$ and sample density $\phi$; the proposed stopping rule (6) is used.

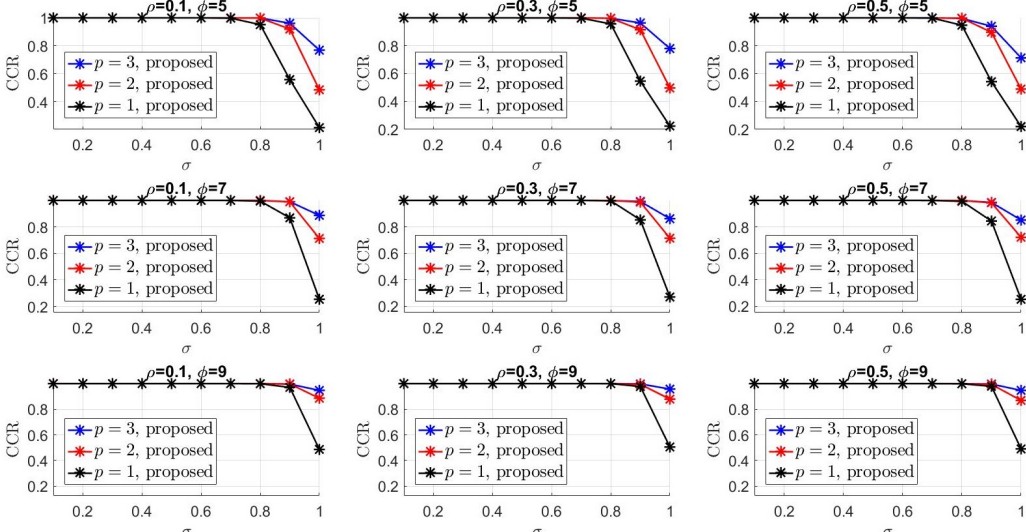

Figure 6: CCR versus noise standard deviation $\sigma$ for nine different pairs of subspace affinity $\rho$ and sample density $\phi$; the proposed stopping rule (6) is used.

proposed GOMP with multiple neighbor selection ($p > 1$) yields higher TNR. In particular, GOMP with $p = 3$ achieves the highest TNR because it involves the fewest iterations (only two) and, thus, is subject to the least perturbation of the residual vector. Then, it is seen from Fig. 4 that improved TNR in turn leads to higher CCR. The above experiment is conducted again by instead using the proposed stopping rule (6) to halt the algorithm, and the results are shown in Fig. 5 and Fig. 6. It can be seen that the proposed GOMP still outperforms OMP. Fig. 7 further plots the ANRN of the two methods, both employing the proposed stopping rule (6), for different noise standard deviation $\sigma$. We first observe from the figure that, when noise is so small that $\sigma < 0.1$, ANRN of all cases is about six, in support of the assertion in Theorem 2 that as many neighbors as the subspace dimension suffice to well explain the data point under consideration. When $\sigma$ grows up to 0.1, the proposed GOMP (with $p = 2, 3$) tends to retrieve more neighbors. As we have mentioned in the discussions at the end of Section 3, further noise corruption would enlarge the dimension of the span of the data cluster; as a result, the algorithm is terminated after more than $\lceil d_L/p \rceil$ iterations so as to recover more neighbors (than the ground truth subspace dimension) to explain the data point. When

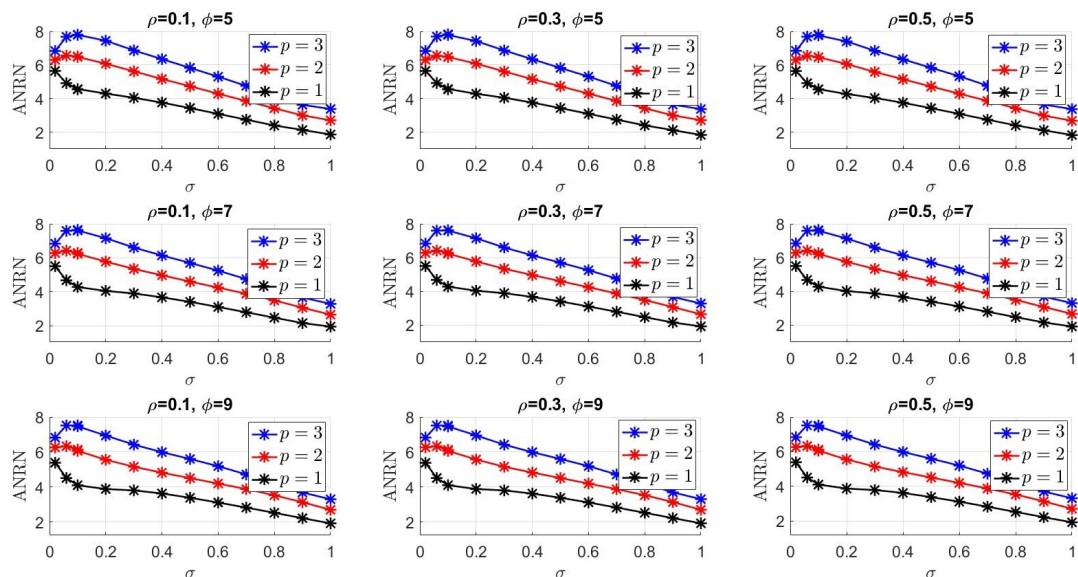

Figure 7: ANRN versus noise standard deviation $\sigma$ for nine different pairs of subspace affinity $\rho$ and sample density $\phi$; the proposed stopping rule (6) is used.

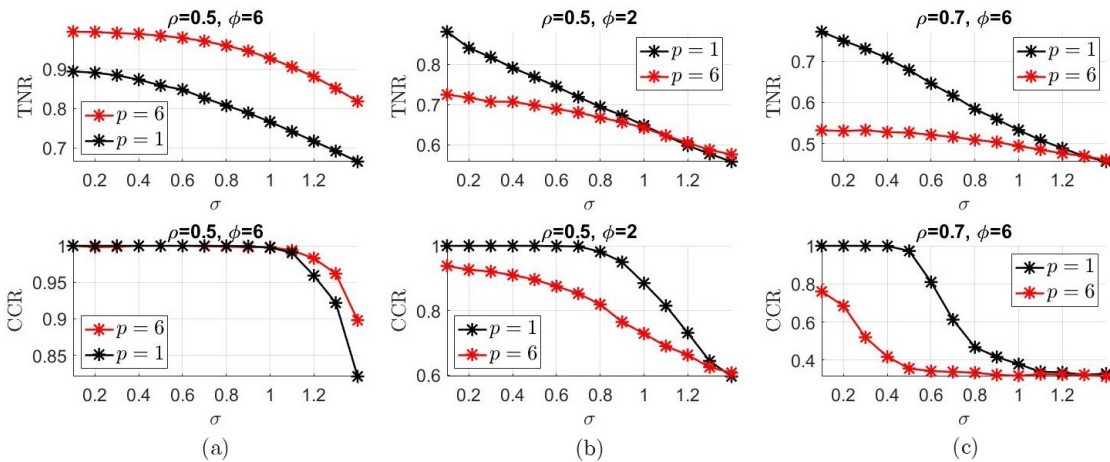

Figure 8: TNR and CCR versus noise standard deviation $\sigma$ for three different pairs of subspace affinity $\rho$ and sample density $\phi$; the subspace dimension ($d_k = 6$) is known. We set $p = 6$ for GOMP so that only one iteration is conducted, whereas OMP conducts $M = 6$ iterations.

$\sigma$ gets even larger, the residual is then severely dominated by noise, only to halt neighbor search earlier, say, in less than $\lceil d_L/p \rceil$ iterations; this ends up with fewer recovered neighbors and reduced ANRN. Finally, we consider the special case that subspace dimension $d_k$ is known, and set $p = d_k(= 6)$ so that GOMP executes just one iteration. Specifically, we consider the following three affinity-density pairs $(\rho, \phi)=(0.5,6)$, $(0.5,2)$, $(0.7,6)$, and the total number of iterations for OMP is set to be $M = 6$. Fig. 8 plots the TNR and CCR curves. Fig. 8-(a) clearly demonstrates that GOMP outperforms OMP when the subspace affinity $\rho$ is small and the sample density $\phi$ is large. The results are expected since, with only one single iteration and $p = d_k$, GOMP can perform well only when each data point has at least $p$ correct neighbors nearby, a situation fulfilled when subspaces are separated far away from each other (small $\rho$) and there are many neighbors around (large $\phi$). However, OMP performs better as the sample density $\phi$ decreases to 2 so that correct neighbors are scant (see Fig. 8-(b)), or the subspace affinity $\rho$ increases to 0.7, resulting in many incorrect neighbors (see Fig. 8-(c)). Hence, while GOMP conducts just one iteration and enjoys a smaller AoD, lack of enough good neighbors outweighs the benefit of small AoD.

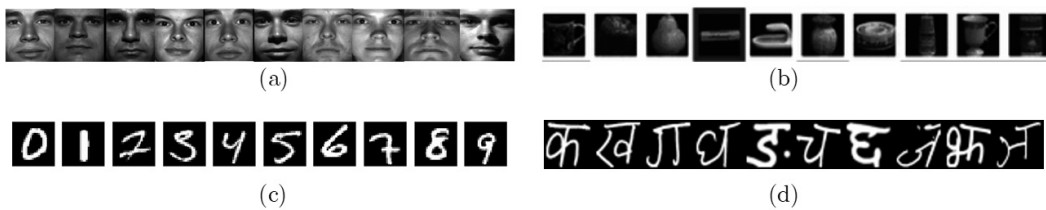

Figure 9: Samples of four real datasets. (a) The Extended Yale B human face dataset. (b) The Coil-100 object image dataset. (c) The MNIST handwritten digit dataset. (d) The Devanagari handwritten character dataset.

Table 2: Running time (in minutes) of OMP ($p = 1$) and GOMP ($p = 2, 3, 4, 6$) when 12 neighbors are recovered.

| Running time (min) | $p = 1$ | $p = 2$ | $p = 3$ | $p = 4$ | $p = 6$ |
|---|---|---|---|---|---|
| Extended Yale B | 1.51 | 0.85 | 0.61 | 0.48 | 0.3 |
| Coil-100 | 2.83 | 1.45 | 0.98 | 0.75 | 0.51 |
| MNIST | 194 | 100 | 68 | 50 | 37 |
| Devanagari | 218 | 130 | 91 | 72 | 47 |

## 4.2 Real-World Data

We proceed to evaluate the performance of the proposed method by using the following four real-world datasets (an illustration of their test samples is shown in Fig. 9).

- The Extended Yale B human face dataset (Lee et al., 2005), comprised of photos of 38 people each with 65 images of 192×168 pixels. To reduce the computational cost, we use the dimensionality reduction technique as in (Elhamifar & Vidal, 2013; Peng et al., 2013; Matsushima & Brbic, 2019) for reducing the image size to 48×42.

- The Coil-100 dataset (Nene et al., 1996), composed of 7200 images (each with size 32×32) of 100 objects (72 images for each object).

- The MNIST dataset (LeCun et al., 1998), containing 70000 images of handwritten digits with size 28×28; at least 6000 images for each digit.

- The Devanagari dataset (Acharya et al., 2015), containing images of handwritten Devanagari characters and digits. We use the character image part, which consists of 72000 images of 36 characters with size 32×32 (2000 images per character).

In the first part of the simulations, we compare the proposed GOMP with OMP using the above four real datasets. Assuming that a total number of 12 neighbors are to be recovered, Table 2 lists the running time for OMP ($p = 1$) and GOMP ($p = 2, 3, 4, 6$); the experiments are conducted using MATLAB 2016b on a desktop with an AMD 3700X CPU of 3.6GHz and 64 GB RAM. It can be seen that GOMP enjoys a shorter running time, confirming the computational efficiency of GOMP. Table 3 then compares the resultant TNR, showing that GOMP yields higher TNR when the same number of neighbors are to be found. We note that, unlike the synthetic data case, the TNR of the two greedy methods is less than 0.5 for Yale-B, Coil-100, and Devanagari datasets, in disagreement with the high-TNR assertion of our recovery rate analyses in Section 3. This is because real-world datasets seldom meet the assumptions underlying the semi-random model (Assumptions 1 and 2): neither the signal points are uniform nor the noise is Gaussian (e.g., real human face data are typically corrupted by sparse outliers (Elhamifar & Vidal, 2013; Soltanolkotabi & Candès, 2012)). Hence, the developed recovery rate formulae in Section 3 may not fully explain the simulated results when

Table 3: TNR of OMP ($p = 1$) and GOMP ($p = 2, 3, 4, 6$) when 12 neighbors are recovered.

| TNR | $p = 1$ | $p = 2$ | $p = 3$ | $p = 4$ | $p = 6$ |
|---|---|---|---|---|---|
| Extended Yale B | 0.37 | 0.41 | 0.46 | 0.49 | 0.58 |
| Coil-100 | 0.20 | 0.21 | 0.26 | 0.33 | 0.45 |
| MNIST | 0.58 | 0.61 | 0.64 | 0.66 | 0.68 |
| Devanagari | 0.2 | 0.23 | 0.30 | 0.39 | 0.51 |

Table 4: FDR of OMP ($p = 1$) and GOMP ($p = 2, 3, 4, 6$) when 12 neighbors are recovered.

| FDR | $p = 1$ | $p = 2$ | $p = 3$ | $p = 4$ | $p = 6$ |
|---|---|---|---|---|---|
| Extended Yale B | 0.85 | 0.86 | 0.87 | 0.87 | 0.90 |
| Coil-100 | 0.94 | 0.93 | 0.93 | 0.94 | 0.95 |
| MNIST | 0.80 | 0.81 | 0.81 | 0.83 | 0.84 |
| Devanagari | 0.51 | 0.56 | 0.60 | 0.65 | 0.72 |

Table 5: CCR of OMP ($p = 1$) and GOMP ($p = 2, 3, 4, 6$) when 12 neighbors are recovered.

| CCR | $p = 1$ | $p = 2$ | $p = 3$ | $p = 4$ | $p = 6$ |
|---|---|---|---|---|---|
| Extended Yale B | 0.47 | 0.49 | 0.56 | 0.63 | 0.67 |
| Coil-100 | 0.29 | 0.32 | 0.37 | 0.42 | 0.48 |
| MNIST | 0.43 | 0.51 | 0.63 | 0.64 | 0.66 |
| Devanagari | 0.26 | 0.31 | 0.35 | 0.37 | 0.38 |

real-world datasets are considered. Despite TNR may not be high in certain cases, we would like to remark that the existence of false neighbors does not largely degrade final data clustering accuracy. To see this, we further consider the feature detection rate (FDR) (Heckel & Bölcskei, 2015), defined to be

$$FDR \triangleq \frac{1}{N} \sum_{i=1}^{N} \frac{\|\widetilde{\mathbf{c}}_i^*\|_2}{\|\mathbf{c}_i^*\|_2}, \tag{26}$$

where $\widetilde{\mathbf{c}}_i^*$ is the vector containing the entries of $\mathbf{c}_i^*$ supported on true neighbors. If FDR is high, the weights on the false edges in the similarity graph are small, meaning that the existence of false neighbors is somehow not harmful. Table 4 presents the computed FDR; it can be observed that both OMP and GOMP achieve high FDR (above 0.7 for most cases). Table 5 then lists the CCR, showing that GOMP achieves higher CCR thanks to higher TNR.

Finally, we illustrate the ANRN of GOMP when employing the proposed stopping rule (6), and discuss the selection of the parameter $p$ aimed at fulfilling the dimension-aware property. For this purpose, knowledge about ground truth subspace dimensions and noise level is needed. To determine the subspace dimensions, we vectorize all data points associated with each cluster, stack them into a matrix, and calculate the normalized singular values (with respect to the maximal one). For each dataset, we choose the first 9 clusters for singular value computation and the results are plotted in Fig. 10, respectively. Observe from the four figures that the normalized singular values of all datasets exhibit a sudden slump from 1 to 0.1, beyond which they decrease slowly; the curves are seen to bend to the floors around 7∼9 for Yale-B, 5∼7 for Coil-100, 10∼15 for MNIST, and 15∼20 for Devanagari, respectively. The above singular value index ranges are then used as the

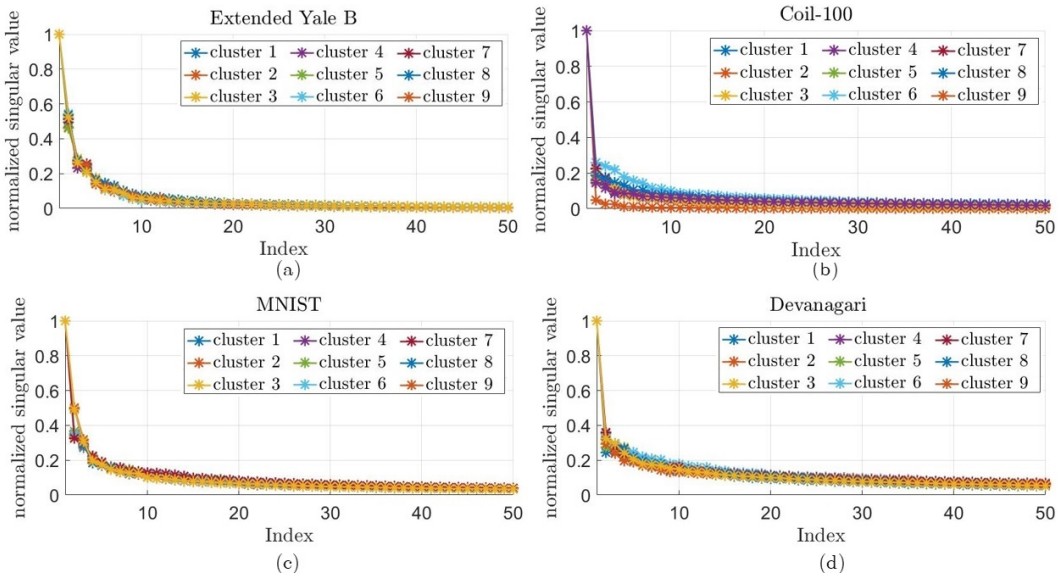

Figure 10: Normalized singular values of data matrices of the first 9 clusters of (a) Extended Yale B dataset, (b) Coil-100 dataset, (c) MNIST dataset, and (d) Devanagari dataset.

Table 6: Estimated subspace dimensions, noise powers, subspace affinity, data size, and ANRN of GOMP with proposed stopping rule.

|  | Coil-100 | MNIST | Devanagari | Extended Yale B |
|---|---|---|---|---|
| Range of subspace dimension | [6,10] | [10,15] | [15,20] | [7,9] |
| Estimated noise powers | 0.036 | 0.194 | 0.199 | 0.032 |
| Estimated subspace affinity | 0.447 | 0.553 | 0.628 | 0.706 |
| Data size | 7200 | 70000 | 72000 | 2470 |
| ANRN ($p = 2$) | 4.7 | 8.6 | 6.2 | 7.5 |
| ANRN ($p = 3$) | 6.3 | 11.6 | 7.4 | 11.2 |
| ANRN ($p = 4$) | 8.4 | 13.5 | 9.4 | 14.3 |
| ANRN ($p = 6$) | 12.1 | 17.7 | 16.8 | 19.6 |

estimated subspace dimensions. Also, the noise power of each dataset is estimated as the averaged sum of the rest nondominant singular values. Table 6 then shows the ranges of estimated subspace dimensions, average subspace affinity, estimated noise powers, and the computed ANRN of GOMP using the proposed stopping rule (6) with $p = 2, 3, 4, 6$. We can first observe from the table that, for the Coil-100 and MNIST datasets (both with relatively small affinity, though the latter subject to large noise), their ANRNs' are within the ranges 4.7∼12.1 and 8.6∼17.7, respectively, which are fairly close to the estimated subspace dimensions $[6, 10]$ and $[10, 15]$. This implies that, for these two datasets, the proposed stopping rule (6) is aware of the subspace dimension. However, for the Devanagari dataset (with a medium-to-large affinity) and Extended Yale B dataset (with a large affinity, despite small noise corruption), their ANRNs' deviate largely from the subspace dimensions, and hence the dimension-aware property seems to fail. We particularly observe that, for the Devanagari dataset, the algorithm is halted earlier (in less than $\lceil d/p \rceil$ iterations) with a small $p$ ($p = 2, 3, 4$), resulting in fewer recovered neighbors and reduced ANRN. This is mainly because, with large noise corruption, the residual is quickly dominated by noise so as to trigger the stopping rule (6) early (hence, less than $\lceil d/p \rceil$ iterations); in this case, a large $p$ (say $p = 6$) to soon recover many neighbors is

desired. Such a tendency is also seen in the synthetic data case (see Fig. 7). Regarding the Extended Yale B dataset, GOMP with a small $p$ ($p = 2, 3$) is instead preferred. This is because, with a large affinity, different clusters are close to each other; if $p$ is large, the algorithm is likely to pick neighbors from incorrect clusters and is then misled to find neighbors from some subspace (other than the ground truth one) with a larger dimension, thereby ending up with higher ANRN. Based on the above simulation findings we conclude that, as long as the subspace affinity is small, the proposed stopping rule (6) is aware of the subspace dimension, allowing for a wide range of $p$. If the affinity is medium-to-large, a good $p$ is highly dependent on the noise power: large (small, respectively) $p$ for large (small, respectively) noise power. Overall, the selection of $p$ is therefore closely related to the subspace affinity and noise power, which are nonetheless unknown unless additional off-line estimation is further conducted. Notably, we can also observe from Table 6 that the good $p$ increases with the data size $N$; this has also been inferred based on our recovery rate analysis for the semi-random model (see the discussions in Section 3.2). Hence a very simple guideline, without the need of knowing the subspace dimensions and noise strength, is to choose small $p$ (say, $p = 2, 3$) for small-size datasets (with thousands of data points or less) and large $p$ ($p = 5, 6$) for large-scale datasets (with tens of thousands of data points or more).

In the second part of this simulation, we compare GOMP using the proposed stopping rule (6) (dubbed by "proposed") with the following neighbor identification schemes:

- GOMP terminated after $\lceil d/p \rceil$ iterations, in which $d$ is the ground-truth subspace dimension determined as the floor of the average of the subspace dimensions within the range in Table 6. This serves as the ideal implementation of GOMP that employs the knowledge of the subspace dimension (hereafter coined as "ideal").

- OMP (Tschannen & Bölcskei, 2018) terminated after a pre-determined number $M$ of iterations.

- Active OMP (AOMP) (Chen et al., 2018) terminated after a pre-determined number $M$ of iterations; the neighbor dropping probability is set to be 0.8, as used in (Chen et al., 2018).

- Restricted connection OMP (ROMP) (Zhu et al., 2019); following the same setting as in (Zhu et al., 2019), the number of iterations is set to be 3, and the number of restricted connections is set to be 2.

- SSC with conventional $\ell_1$-minimization (LASSO) technique (Elhamifar & Vidal, 2013); the regularization parameter $\lambda$ is obtained via an exhaustive search over the interval $(0, 0.5]$, as in (Matsushima & Brbic, 2019).

- Exemplar-based subspace clustering (You et al., 2018); the number of exemplar data points is set as 160, as used in (You et al., 2018), and the regularization parameter $\lambda$ is obtained via an exhaustive search over the interval $(0, 0.5]$, as in (Matsushima & Brbic, 2019).

- Scalable sparse subspace clustering (S3C) (Peng et al., 2013); following the same setting as in (Peng et al., 2013), we randomly choose 1000 data points as "in-sample data" to find coarse estimates of the ground-truth subspaces; the regularization parameter $\lambda$ is obtained via an exhaustive search over $(0, 0.5]$, as in (Matsushima & Brbic, 2019).

- Selective sampling-based scalable sparse subspace clustering (S5C) (Matsushima & Brbic, 2019); to carry out representation learning to find representative data points, the batch size is set to be 1, as used in (Matsushima & Brbic, 2019), and the size of representation set is chosen to be $10L$. Again, the regularization parameter $\lambda$ is obtained via an exhaustive search over $(0, 0.5]$ as in (Matsushima & Brbic, 2019).

- Elastic net subspace clustering with oracle guided elastic net (ORN) (You et al., 2016a); the noise trade-off parameter $\gamma$ is set to be 1, and the regularization parameter $\lambda$ is obtained via an exhaustive search over $(0, 0.5]$, as in (Matsushima & Brbic, 2019).

Note that the last four methods, namely, ESC, S3C, S5C, and ORN, are reduced-complexity variants of the $\ell_1$-minimization scheme (Elhamifar & Vidal, 2013). For both OMP and AOMP, we conduct an exhaustive

Table 7: Simulation parameters for GOMP, OMP, and AOMP.

| Parameters | Extended Yale B | Coil-100 | MNIST | Devanagari |
|---|---|---|---|---|
| GOMP-ideal ($p$) | 2 | 3 | 4 | 6 |
| GOMP-proposed ($p$) | 2 | 3 | 4 | 6 |
| OMP ($M$) | 8 | 4 | 4 | 5 |
| AOMP ($M$) | 12 | 4 | 6 | 6 |

Table 8: CCR of different clustering algorithms (the blank denotes that the time limit of 7 days is exceeded).

| CCR | OMP | AOMP | ROMP | SSC | ESC | S3C | S5C | ORN | proposed | ideal |
|---|---|---|---|---|---|---|---|---|---|---|
| Yale B | 0.73 | 0.81 | 0.77 | 0.81 | 0.44 | 0.33 | 0.71 | 0.83 | 0.84 | 0.84 |
| Coil 100 | 0.46 | 0.35 | 0.38 | 0.79 | 0.23 | 0.14 | 0.53 | 0.76 | 0.71 | 0.79 |
| MNIST | 0.61 | 0.64 | 0.63 | × | 0.27 | 0.22 | 0.55 | 0.72 | 0.64 | 0.66 |
| Deva. | 0.28 | 0.33 | 0.28 | × | 0.20 | 0.15 | 0.33 | 0.40 | 0.35 | 0.38 |

Table 9: Running time (in minutes) of different clustering algorithms (the blank denotes that the time limit of 7 days is exceeded).

| (minute) | OMP | AOMP | ROMP | SSC | ESC | S3C | S5C | ORN | proposed | ideal |
|---|---|---|---|---|---|---|---|---|---|---|
| Yale B | 0.96 | 1.55 | 0.87 | 577 | 4 | 140 | 11 | 13 | 0.55 | 0.43 |
| Coil 100 | 0.95 | 2.7 | 0.82 | 988 | 6 | 138 | 42 | 31 | 0.55 | 0.5 |
| MNIST | 77 | 56 | 54 | × | 83 | 109 | 46 | 69 | 52 | 37 |
| Deva. | 96 | 74 | 69 | × | 96 | 142 | 54 | 80 | 59 | 47 |

search over the integer set $\{1, 2, ..., 18\}$ to find the best number $M$ of iterations that achieve the highest CCR; such an approach has been adopted in the simulation study of (Matsushima & Brbic, 2019); for both GOMP-proposed and GOMP-ideal, the parameter $p$ for each dataset is selected according to the suggested guidelines (see Table 7 for a list of the parameters used for OMP and GOMP algorithms). Table 8 shows the CCR of all methods. Compared to the competing greedy algorithms OMP, AOMP, and ROMP, GOMP-proposed (proposed) achieves higher CCR. As against the five $\ell_1$-minimization based solutions, GOMP-proposed (proposed) is next to ORN (You et al., 2016a), which is essentially a LASSO regression modified to further promote connectivity. It is also observed that GOMP-proposed (proposed) performs just slightly inferior to GOMP-ideal (ideal), which requires knowledge of the subspace dimension. This indicates that the proposed stopping rule (6) in conjunction with the suggested $p$ is aware of the subspace dimension. Table 9 then lists the running time of all methods. The results show that GOMP-proposed (proposed) is very computationally efficient: it enjoys the lowest running time in most cases. For the MNIST and Devanagari datasets, S5C is faster but is inferior to GOMP-ideal (ideal).

## 5 Proofs

### 5.1 Proof of Theorem 1

We set about the proof by defining the following per-iteration neighbor recovery event:

$$E_m \triangleq \{\text{at least } k_m \text{ true neighbors are obtained in the } m\text{th iteration}\}, 1 \le m \le M. \tag{27}$$

It suffices to show the event $\bigcap_{m=1}^{M} E_m$ holds with a probability as high as claimed in (15). The following lemmas are needed for deriving Theorem 1.

**Lemma 1.** *((Davenport et al., 2011, Ex. 25); (Heckel & Bölcskei, 2015, eq. 59)) Let $\mathbf{a} \in \mathbb{R}^m$ be uniformly distributed over the unit-sphere of $\mathbb{R}^m$, and $\mathbf{b} \in \mathbb{R}^m$ be a random vector independent of $\mathbf{a}$. Then, for $\epsilon \geq 0$ we have*

$$Pr\{|\mathbf{a}^T\mathbf{b}| > \epsilon\|\mathbf{b}\|_2|\} \leq 2e^{-m\epsilon^2/2}, \tag{28}$$

$$Pr\{|\mathbf{a}^T\mathbf{b}| < \epsilon\|\mathbf{b}\|_2|\} \leq \sqrt{2/\pi}\epsilon. \tag{29}$$

$\square$

**Lemma 2.** *Let $\mathbf{U}_l \in \mathbb{R}^{n \times d_l}$ be a matrix whose columns form an orthonormal basis for subspace $\mathcal{S}_l$, $1 \leq l \leq L$. For $\mathbf{x}_i \notin \mathcal{S}_L$, we have*

$$Pr\left\{\left|\left\langle\mathbf{x}_i, \frac{\mathbf{r}_{m,\|}^{(N)}}{\|\mathbf{r}_{m,\|}^{(N)}\|_2}\right\rangle\right| > \max_{l \neq L} 4\log(N)\frac{\|\mathbf{U}_l^T\mathbf{U}_L\|_F}{\sqrt{d_l d_L}}\right\} < \frac{2}{N^{(8\log N)/d_L}} \tag{30}$$

*Proof* : See Appendix A. $\square$

**Lemma 3.** *For $\mathbf{x}_i \notin \mathcal{S}_L$, we have*

$$Pr\left\{\left|\left\langle\mathbf{x}_i, \frac{\mathbf{r}_{m,\perp}^{(N)}}{\|\mathbf{r}_{m,\perp}^{(N)}\|_2}\right\rangle\right| \leq \sqrt{\frac{6\log N}{n - d_L}}\right\} \geq 1 - 2cN^{-3} \tag{31}$$

*Proof* : See Appendix B. $\square$

**Lemma 4.** *(Tschannen & Bölcskei, 2018, Lemma 10) The event $\bigcap_{i=1}^{N}\{\|\mathbf{e}_i\|_2 \leq 3\sigma/2\}$ occurs with a probability at least $1 - Ne^{-n/8}$.* $\square$

If $k_m = 0$, it follows $\Pr\{E_m\} = 1$ because the number of true neighbors is never negative. We then consider the case that $k_m > 0$. Since the already chosen data vectors in all the previous $m-1$ iterations are orthogonal to the residual $\mathbf{r}_{m-1}^{(N)}$, i.e., $|\langle\mathbf{y}_j, \mathbf{r}_{m-1}^{(N)}\rangle| = 0$ for all $j \in \Lambda_{m-1}$, the "yet-to-be-selected" candidate neighbors are $\mathbf{y}_j$'s for $j \in \{1, 2, ..., N-1\}\backslash\Lambda_{m-1}$. Given that $\Lambda_{m-1} = \Lambda_{m-1}^t \cup \Lambda_{m-1}^f$ , where the disjoint subsets $\Lambda_{m-1}^t$ and $\Lambda_{m-1}^f$ consist of, respectively, the indexes of true and false neighbors already selected up to the first $m-1$ iterations, $|\mathcal{Y}_L| - 1 - |\Lambda_{m-1}^t|$ and $N - |\mathcal{Y}_L| - |\Lambda_{m-1}^f|$ data vectors remain in the same and different clusters as $\mathbf{y}_N$, respectively. Let the true neighbor candidates $\mathbf{y}_{t_j}$, where $\mathbf{x}_{t_j} \in \mathcal{S}_L$ and $t_j \notin \Lambda_{m-1}^t$, be sorted according to $|\langle\mathbf{y}_{t_1}, \mathbf{r}_{m-1}^{(N)}\rangle| \geq ... \geq |\langle\mathbf{y}_{t_{|\mathcal{Y}_L|-1-|\Lambda_{m-1}^t|}}, \mathbf{r}_{m-1}^{(N)}\rangle|$, and likewise the false neighbor candidates $\mathbf{y}_{f_k}$, where $\mathbf{x}_{f_k} \notin \mathcal{S}_L$ and $f_k \notin \Lambda_{m-1}^f$, in the way that $|\langle\mathbf{y}_{f_1}, \mathbf{r}_{m-1}^{(N)}\rangle| \geq ... \geq |\langle\mathbf{y}_{f_{N-|\mathcal{Y}_L|-|\Lambda_{m-1}^f|}}, \mathbf{r}_{m-1}^{(N)}\rangle|$. Also, let the noiseless signal vectors $\mathbf{x}_{\bar{t}_g} \in \mathcal{S}_L$, $\bar{t}_g \notin \Lambda_{m-1}^t$, be ordered so that $|\langle\mathbf{x}_{\bar{t}_1}, \mathbf{r}_{m-1}^{(N)}\rangle| \geq ... \geq |\langle\mathbf{x}_{\bar{t}_{|\mathcal{Y}_L|-1-|\Lambda_{m-1}^t|}}, \mathbf{r}_{m-1}^{(N)}\rangle|$, while those $\mathbf{x}_{\bar{f}_h} \notin \mathcal{S}_L$, $\bar{f}_h \notin \Lambda_{m-1}^f$, in the way $|\langle\mathbf{x}_{\bar{f}_1}, \mathbf{r}_{m-1}^{(N)}\rangle| \geq ... \geq |\langle\mathbf{x}_{\bar{f}_{N-|\mathcal{Y}_L|-|\Lambda_{m-1}^f|}}, \mathbf{r}_{m-1}^{(N)}\rangle|$.

The event $E_m$ defined in (27) can be expressed in terms of the above ordered statistics. By definition, $E_m$ occurs when at least $k_m$ true neighbors, and so at most $p - k_m$ false neighbors, are recovered in the $m$th iteration. That is to say, $\mathbf{y}_{t_{k_m}}$ is chosen as a neighbor but $\mathbf{y}_{f_{p-k_m+1}}$ is not, justifying the inequality $|\langle\mathbf{y}_{f_{p-k_m+1}}, \mathbf{r}_{m-1}^{(N)}\rangle| \leq |\langle\mathbf{y}_{t_{k_m}}, \mathbf{r}_{m-1}^{(N)}\rangle|$. The converse is obviously true. Hence we can rewrite $E_m$ as

$$E_m = \{|\langle\mathbf{y}_{f_{p-k_m+1}}, \mathbf{r}_{m-1}^{(N)}\rangle| \leq |\langle\mathbf{y}_{t_{k_m}}, \mathbf{r}_{m-1}^{(N)}\rangle|\}. \tag{32}$$

The expression (32) involves ordered statistics of the absolute inner products between the residual and noisy data points $\mathbf{y}_j$'s, whose distribution is however quite complicated. To ease our lower bound derivation, we shall instead seek sufficient conditions, specified by absolute inner products between the residual and

noise-free signal points $\mathbf{x}_j$'s, for $|\langle \mathbf{y}_{f_{p-k_m+1}}, \mathbf{r}_{m-1}^{(N)} \rangle| \leq |\langle \mathbf{y}_{t_{k_m}}, \mathbf{r}_{m-1}^{(N)} \rangle|$; this allows us to employ the assumed uniform distribution of $\mathbf{x}_j$'s (Assumption 1) as well as the ground-truth subspace orientation (Assumption 3) to facilitate analysis. For this we first note from (12) that, for all $1 \leq j \leq N-1$, we have

$$\langle \mathbf{y}_j, \mathbf{r}_{m-1}^{(N)} \rangle = \langle \mathbf{x}_j, \mathbf{r}_{m-1,\|}^{(N)} \rangle + \langle \mathbf{x}_j, \mathbf{r}_{m-1,\perp}^{(N)} \rangle + \langle \mathbf{e}_j, \mathbf{r}_{m-1}^{(N)} \rangle. \tag{33}$$

Then for all $u \leq k_m$, it follows

$$\begin{aligned}
|\langle \mathbf{y}_{\bar{t}_u}, \mathbf{r}_{m-1}^{(N)} \rangle| &= |\langle \mathbf{x}_{\bar{t}_u}, \mathbf{r}_{m-1,\|}^{(N)} \rangle + \langle \mathbf{x}_{\bar{t}_u}, \mathbf{r}_{m-1,\perp}^{(N)} \rangle + \langle \mathbf{e}_{\bar{t}_u}, \mathbf{r}_{m-1}^{(N)} \rangle| \\
&\stackrel{(a)}{=} |\langle \mathbf{x}_{\bar{t}_u}, \mathbf{r}_{m-1,\|}^{(N)} \rangle + \langle \mathbf{e}_{\bar{t}_u}, \mathbf{r}_{m-1}^{(N)} \rangle| \\
&\geq |\langle \mathbf{x}_{\bar{t}_u}, \mathbf{r}_{m-1,\|}^{(N)} \rangle| - \max_{1 \leq j \leq |\mathcal{Y}_L| - 1 - |\Lambda_{m-1}^t|} |\langle \mathbf{e}_{t_j}, \mathbf{r}_{m-1}^{(N)} \rangle| \\
&\stackrel{(b)}{\geq} |\langle \mathbf{x}_{\bar{t}_{k_m}}, \mathbf{r}_{m-1,\|}^{(N)} \rangle| - \max_{1 \leq j \leq |\mathcal{Y}_L| - 1 - |\Lambda_{m-1}^t|} |\langle \mathbf{e}_{t_j}, \mathbf{r}_{m-1}^{(N)} \rangle|,
\end{aligned} \tag{34}$$

where (a) holds since $\mathbf{x}_{t_j} \in \mathcal{S}_L$ for $\mathbf{x}_{t_j} \in \mathcal{S}_L$, and (b) follows from the ordering of $\mathbf{x}_{\bar{t}_g}$'s. Hence, at least $k_m$ absolute inner products $|\langle \mathbf{y}_{\bar{t}_u}, \mathbf{r}_{m-1}^{(N)} \rangle|$ are no smaller than the term on the right-hand-side (RHS) of (34); in particular, for $|\langle \mathbf{y}_{t_{k_m}}, \mathbf{r}_{m-1}^{(N)} \rangle|$, the $k_m$th largest $|\langle \mathbf{y}_{t_j}, \mathbf{r}_{m-1}^{(N)} \rangle|$, we must have

$$|\langle \mathbf{y}_{t_{k_m}}, \mathbf{r}_{m-1}^{(N)} \rangle| \geq |\langle \mathbf{x}_{\bar{t}_{k_m}}, \mathbf{r}_{m-1,\|}^{(N)} \rangle| - \max_{1 \leq j \leq |\mathcal{Y}_L| - 1 - |\Lambda_{m-1}^t|} |\langle \mathbf{e}_{t_j}, \mathbf{r}_{m-1}^{(N)} \rangle|. \tag{35}$$

Similarly, by the ordering of $|\langle \mathbf{x}_{\bar{f}_k}, \mathbf{r}_{m-1,\|}^{(N)} \rangle|$, for all $u \geq p - k_m + 1$, we have

$$\begin{aligned}
|\langle \mathbf{y}_{f_{p-k_m+1}}, \mathbf{r}_{m-1}^{(N)} \rangle| \leq{} &|\langle \mathbf{x}_{\bar{f}_{p-k_m+1}}, \mathbf{r}_{m-1,\|}^{(N)} \rangle| + \max_{1 \leq k \leq N - |\mathcal{Y}_L| - |\Lambda_{m-1}^f|} |\langle \mathbf{x}_{f_k}, \mathbf{r}_{m-1,\perp}^{(N)} \rangle| \\
&+ \max_{1 \leq k \leq N - |\mathcal{Y}_L| - |\Lambda_{m-1}^f|} |\langle \mathbf{e}_{f_k}, \mathbf{r}_{m-1}^{(N)} \rangle|.
\end{aligned} \tag{36}$$

Putting (35) and (36) together, the condition $|\langle \mathbf{y}_{f_{p-k_m+1}}, \mathbf{r}_{m-1}^{(N)} \rangle| \leq |\langle \mathbf{y}_{t_{k_m}}, \mathbf{r}_{m-1}^{(N)} \rangle|$ is guaranteed, and hence the event $E_m$ occurs, once the following inequality is true

$$\begin{aligned}
&|\langle \mathbf{x}_{\bar{f}_{p-k_m+1}}, \mathbf{r}_{m-1,\|}^{(N)} \rangle| + \max_{1 \leq k \leq N - |\mathcal{Y}_L| - |\Lambda_{m-1}^f|} |\langle \mathbf{x}_{f_k}, \mathbf{r}_{m-1,\perp}^{(N)} \rangle| \\
&+ \max_{1 \leq k \leq N - |\mathcal{Y}_L| - |\Lambda_{m-1}^f|} |\langle \mathbf{e}_{f_k}, \mathbf{r}_{m-1}^{(N)} \rangle| + \max_{1 \leq j \leq |\mathcal{Y}_L| - 1 - |\Lambda_{m-1}^t|} |\langle \mathbf{e}_{t_j}, \mathbf{r}_{m-1}^{(N)} \rangle| \leq |\langle \mathbf{x}_{\bar{t}_{k_m}}, \mathbf{r}_{m-1,\|}^{(N)} \rangle|.
\end{aligned} \tag{37}$$

Recall that our goal is to show that $\bigcap_{m=1}^M E_m$ occurs with a high probability. Under the semi-random model, we go on to estimate the probability of which the inequality (37) holds for all $1 \leq m \leq M$, and in turn a lower bound for $\Pr\{\bigcap_{m=1}^M E_m\}$. The basic idea is to "split and then lump": find an upper bound for each individual left-hand-side (LHS) term of (37) and estimate, one by one, the probability about which the obtained inequality holds, and similarly a lower bound for the RHS term along with an estimated probability of its validity; putting the results altogether gives a sufficient condition for (37) along with the desired probability lower bound.

a) An Upper Bound for the 1$^{\text{st}}$ LHS Term of (37): For a fixed $\alpha$, we first note that

$$
\begin{aligned}
&\Pr\{|\langle \mathbf{x}_{\overline{f}_{p-k_m+1}}, \mathbf{r}^{(N)}_{m-1,\|}\rangle| \le \alpha\} \\
&= 1 - \Pr\{|\langle \mathbf{x}_{\overline{f}_{p-k_m+1}}, \mathbf{r}^{(N)}_{m-1,\|}\rangle| > \alpha\} \\
&\overset{(a)}{\ge} 1 - \Pr\{\exists I \subset [N - |\mathcal{Y}_L| - |\Lambda^f_{m-1}|] \text{ with} \\
&\qquad\qquad |I| = p - k_m + 1 \text{ s.t. } |\langle \mathbf{x}_{f_k}, \mathbf{r}^{(N)}_{m-1,\|}\rangle| > \alpha, \forall k \in I\} \\
&\overset{(b)}{\ge} 1 - \binom{N - |\mathcal{Y}_L|}{p - k_m + 1}\Big(\max_{\substack{I:|I|=p-k_m+1 \\ I \subset [N-|\mathcal{Y}_L|-|\Lambda^f_{m-1}|]}} \Pr\{|\langle \mathbf{x}_{f_k}, \mathbf{r}^{(N)}_{m-1,\|}\rangle| > \alpha, \forall k \in I\}\Big) \\
&\overset{(c)}{\ge} 1 - \Big(\frac{e(N-|\mathcal{Y}_L|)}{p-k_m+1}\Big)^{p-k_m+1}\Big(\max_{\substack{I:|I|=p-k_m+1 \\ I \subset [N-|\mathcal{Y}_L|-|\Lambda^f_{m-1}|]}} \Pr\{|\langle \mathbf{x}_{f_k}, \mathbf{r}^{(N)}_{m-1,\|}\rangle| > \alpha, \forall k \in I\}\Big),
\end{aligned}
\tag{38}
$$

where in (a) the notation $[Q] \triangleq \{1, 2, ..., Q\}, Q \in \mathbb{N}$, in (b) $\binom{Q}{k} \triangleq \frac{Q!}{(Q-k)!k!}$, and (c) holds by (Cormen et al., 2009, inequality (C.5)). By Lemma 1 and since data points are independent, for $|I| = p - k_m + 1$ we have

$$
\Pr\Big\{|\langle \mathbf{x}_{f_k}, \mathbf{r}^{(N)}_{m,\|}\rangle| > \max_{l \ne L} 4\log(N)\frac{\|\mathbf{U}^T_l \mathbf{U}_L\|_F}{\sqrt{d_l d_L}}\|\mathbf{r}^{(N)}_{m,\|}\|_2, \forall k \in I\Big\} < \Big(\frac{2}{N^{(8\log N)/d_L}}\Big)^{p-k_m+1}.
\tag{39}
$$

Combining (38) and (39) and setting $\alpha = \max_{l \ne L} 4\log(N)\|\mathbf{U}^T_l \mathbf{U}_L\|_F\|\mathbf{r}^{(N)}_{m,\|}\|_2 / \sqrt{d_l d_L}$, it then follows

$$
|\langle \mathbf{x}_{\overline{f}_{p-k_m+1}}, \mathbf{r}^{(N)}_{m-1,\|}\rangle| \le \max_{l \ne L} 4\log(N)\frac{\|\mathbf{U}^T_l \mathbf{U}_L\|_F}{\sqrt{d_l d_L}}\|\mathbf{r}^{(N)}_{m,\|}\|_2
\tag{40}
$$

holds with a probability at least $1 - \big(\frac{2e(N-|\mathcal{Y}_L|)}{(p-k_m+1)N^{(8\log N)/d_L}}\big)^{p-k_m+1}$.

b) An Upper Bound for the 2$^{\text{nd}}$ LHS Term of (37): We first note that

$$
\begin{aligned}
&\Pr\Big\{\max_{1 \le k \le N-|\mathcal{Y}_L|-|\Lambda^f_{m-1}|} |\langle \mathbf{x}_{f_k}, \mathbf{r}^{(N)}_{m-1,\perp}\rangle| \le \alpha\Big\} \\
&= 1 - \Pr\Big\{\max_{1 \le k \le N-|\mathcal{Y}_L|-|\Lambda^f_{m-1}|} |\langle \mathbf{x}_{f_k}, \mathbf{r}^{(N)}_{m-1,\perp}\rangle| > \alpha\Big\} \\
&\ge 1 - \Pr\{\exists k \in [N - |\mathcal{Y}_L| - |\Lambda^f_{m-1}|] \text{ s.t. } |\langle \mathbf{x}_{f_k}, \mathbf{r}^{(N)}_{m-1,\perp}\rangle| > \alpha\} \\
&= 1 - \Pr\Big\{\Big(\bigcap_{k=1}^{N-|\mathcal{Y}_L|-|\Lambda^f_{m-1}|}\{|\langle \mathbf{x}_{f_k}, \mathbf{r}^{(N)}_{m-1,\perp}\rangle| \le \alpha\}\Big)^c\Big\} \\
&= 1 - \Pr\Big\{\bigcup_{k=1}^{N-|\mathcal{Y}_L|-|\Lambda^f_{m-1}|}\{|\langle \mathbf{x}_{f_k}, \mathbf{r}^{(N)}_{m-1,\perp}\rangle| > \alpha\}\Big\} \\
&\ge 1 - \sum_{k=1}^{N-|\mathcal{Y}_L|-|\Lambda^f_{m-1}|} \Pr\{|\langle \mathbf{x}_{f_k}, \mathbf{r}^{(N)}_{m-1,\perp}\rangle| > \alpha\}.
\end{aligned}
\tag{41}
$$

By Lemma 3, we have

$$
\Pr\Big\{|\langle \mathbf{x}_{f_k}, \mathbf{r}^{(N)}_{m,\perp}\rangle| > \sqrt{\frac{6\log N}{n-d_L}}\|\mathbf{r}^{(N)}_{m,\perp}\|_2\Big\} < 2cN^{-3}, \; 1 \le k \le N - |\mathcal{Y}_L| - |\Lambda^f_{m-1}|.
\tag{42}
$$

Combing (41) and (42) and with $\alpha = \sqrt{\frac{6 \log N}{n - d_L}} \|\mathbf{r}_{m,\perp}^{(N)}\|_2$, we obtain

$$\max_{1 \leq k \leq N - |\mathcal{Y}_L| - |\Lambda_{m-1}^f|} |\langle \mathbf{x}_{f_k}, \mathbf{r}_{m-1,\perp}^{(N)} \rangle| \leq \sqrt{\frac{6 \log N}{n - d_L}} \|\mathbf{r}_{m,\perp}^{(N)}\|_2 \tag{43}$$

holds with a probability at least $1 - 2cN^{-2}$.

c) An Upper Bound for the 3$^{\mathrm{rd}}$ LHS Term of (37): Using similar techniques as in deriving (41) we can first reach

$$\Pr\left\{ \max_{1 \leq k \leq N - |\mathcal{Y}_L| - |\Lambda_{m-1}^f|} |\langle \mathbf{e}_{f_k}, \mathbf{r}_{m-1}^{(N)} \rangle| \leq \alpha \right\} \geq 1 - \sum_{k=1}^{N - |\mathcal{Y}_L| - |\Lambda_{m-1}^f|} \Pr\{|\langle \mathbf{e}_{f_k}, \mathbf{r}_{m-1}^{(N)} \rangle| > \alpha\}. \tag{44}$$

Using (28) and with $\mathbf{a} = \mathbf{e}_{f_k}/\|\mathbf{e}_{f_k}\|_2$ and $\mathbf{b} = \mathbf{r}_{m-1}^{(N)}$ , it follows

$$\Pr\left\{ |\langle \mathbf{e}_{f_k}, \mathbf{r}_m^{(N)} \rangle| > \sqrt{\frac{6 \log N}{n}} \|\mathbf{r}_m^{(N)}\|_2 \|\mathbf{e}_{f_k}\|_2 \right\}$$
$$= \Pr\left\{ \left| \left\langle \frac{\mathbf{e}_{f_k}}{\|\mathbf{e}_{f_k}\|_2}, \mathbf{r}_m^{(N)} \right\rangle \right| > \sqrt{\frac{6 \log N}{n}} \|\mathbf{r}_m^{(N)}\|_2 \right\} \leq 2N^{-3}. \tag{45}$$

Under Assumption 1 we have

$$\|\mathbf{r}_m^{(N)}\|_2 = \|\mathrm{P}_{\mathcal{R}(\mathbf{Y}_{\Lambda_{m-1}})} \mathbf{y}_N\|_2 \leq \|\mathbf{y}_N\|_2 = \|\mathbf{x}_N + \mathbf{e}_N\|_2 \leq 1 + \|\mathbf{e}_N\|_2, \tag{46}$$

in which $\mathcal{R}(\mathbf{Y}_{\Lambda_{m-1}})$ is the column space of $\mathbf{Y}_{\Lambda_{m-1}}$ and the last inequality holds by the triangle inequality. Combing (45) and (46) yields

$$\Pr\left\{ |\langle \mathbf{e}_{f_k}, \mathbf{r}_m^{(N)} \rangle| > \sqrt{\frac{6 \log N}{n}} (1 + \|\mathbf{e}_N\|_2) \|\mathbf{e}_{f_k}\|_2 \right\} \leq 2N^{-3}. \tag{47}$$

Setting $\alpha = \sqrt{\frac{6 \log N}{n}} (1 + \|\mathbf{e}_N\|_2) \max\limits_{1 \leq k \leq N - |\mathcal{Y}_L| - |\Lambda_{m-1}^f|} \|\mathbf{e}_{f_k}\|_2$, (44) and (47) imply

$$\max_{1 \leq k \leq N - |\mathcal{Y}_L| - |\Lambda_{m-1}^f|} |\langle \mathbf{e}_{f_k}, \mathbf{r}_{m-1}^{(N)} \rangle| \leq \sqrt{\frac{6 \log N}{n}} (1 + \|\mathbf{e}_N\|_2) \max_{1 \leq k \leq N - |\mathcal{Y}_L| - |\Lambda_{m-1}^f|} \|\mathbf{e}_{f_k}\|_2 \tag{48}$$

holds with a probability at least $1 - 2N^{-2}$.

d) An Upper Bound for the 4$^{\mathrm{th}}$ LHS Term of (37): By following the same procedures as from (44) to (48), we can readily show the inequality

$$\max_{1 \leq j \leq |\mathcal{Y}_L| - 1 - |\Lambda_{m-1}^t|} |\langle \mathbf{e}_{t_j}, \mathbf{r}_{m-1}^{(N)} \rangle| \leq \sqrt{\frac{6 \log N}{n}} (1 + \|\mathbf{e}_N\|_2) \max_{1 \leq j \leq |\mathcal{Y}_L| - 1 - |\Lambda_{m-1}^t|} \|\mathbf{e}_{t_j}\|_2. \tag{49}$$

e) A Lower Bound for the RHS Term of (37): Next, we go on to find a lower bound for the RHS term of (37). Using (29) and with $\mathbf{a} = \mathbf{x}_{t_j}$ and $\mathbf{b} = \mathbf{r}_{m-1,\|}^{(N)}$, we have

$$\Pr\left\{ |\langle \mathbf{x}_{t_j}, \mathbf{r}_{m,\|}^{(N)} \rangle| < \frac{\tau}{\sqrt{d_L}} \|\mathbf{r}_{m,\|}^{(N)}\|_2 \right\} \leq \sqrt{\frac{2}{\pi}} \tau, \ 1 \leq j \leq |\mathcal{Y}_L| - 1 - |\Lambda_{m-1}^t|, \tag{50}$$

which together with the assumption that $\mathbf{x}_{t_j}$'s are independent yields

$$\Pr\left\{ |\langle \mathbf{x}_{t_j}, \mathbf{r}_{m,\|}^{(N)} \rangle| < \frac{\tau}{\sqrt{d_L}} \|\mathbf{r}_{m,\|}^{(N)}\|_2, \ \forall j \in I \right\} \leq \left( \sqrt{\frac{2}{\pi}} \tau \right)^{|\mathcal{Y}_L| - |\Lambda_{m-1}^t| - k_m},$$
$$I \subset \left[ |\mathcal{Y}_L| - 1 - |\Lambda_{m-1}^t| \right], \ |I| = |\mathcal{Y}_L| - |\Lambda_{m-1}^t| - k_m. \tag{51}$$

It then follows

$$
\begin{aligned}
\Pr&\Big\{\langle \mathbf{x}_{t_{k_m}}, \mathbf{r}^{(N)}_{m-1,\|}\rangle| \geq \frac{\tau}{\sqrt{d_L}}\|\mathbf{r}^{(N)}_{m,\|}\|_2\Big\} \\
&= 1 - \Pr\Big\{\langle \mathbf{x}_{t_{k_m}}, \mathbf{r}^{(N)}_{m-1,\|}\rangle| < \frac{\tau}{\sqrt{d_L}}\|\mathbf{r}^{(N)}_{m,\|}\|_2\Big\} \\
&\geq 1 - \Pr\Big\{\exists I \subset \big[|\mathcal{Y}_L| - 1 - |\Lambda^t_{m-1}|\big] \text{ with } |I| = |\mathcal{Y}_L| - 1 - |\Lambda^t_{m-1}| - k_m + 1 \\
&\qquad\qquad\qquad\qquad \text{s.t. } |\langle \mathbf{x}_{t_j}, \mathbf{r}^{(N)}_{m,\|}\rangle| < \frac{\tau}{\sqrt{d_L}}\|\mathbf{r}^{(N)}_{m,\|}\|_2, \ \forall j \in I\Big\} \\
&\geq 1 - \binom{|\mathcal{Y}_L| - 1}{k_m - 1} \max_{\substack{|I|=|\mathcal{Y}_L|-|\Lambda^t_{m-1}|-k_m \\ I\subset[|\mathcal{Y}_L|-1-|\Lambda^t_{m-1}|]}} \Pr\Big\{|\langle \mathbf{x}_{t_j}, \mathbf{r}^{(N)}_{m,\|}\rangle| < \frac{\tau}{\sqrt{d_L}}\|\mathbf{r}^{(N)}_{m,\|}\|_2, \ \forall j \in I\Big\} \\
&\overset{(a)}{\geq} 1 - \binom{|\mathcal{Y}_L| - 1}{k_m - 1} \max_{\substack{|I|=|\mathcal{Y}_L|-|\Lambda^t_{m-1}|-k_m \\ I\subset[|\mathcal{Y}_L|-1-|\Lambda^t_{m-1}|]}} \Big(\sqrt{\frac{2}{\pi}}\tau\Big)^{|\mathcal{Y}_L|-|\Lambda^t_{m-1}|-k_m} \\
&\geq 1 - \Big(\frac{e(|\mathcal{Y}_L| - 1)}{k_m - 1}\Big)^{k_m - 1} \Big(\sqrt{\frac{2}{\pi}}\tau\Big)^{|\mathcal{Y}_L|-d_L-k_m},
\end{aligned}
\tag{52}
$$

where (a) holds thanks to (51). Hence, the inequality

$$
|\langle \mathbf{x}_{\bar{t}_{k_m}}, \mathbf{r}^{(N)}_{m-1,\|}\rangle| \geq \frac{\tau}{\sqrt{d_L}}\|\mathbf{r}^{(N)}_{m-1,\|}\|_2
\tag{53}
$$

holds with a probability at least $1 - \Big(\frac{e(|\mathcal{Y}_L|-1)}{k_m-1}\Big)^{k_m-1} \Big(\sqrt{\frac{2}{\pi}}\tau\Big)^{|\mathcal{Y}_L|-|\Lambda^t_{m-1}|-k_m}$.

With the bounds in (40), (43), (48), (49) and (53), the inequality in (37) is true, and hence the event $E_m$ occurs, when the following inequality holds:

$$
\begin{aligned}
\max_{l \neq L} 4\log(N)\frac{\|\mathbf{U}^T_l\mathbf{U}_L\|_F}{\sqrt{d_l d_L}}\|\mathbf{r}^{(N)}_{m,\|}\|_2 &+ \sqrt{\frac{6\log N}{n - d_L}}\|\mathbf{r}^{(N)}_{m,\perp}\|_2 \\
&+ 2\sqrt{\frac{6\log N}{n}}(1 + \|\mathbf{e}_N\|_2)\max_{1\leq i\leq N}\|\mathbf{e}_i\|_2 \leq \frac{\tau}{\sqrt{d_L}}\|\mathbf{r}^{(N)}_{m-1,\|}\|_2,
\end{aligned}
\tag{54}
$$

or equivalently,

$$
\max_{l \neq L}\frac{\|\mathbf{U}^T_l\mathbf{U}_L\|_F}{\sqrt{d_l}} + \frac{\sqrt{3d_L}\|\mathbf{r}^{(N)}_{m,\perp}\|_2}{\sqrt{8(n-d_L)\log N}\|\mathbf{r}^{(N)}_{m,\|}\|_2} + \frac{2\sqrt{3d_L}(1+\|\mathbf{e}_N\|_2)}{\sqrt{8n\log N}\|\mathbf{r}^{(N)}_{m-1,\|}\|_2}\max_{1\leq i\leq N}\|\mathbf{e}_i\|_2 \leq \frac{\tau}{4\log N},
\tag{55}
$$

which is obtained by multiplying the inequality (54) throughout by the factor $\sqrt{d_L} \times 1/(4\log(N)\|\mathbf{r}^{(N)}_{m-1,\|}\|_2)$. It can then be concluded that occurs once (40), (43), (48), (49), (53) and (55) hold for all $1 \leq m \leq M$. Hence, a lower bound for $\Pr\{\bigcap_{m=1}^M E_m\}$ can be obtained by finding a lower bound for the probability that (55) holds for all $1 \leq m \leq M$.

Still, our approach seeks an upper bound for the LHS term in (55), hence a sufficient condition guaranteeing (55), and proves this bound holds with a high probability. Towards this end, we derive an upper bound for $\|\mathbf{r}^{(N)}_{m-1,\perp}\|_2$, $1 \leq m \leq M$, an upper bound for $\|\mathbf{e}^{(N)}_i\|_2$, $1 \leq i \leq N$, and a lower bound for $\|\mathbf{r}^{(N)}_{m-1,\|}\|_2$, $1 \leq m \leq M$; lumping these altogether yields the claimed result. To begin with, we write

$$
\|\mathbf{r}^{(N)}_{m-1,\perp}\|_2 = \|\mathrm{P}_{\mathcal{S}^\perp_L}(\mathbf{I} - \mathbf{Y}_{\Lambda_{m-1}}(\mathbf{Y}^T_{\Lambda_{m-1}}\mathbf{Y}_{\Lambda_{m-1}})^{-1}\mathbf{Y}^T_{\Lambda_{m-1}})\mathbf{y}_N\|_2 \leq \|\mathrm{P}_{\mathcal{S}^\perp_L}\mathbf{y}_N\|_2 \leq \|\mathbf{e}_N\|_2.
\tag{56}
$$

According to lemma 4, the event $\bigcap_{i=1}^N\{\|\mathbf{e}_i\|_2 \leq 3\sigma/2\}$ occurs with a probability at least $1 - Ne^{-n/8}$. Hence, we conclude that the following sets of inequalities

$$
\|\mathbf{r}^{(N)}_{m-1,\perp}\|_2 \leq 3\sigma/2, 1 \leq m \leq M,
\tag{57}
$$

$$\|\mathbf{e}_i\|_2 \leq 3\sigma/2, 1 \leq i \leq N, \tag{58}$$

hold at once with a probability at least $1 - Ne^{-n/8}$. Next, we will find a lower bound for $\|\mathbf{r}_{m-1,\|}^{(N)}\|_2$'s. For $1 \leq m \leq M$, we have

$$\begin{aligned}
\mathbf{r}_{m-1,\|}^{(N)} &= \mathrm{P}_{\mathcal{S}_L}(\mathbf{I} - \mathbf{Y}_{\Lambda_{m-1}}(\mathbf{Y}_{\Lambda_{m-1}}^T \mathbf{Y}_{\Lambda_{m-1}})^{-1}\mathbf{Y}_{\Lambda_{m-1}}^T)\mathbf{y}_N \\
&= \mathrm{P}_{\mathcal{S}_L}(\mathbf{y}_N - \mathbf{Y}_{\Lambda_{m-1}}\mathbf{c}_{m-1}^*) = \mathbf{y}_{N,\|} - \mathbf{Y}_{\Lambda_{m-1,\|}}\mathbf{c}_{m-1}^*,
\end{aligned} \tag{59}$$

where $\mathbf{y}_{N,\|} \triangleq \mathrm{P}_{\mathcal{S}_L}\mathbf{y}_N$, $\mathbf{Y}_{\Lambda_{m-1,\|}} \triangleq \mathrm{P}_{\mathcal{S}_L}\mathbf{Y}_{\Lambda_{m-1}}$ and $\mathbf{c}_{m-1}^* \triangleq \underset{\mathbf{c}_{m-1}}{\mathrm{argmin}}\|\mathbf{y}_N - \mathbf{Y}_{\Lambda_{m-1}}\mathbf{c}_{m-1}\|_2$ . Then a lower bound for $\|\mathbf{r}_{m-1,\|}^{(N)}\|_2$ is obtained as

$$\begin{aligned}
\|\mathbf{r}_{m-1,\|}^{(N)}\|_2 &= \|\mathbf{y}_{N,\|} - \mathbf{Y}_{\Lambda_{m-1,\|}}\mathbf{c}_{m-1}^*\|_2 \\
&\geq \min_{\mathbf{c}} \|\mathbf{y}_{N,\|} - \mathbf{Y}_{\Lambda_{m-1,\|}}\mathbf{c}\|_2 \\
&= \|\mathbf{y}_{N,\|} - \mathbf{Y}_{\Lambda_{m-1,\|}}(\mathbf{Y}_{\Lambda_{m-1,\|}}^T \mathbf{Y}_{\Lambda_{m-1,\|}})^{-1}\mathbf{Y}_{\Lambda_{m-1,\|}}^T \mathbf{y}_{N,\|}\|_2 \\
&\geq \|\mathbf{y}_{N,\|} - \mathbf{Y}_{\Lambda_{M-1,\|}}(\mathbf{Y}_{\Lambda_{M-1,\|}}^T \mathbf{Y}_{\Lambda_{M-1,\|}})^{-1}\mathbf{Y}_{\Lambda_{M-1,\|}}^T \mathbf{y}_{N,\|}\|_2 = \|\mathrm{P}_{\mathcal{B}}\mathbf{y}_{N,\|}\|_2,
\end{aligned} \tag{60}$$

where $\mathcal{B} = \mathcal{S}_L \cap \mathcal{R}(\mathbf{Y}_{\Lambda_{M-1,\|}})^{\perp}$ is a subspace of dimension $d_L - p(M-1) > 0$. Since $\mathbf{y}_{N,\|} = \mathbf{x}_N + \mathbf{e}_{N,\|}$ and both the distributions of $\mathbf{x}_N$ and $\mathbf{e}_{N,\|}$ are rotationally invariant in $\mathcal{S}_L$, the normalized $\mathbf{y}_{N,\|}/\|\mathbf{y}_{N,\|}\|_2$ is uniformly distributed over $\mathcal{S}_L \cap \mathbb{S}^{n-1}$. Let $\mathbf{V}_{\mathcal{B}} = [\mathbf{v}_1 \ \mathbf{v}_2...\mathbf{v}_{d_L-p(M-1)}] \in \mathbb{R}^{n \times (d_L-p(M-1))}$ be a matrix whose columns form an orthonormal basis for $\mathcal{B}$; augment $\mathbf{V}_{\mathcal{B}}$ by adding extra $p(M-1)$ columns so that the $d_L$ columns of $\mathbf{V}_{\mathcal{B}} = [\mathbf{v}_1 \ \mathbf{v}_2...\mathbf{v}_{d_L-p(M-1)}...\mathbf{v}_{d_L}] \in \mathbb{R}^{n \times d_L}$ form an orthonormal basis for $\mathcal{S}_L$. Clearly, we have $\mathbf{y}_{N,\|}/\|\mathbf{y}_{N,\|}\|_2 = \mathbf{V}\mathbf{a}$, where $\mathbf{a} = [a_1 \ a_2...a_{d_L}]^T$ obeys uniform distribution over the unit sphere of $\mathbb{R}^{d_L}$ and, in particular, $\mathrm{P}_{\mathcal{B}}(\mathbf{y}_{N,\|}/\|\mathbf{y}_{N,\|}\|_2) = \mathbf{V}_{\mathcal{B}}\widetilde{\mathbf{a}}$, where $\widetilde{\mathbf{a}} = [a_1 \ a_2...a_{d_L-p(M-1)}]^T$ obeys the distribution specified in (Knokhlov, 2006, eq. (7)). Noticing $\|\mathrm{P}_{\mathcal{B}}(\mathbf{y}_{N,\|}/\|\mathbf{y}_{N,\|}\|_2)\|_2 = \|\mathbf{V}_{\mathcal{B}}\widetilde{\mathbf{a}}\|_2 = \|\widetilde{\mathbf{a}}\|_2$, the following set of inequalities hold

$$\begin{aligned}
\Pr\left\{\left\|\mathrm{P}_{\mathcal{B}}\left(\frac{\mathbf{y}_{N,\|}}{\|\mathbf{y}_{N,\|}\|_2}\right)\right\|_2 \leq \lambda\right\} &= \Pr\{\|\widetilde{\mathbf{a}}\|_2 \leq \lambda\} \\
&\overset{(a)}{\leq} \left(\frac{\Gamma(d_L/2)}{\pi^{(d_L-p(M-1))/2}\Gamma(p(M-1)/2)}\right)\int_{\|\widetilde{\mathbf{a}}\|_2 \leq \lambda}(1 - \|\widetilde{\mathbf{a}}\|_2^2)^{(p(M-1)-2)/2}d\widetilde{\mathbf{a}} \\
&\overset{(b)}{\leq} \left(\frac{d_L}{2\pi}\right)^{(d_L-p(M-1))/2}\int_{\|\widetilde{\mathbf{a}}\|_2 \leq \lambda}(1 - \|\widetilde{\mathbf{a}}\|_2^2)^{(p(M-1)-2)/2}d\widetilde{\mathbf{a}} \\
&\leq \left(\frac{d_L}{2\pi}\right)^{(d_L-p(M-1))/2}\int_{\|\widetilde{\mathbf{a}}\|_2 \leq \lambda}1 \, d\widetilde{\mathbf{a}} \\
&= \left(\frac{d_L}{2\pi}\right)^{(d_L-p(M-1))/2}v(d_L - p(M-1))\lambda^{d_L-p(M-1)} \\
&\overset{(c)}{\leq} \left(\frac{d_L}{2\pi}\right)^{(d_L-p(M-1))/2}6\lambda^{d_L-p(M-1)},
\end{aligned} \tag{61}$$

where (a) holds by (Knokhlov, 2006, eq. (7)), (b) follows from (Foucart & Rauhut, 2013, eq. (8.1)), in which $v(r)$ is the volume of unit-ball in $\mathbb{R}^r$, and (c) is true since $v(r) \leq 6$ for all $r \in \mathbb{N}$ (Smith & Vamanamurthy, 1989). Using (60) and (61) with $\lambda = \sigma\sqrt{2/d_L}$, we can obtain

$$\|\mathbf{r}_{m-1,\|}^{(N)}\|_2 > \sqrt{\frac{2}{d_L}}\sigma\|\mathbf{y}_{N,\|}\|_2, \ 1 \leq m \leq M, \tag{62}$$

holds with a probability at least $1 - 6(\sigma/\sqrt{\pi})^{d_L-p(M-1)}$. Since $\mathbf{y}_{N,\|} = \mathbf{x}_N + \mathbf{e}_{N,\|}$ and $\|\mathbf{x}_N\|_2 = 1$ (see Assumption 1), triangle inequality gives $\|\mathbf{y}_{N,\|}\|_2 \geq 1 - \|\mathbf{e}_{N,\|}\|_2$, which together with (62) implies

$$\|\mathbf{r}_{m-1,\|}^{(N)}\|_2 > \sqrt{\frac{2}{d_L}}\sigma(1 - \|\mathbf{e}_{N,\|}\|_2), \ 1 \leq m \leq M, \tag{63}$$

holds with a probability as high as (59). With the aid of (57), (58), and (63), we go on to find a sufficient condition ensuring (55) by finding an upper bound for the LHS of (55). Once (57), (58), and (63) hold, we can obtain the following set of inequalities

$$
\begin{aligned}
&\max_{l \neq L} \frac{\|\mathbf{U}_l^T \mathbf{U}_L\|_F}{\sqrt{d_l}} + \frac{\sqrt{3d_L}\|\mathbf{r}_{m,\perp}^{(N)}\|_2}{\sqrt{8(n-d_L)\log N}\|\mathbf{r}_{m,\|}^{(N)}\|_2} + \frac{2\sqrt{3d_L}(1 + \|\mathbf{e}_N\|_2)}{\sqrt{8n\log N}\|\mathbf{r}_{m-1,\|}^{(N)}\|_2} \max_{1 \leq i \leq N} \|\mathbf{e}_i\|_2 \\
&\leq \max_{l \neq L} \frac{\|\mathbf{U}_l^T \mathbf{U}_L\|_F}{\sqrt{d_l}} + \frac{3\sqrt{3}d_L(3 + 3\sigma)}{(8 - 12\sigma)\sqrt{n\log N}} \\
&\overset{(a)}{\leq} \max_{l \neq L} aff(\mathcal{S}_l, \mathcal{S}_L) + \frac{3\sqrt{3}d_L(3 + 3\sigma)}{(8 - 12\sigma)\sqrt{n\log N}} \overset{(b)}{\leq} \frac{\tau}{4\log N},
\end{aligned}
\tag{64}
$$

where (a) follows from the definition (13) and (b) is true thanks to Assumption 3 As a result, when (57), (58) and (63) hold, the inequality in (64) for all $1 \leq m \leq M$ is guaranteed. By employing the union bound technique it can be concluded that the inequality (55) for all $1 \leq m \leq M$ holds with a probability at least $1 - 6(\sigma/\sqrt{\pi})^{d_L - p(M-1)} - Ne^{-n/8}$. Finally, by using (40), (43), (48), (49), (53) and (55), the proof is thus completed again by using the union bound. □

## 5.2 Proof of Corollary 1

The lower bound (17) is simply obtained as the probability of the event that the numbers $k_m$'s of recovered true neighbors throughout all $M$ iterations yield the maximal lower bound (15). That is to say, (17) is the maximum of (15) over all feasible $0 \leq k_m \leq p$, $1 \leq m \leq M$; more precisely, the minimum objective of the following optimization problem

$$
\begin{aligned}
\min_{(k_1, \ldots, k_m)} & \quad J_0 = \sum_{m=1}^{M} J(k_m) \\
\text{s.t.} & \quad \sum_{m=1}^{M} k_m - k_t = 0 \\
& \quad k_m \in \{0, 1, 2, \ldots, p\}, \ \forall 1 \leq m \leq M,
\end{aligned}
\tag{65}
$$

where

$$
\begin{aligned}
J(k_m) \triangleq & \left( \left( \frac{2e(N - |\mathcal{Y}_L|)}{(p - k_m + 1)N^{8\log N/d_L}} \right)^{p - k_m + 1} \right. \\
& \left. + \left( \sqrt{\frac{2}{\pi}}\tau \right)^{|\mathcal{Y}_L| - d_L - k_m} \left( \frac{e(|\mathcal{Y}_L| - 1)}{k_m - 1} \right)^{k_m - 1} + \frac{4 + 2c}{N^2} \right) \mathbf{1}(k_m > 0)
\end{aligned}
\tag{66}
$$

Below, we will show that the two-level sequence

$$
(k_1, \ldots, k_m) = \Big( \underbrace{q_t + 1, q_t + 1, \ldots, q_t + 1}_{r_t - \text{fold}}, \underbrace{q_t, q_t, \ldots, q_t}_{(M - r_t) - \text{fold}} \Big)
\tag{67}
$$

solves (65), consequently leading to (17). We recall the following definition and lemma, which are needed in our proof.

**Definition 1.** *(Pečarić et al., 1992) Let $\mathbf{x} = [x_1\ x_2 \ldots x_M]^T \in \mathbb{R}^M$ and $\mathbf{y} = [y_1\ y_2 \ldots y_M]^T \in \mathbb{R}^M$ be two real vectors whose entries are ordered in the way that $x_{[1]} \geq x_{[2]} \geq \ldots \geq x_{[M]}$ and $y_{[1]} \geq y_{[2]} \geq \ldots \geq y_{[M]}$, respectively. Then $\mathbf{x}$ is said to be majorized by $\mathbf{y}$ if*

$$
\sum_{m=1}^{s} x_{[m]} \leq \sum_{m=1}^{s} y_{[m]}, \forall 1 \leq s \leq M
\tag{68}
$$

*and*

$$
\sum_{m=1}^{s} x_m = \sum_{m=1}^{s} y_m.
\tag{69}
$$

*We say $f : \mathcal{A} \subset \mathbb{R}^M$ is Schur-convex if $f(\mathbf{x}) \leq f(\mathbf{y})$ whenever $\mathbf{x}$ is majorized by $\mathbf{y}$, $\mathbf{x}, \mathbf{y} \in \mathcal{A}$.* □

**Lemma 5.** *(Pečarić et al., 1992, Theorem 12.25) Let $f : \mathcal{A} \subset \mathbb{R}^M$ be a permutation invariant function, that is, $f(\mathbf{x}) = f(\mathbf{Px})$ for all $\mathbf{x} \in \mathcal{A}$ and permutation matrices $\mathbf{P} \in \mathbb{R}^{M \times M}$, whose first partial derivatives exist in $\mathcal{A}$. Then $f$ is Schur-convex in $\mathcal{A}$ if and only if*

$$(x_i - x_j)\left(\frac{\partial f}{\partial x_i} - \frac{\partial f}{\partial x_j}\right) \geq 0, \forall \mathbf{x} = [x_1 \ x_2 ... x_M]^T \in \mathcal{A} \tag{70}$$

*holds for all $1 \leq i \neq j \leq M$.* □

Using vector-matrix notation, all we have to do is to show

$$\mathbf{k}^* \triangleq \Big[ \underbrace{q_t + 1, q_t + 1, ..., q_t + 1}_{r_t-\text{fold}}, \underbrace{q_t, q_t, ..., q_t}_{(M-r_t)-\text{fold}} \Big]^T \tag{71}$$

solves the following optimization problem:

$$\min_{\mathbf{k}=[k_1,...,k_m]^T} J_0 = \sum_{m=1}^{M} J(\mathbf{k}^T \mathbf{v}_m) \tag{72}$$
$$\text{s.t. } \mathbf{k}^T \mathbf{1} = k_t$$
$$\mathbf{k}^T \mathbf{v}_m \in \{0, 1, 2, ..., p\}, \ \forall 1 \leq m \leq M,$$

in which $\mathbf{v}_m \triangleq [\ \underbrace{0...0}_{m-1 \text{ fold}} \ 1 \ 0...0\ ]^T$ is the $m$th standard unit vector. It suffices to prove

$$J_0(\mathbf{k}^*) \leq J_0(\mathbf{k}) \tag{73}$$

for any $\mathbf{k} \in \mathcal{D}_J$, the feasible set of (72), based on Lemma 5. For this, we first note that the objective $J_0$ is not differentiable, since the function $J$ in (66) involves the indicator function. To rid of this difficulty, we consider the differentiable surrogate $\widetilde{J} : \mathbb{R} \to \mathbb{R}$ for $J$, constructed according to

$$\widetilde{J}(x) = J(x), \forall x \geq 0.9 \text{ and } x = 0, \tag{74}$$

and

$$\widetilde{J}'(0) = J'(1). \tag{75}$$

Thanks to (74), the function $\widetilde{J}$ thus obtained satisfies $\widetilde{J}(k_m) = J(k_m)$ for all $k_m \in \{0, 1, ..., p\}$ and $\widetilde{J}'(k_m) = J'(k_m)$ for all $k_m \in \{1, 2, ..., p\}$. The corresponding differentiable surrogate for $J_0$ is accordingly given by

$$\widetilde{J}_0(\mathbf{k}) = \sum_{m=1}^{M} \widetilde{J}(\mathbf{k}^T \mathbf{v}_m). \tag{76}$$

Clearly, $\widetilde{J}_0(\mathbf{k}) = J_0(\mathbf{k})$ for all $\mathbf{k} \in \mathcal{D}_J$, due to condition (74); all the better, $\widetilde{J}_0$ is Schur-convex in $\mathcal{D}_J$ (a proof is given in Appendix C). Hence, for $\mathbf{k}, \mathbf{q} \in \mathcal{D}_J$ such that $\mathbf{k}$ is majorized by $\mathbf{q}$, we have $J_0(\mathbf{k}) = \widetilde{J}_0(\mathbf{k}) \leq \widetilde{J}_0(\mathbf{q}) = J_0(\mathbf{q})$. The inequality (73) is guaranteed once $\mathbf{k}^*$ is majorized by any feasible $\mathbf{k}$, which is indeed true as shown below. For a feasible $\mathbf{k} = [k_1 \ k_2 ... k_M]^T$ such that $k_1 \geq k_2 \geq ... \geq k_M$ without loss of generality. By Definition 1, it suffices to show

$$\sum_{m=1}^{s} k_m^* \leq \sum_{m=1}^{s} k_m, \forall 1 \leq s \leq M. \tag{77}$$

Assume otherwise that there exists $1 \leq s \leq M$ such that $\sum_{m=1}^{q} k_m^* \leq \sum_{m=1}^{q} k_m$, $1 \leq q \leq s-1$, whereas $\sum_{m=1}^{s} k_m^* > \sum_{m=1}^{s} k_m$. Then we have $k_s^* > k_s$, which together with $k_s^* \in \{q_t, q_t + 1\}$ implies $q_t \geq k_m$, for $s + 1 \leq m \leq M$, ending in the following contradiction:

$$k_t = \sum_{m=1}^{s} k_m + \sum_{m=s+1}^{M} k_m \leq \sum_{m=1}^{s} k_m + \sum_{m=s+1}^{M} q_t \leq \sum_{m=1}^{s} k_m + \sum_{m=s+1}^{M} k_m^* < \sum_{m=1}^{s} k_m^* + \sum_{m=s+1}^{M} k_m^* = k_t. \tag{78}$$

□

### 5.3 Proof of Theorem 2

Below we first derive an equivalent condition for the proposed stopping rule that is more amenable to analysis. Recall the residual $\mathbf{r}_m^{(N)}$ obtained in the $m$th iteration is the orthogonal projection of the previous residual $\mathbf{r}_{m-1}^{(N)}$ onto $\mathcal{R}(\mathbf{Y}_{\Lambda_m})^{\perp}$. Since $\|\mathbf{r}_{m-1}^{(N)}\|_2 = \|\mathbf{r}_m^{(N)}\|_2 + \|\mathbf{r}_{m-1}^{(N)} - \mathbf{r}_m^{(N)}\|_2$, obtained from the Pythagorean theorem, the proposed stopping rule can be rewritten as

$$1 - \frac{\|\mathbf{r}_m^{(N)}\|_2}{\|\mathbf{r}_{m-1}^{(N)}\|_2} = 1 - \sqrt{1 - \|\widetilde{\mathbf{r}}_m^{(N)}\|_2^2} \leq \sqrt{\frac{p}{n}}, \tag{79}$$

where

$$\widetilde{\mathbf{r}}_m^{(N)} \triangleq \frac{\mathbf{r}_{m-1}^{(N)} - \mathbf{r}_m^{(N)}}{\|\mathbf{r}_{m-1}^{(N)}\|_2} \in \mathcal{R}(\mathbf{Y}_{\Lambda_m}) \tag{80}$$

is the normalized difference of the residual vectors. Rearranging the inequality in (79) yields the following equivalent stopping condition

$$\|\widetilde{\mathbf{r}}_m^{(N)}\|_2 \leq \sqrt{2\sqrt{p/n} - p/n}. \tag{81}$$

Below we show that, with a high chance, (81) does not hold when $m \leq \lceil d_L/p \rceil$ and is achieved (hence, GOMP stops) with $m = \lceil d_L/p \rceil + 1$. Formally, we prove the following inequalities

$$\|\widetilde{\mathbf{r}}_m^{(N)}\|_2 > \sqrt{2\sqrt{p/n} - p/n}, \ \forall 1 \leq m \leq \lceil d_L/p \rceil \tag{82}$$

and

$$\|\widetilde{\mathbf{r}}_{\lceil d_L/p \rceil + 1}^{(N)}\|_2 \leq \sqrt{2\sqrt{p/n} - p/n} \tag{83}$$

hold at once with a probability as high as claimed by Theorem 2.

To proceed, let $\mathbf{y}_j$ be a selected data vector in the $m$th iteration, $1 \leq m \leq \lceil d_L/p \rceil$. As long as (49), (53), (57), (58) and (63) hold, we have

$$
\begin{aligned}
|\langle \mathbf{y}_j, \mathbf{r}_{m-1}^{(N)} \rangle| &\overset{(a)}{\geq} |\langle \mathbf{y}_{k_m}, \mathbf{r}_{m-1}^{(N)} \rangle| \overset{(b)}{\geq} |\langle \mathbf{x}_{\bar{t}_{k_m}}, \mathbf{r}_{m-1,\|}^{(N)} \rangle| - \max_{1 \leq j \leq |\mathcal{Y}_L|-1-|\Lambda_{m-1}^t|} |\langle \mathbf{e}_{t_j}, \mathbf{r}_{m-1}^{(N)} \rangle| \\
&\overset{(c)}{\geq} \left[ \frac{\|\mathbf{r}_{m-1,\|}^{(N)}\|_2}{\sqrt{d_L}} - \sqrt{\frac{6\log N}{n}} \|\mathbf{r}_{m-1,\|}^{(N)}\|_2 \max_{1 \leq i \leq N} \|\mathbf{e}_i\|_2 \right] \\
&= \|\mathbf{r}_{m-1}^{(N)}\|_2 \left[ \frac{\|\mathbf{r}_{m-1,\|}^{(N)}\|_2}{\sqrt{d_L(\|\mathbf{r}_{m-1,\perp}^{(N)}\|_2^2 + \|\mathbf{r}_{m-1,\|}^{(N)}\|_2^2)}} - \sqrt{\frac{6\log N}{n}} \max_{1 \leq i \leq N} \|\mathbf{e}_i\|_2 \right] \\
&\overset{(d)}{\geq} \|\mathbf{r}_{m-1}^{(N)}\|_2 \underbrace{\left[ \frac{\sqrt{2}(1 - 3\sigma/2)}{\sqrt{9d_L^2/4 + 4(1-3\sigma/2)^2 d_L}} - \frac{3\sqrt{6\log N}\sigma}{2\sqrt{n}} \right]}_{\triangleq \eta},
\end{aligned} \tag{84}
$$

in which (a) is true since $\mathbf{y}_j$ is selected as a neighbor, (b) follows form (35), (c) holds due to (49) and (53), and (d) is obtained by combining (57), (58), and (63). By definition (80) we have $\widetilde{\mathbf{r}}_m^{(N)} = \mathrm{P}_{\mathcal{R}(\mathbf{Y}_{\Lambda_m})}(\mathbf{r}_{m-1}^{(N)}/\|\mathbf{r}_{m-1}^{(N)}\|_2)$, hence

$$\|\widetilde{\mathbf{r}}_m^{(N)}\|_2 \geq \|\mathrm{P}_{span\{\mathbf{y}_j\}}\left(\frac{\mathbf{r}_{m-1}^{(N)}}{\|\mathbf{r}_{m-1}^{(N)}\|_2}\right)\|_2 = \frac{|\langle \mathbf{y}_j, \mathbf{r}_{m-1}^{(N)} \rangle|}{\|\mathbf{y}_j\|_2 \|\mathbf{r}_{m-1}^{(N)}\|_2}, \tag{85}$$

which together with (84) implies

$$\|\widetilde{\mathbf{r}}_m^{(N)}\|_2 \geq \frac{|\langle \mathbf{y}_j, \mathbf{r}_{m-1}^{(N)} \rangle|}{\|\mathbf{y}_j\|_2 \|\mathbf{r}_{m-1}^{(N)}\|_2} \geq \frac{\eta \|\mathbf{r}_{m-1}^{(N)}\|_2}{\|\mathbf{y}_j\|_2 \|\mathbf{r}_{m-1}^{(N)}\|_2} = \frac{\eta}{\|\mathbf{y}_j\|_2}. \tag{86}$$

With the aid of (86), inequality (82) is guaranteed once $\eta/\|\mathbf{y}_j\|_2 > \sqrt{2\sqrt{p/n} - p/n}$, which is typically true since the ambient dimension $n$ is drastically large.

As for (83), again by definition (80) we write

$$\widetilde{\mathbf{r}}_{\lceil d_L/p\rceil+1}^{(N)} = \mathrm{P}_{\mathcal{R}(\mathbf{Y}_{\Lambda_{\lceil d_L/p\rceil+1}})}(\mathbf{r}_{\lceil d_L/p\rceil}^{(N)}/\|\mathbf{r}_{\lceil d_L/p\rceil}^{(N)}\|_2). \tag{87}$$

Let $\{\mathbf{b}_1,...,\mathbf{b}_{(\lceil d_L/p\rceil+1)p}\}$ be an orthonormal basis of $\mathcal{R}(\mathbf{Y}_{\Lambda_{\lceil d_L/p\rceil+1}})$, arranged in a way that $\{\mathbf{b}_{p+1},...,\mathbf{b}_{(\lceil d_L/p\rceil+1)p}\}$ is an orthonormal basis of $\mathcal{R}(\mathbf{Y}_{\Lambda_{\lceil d_L/p\rceil}})$. Since $\mathbf{r}_{\lceil d_L/p\rceil}^{(N)}$ is orthogonal to $\mathcal{R}(\mathbf{Y}_{\Lambda_{\lceil d_L/p\rceil}})$, $\langle \mathbf{r}_{\lceil d_L/p\rceil}^{(N)}, \mathbf{b}_j\rangle = 0$ for all $p+1 \le j \le (\lceil d_L/p\rceil + 1)p$. Thus, it follows

$$\begin{aligned}
&\Pr\big\{\|\widetilde{\mathbf{r}}_{\lceil d_L/p\rceil+1}^{(N)}\|_2 \le \sqrt{2\sqrt{p/n} - p/n}\big\} \\
&= \Pr\Big\{\|\mathrm{P}_{\mathcal{R}(\mathbf{Y}_{\Lambda_{\lceil d_L/p\rceil+1}})}(\mathbf{r}_{\lceil d_L/p\rceil}^{(N)}/\|\mathbf{r}_{\lceil d_L/p\rceil}^{(N)}\|_2)\|_2 \le \sqrt{2\sqrt{p/n} - p/n}\Big\} \\
&\ge \Pr\Big\{\bigcap_{k=1}^{p}\big\{\|\mathrm{P}_{span\{\mathbf{b}_k\}}(\mathbf{r}_{\lceil d_L/p\rceil}^{(N)}/\|\mathbf{r}_{\lceil d_L/p\rceil}^{(N)}\|_2)\|_2 \le \sqrt{2\sqrt{p/n} - p/n}/\sqrt{p}\big\}\Big\} \\
&= 1 - \Pr\Big\{\bigcup_{k=1}^{p}\big\{\|\mathrm{P}_{span\{\mathbf{b}_k\}}(\mathbf{r}_{\lceil d_L/p\rceil}^{(N)}/\|\mathbf{r}_{\lceil d_L/p\rceil}^{(N)}\|_2)\|_2 > \sqrt{2\sqrt{p/n} - p/n}/\sqrt{p}\big\}\Big\} \\
&\ge 1 - \sum_{k=1}^{p}\Pr\Big\{\|\mathrm{P}_{span\{\mathbf{b}_k\}}(\mathbf{r}_{\lceil d_L/p\rceil}^{(N)}/\|\mathbf{r}_{\lceil d_L/p\rceil}^{(N)}\|_2)\|_2 > \sqrt{2\sqrt{p/n} - p/n}/\sqrt{p}\Big\} \\
&= 1 - \sum_{k=1}^{p}\Pr\Big\{|\langle\mathbf{r}_{\lceil d_L/p\rceil}^{(N)}/\|\mathbf{r}_{\lceil d_L/p\rceil}^{(N)}\|_2, \mathbf{b}_k\rangle| > \sqrt{2\sqrt{p/n} - p/n}/\sqrt{p}\Big\}.
\end{aligned} \tag{88}$$

According to our proof of Theorem 1, the neighbors selected in all $\lceil d_L/p\rceil$ iterations (i.e., all columns of $\mathbf{Y}_{\Lambda_{\lceil d_L/p\rceil}}$) are correct when (40), (43), (48), (49), (53), (57), (58) and (63) hold for all $1 \le m \le \lceil d_L/p\rceil$. If so, there then exists $\widetilde{\mathbf{c}} \in \mathbb{R}^{p\lceil d_L/p\rceil}$ such that the "noiseless" signal vector $\mathbf{x}_N$ and neighbor matrix $\mathbf{X}_{\Lambda_{\lceil d_L/p\rceil}}$ satisfy

$$\mathbf{x}_N = \mathbf{X}_{\Lambda_{\lceil d_L/p\rceil}}\widetilde{\mathbf{c}}. \tag{89}$$

Noting that $\mathbf{Y}_{\Lambda_{\lceil d_L/p\rceil}} = \mathbf{X}_{\Lambda_{\lceil d_L/p\rceil}} + \mathbf{E}_{\Lambda_{\lceil d_L/p\rceil}}$, where $\mathbf{E}_{\Lambda_{\lceil d_L/p\rceil}}$ is the noise matrix, the residual vector at the $\lceil d_L/p\rceil$th iteration can be written as

$$\begin{aligned}
\mathbf{r}_{\lceil d_L/p\rceil}^{(N)} &= (\mathbf{I} - \mathbf{Y}_{\Lambda_{\lceil d_L/p\rceil}}(\mathbf{Y}_{\Lambda_{\lceil d_L/p\rceil}}^T \mathbf{Y}_{\Lambda_{\lceil d_L/p\rceil}})^{-1}\mathbf{Y}_{\Lambda_{\lceil d_L/p\rceil}}^T)(\mathbf{x}_N + \mathbf{e}_N) \\
&= (\mathbf{I} - \mathbf{Y}_{\Lambda_{\lceil d_L/p\rceil}}(\mathbf{Y}_{\Lambda_{\lceil d_L/p\rceil}}^T \mathbf{Y}_{\Lambda_{\lceil d_L/p\rceil}})^{-1}\mathbf{Y}_{\Lambda_{\lceil d_L/p\rceil}}^T)(\mathbf{x}_N + \mathbf{Y}_{\Lambda_{\lceil d_L/p\rceil}}\widetilde{\mathbf{c}} + \mathbf{e}_N) \\
&= (\mathbf{I} - \mathbf{Y}_{\Lambda_{\lceil d_L/p\rceil}}(\mathbf{Y}_{\Lambda_{\lceil d_L/p\rceil}}^T \mathbf{Y}_{\Lambda_{\lceil d_L/p\rceil}})^{-1}\mathbf{Y}_{\Lambda_{\lceil d_L/p\rceil}}^T)(\mathbf{E}_{\Lambda_{\lceil d_L/p\rceil}}\widetilde{\mathbf{c}} + \mathbf{e}_N),
\end{aligned} \tag{90}$$

which is the projection of the Gaussian random vector $\mathbf{E}_{\Lambda_{\lceil d_L/p\rceil}}\widetilde{\mathbf{c}} + \mathbf{e}_N$ onto $\mathcal{R}(\mathbf{Y}_{\Lambda_{\lceil d_L/p\rceil}})^\perp$. Being a linear combination of rotationally invariant vectors, $\mathbf{E}_{\Lambda_{\lceil d_L/p\rceil}}\widetilde{\mathbf{c}} + \mathbf{e}_N$ remains so; this implies the projection $\mathbf{r}_{\lceil d_L/p\rceil}^{(N)}$ is also rotationally invariant on $\mathcal{R}(\mathbf{Y}_{\Lambda_{\lceil d_L/p\rceil}})^\perp$. Then using (28) with $\mathbf{a} = \mathbf{r}_{\lceil d_L/p\rceil}^{(N)}/\|\mathbf{r}_{\lceil d_L/p\rceil}^{(N)}\|_2$ and $\mathbf{b} = \mathbf{b}_k$, $1 \le k \le p$, we can obtain

$$\Pr\Big\{|\langle\mathbf{r}_{\lceil d_L/p\rceil}^{(N)}/\|\mathbf{r}_{\lceil d_L/p\rceil}^{(N)}\|_2, \mathbf{b}_k\rangle| > \sqrt{2\sqrt{p/n} - p/n}/\sqrt{p}\Big\} \approx 2e^{-\sqrt{n/p}}, \tag{91}$$

which together with (88) implies (83) holds with a probability at least $1 - 2pe^{-\sqrt{n/p}}$. In summary, once the inequalities (40), (43), (48), (49), (53), (57), (58), and (63) hold for all $1 \le m \le \lceil d_L/p\rceil$, then (82) and (83) hold with a probability at least $1 - 2pe^{-\sqrt{n/p}}$. The proof is then completed by employing the union bound.

$\square$

## 6 Conclusion

Fast acquisition of many true neighbors underlies the success of SSC in real-world applications. Under the framework of greedy selection, which is pretty suited for low-complexity implementation, we propose a GOMP sparse regression scheme, together with a novel subspace-dimension-aware stopping rule, to boost neighbor recovery. Thanks to multiple neighbor identification per iteration, the proposed GOMP involves fewer iterations, thereby enjoying even lower algorithmic complexity than conventional OMP. In addition, the residual vector better stands up to noise corruption, consequently bringing about higher neighbor identification accuracy. Our proposed stopping criterion is appealing in that it depends entirely on a knowledge of the ambient space dimension, in marked contrast with the existing solution (Tschannen & Bölcskei, 2018) that requires extra off-line estimation of either subspace dimension or noise power. Besides algorithm development, in-depth neighbor recovery rate analyses were conducted to justify the merits of the proposed GOMP; the obtained analytic results are further validated by computations using both synthetic and real datasets. Overall, our study presents a new greedy-based SSC scheme, with provable performance guarantees, that can pave the way for practical applications. Future work will analyze AoD of both GOMP and OMP under the considered semi-random model, particularly to show GOMP enjoys a smaller average AoD; an analytic study of AoD would further offer certain guidelines on determining the optimum number of neighbors to recover in each iteration, for the purpose of improving the current scheme, which identifies a constant number ($p \geq 1$) of neighbors throughout all iterations, in a more dynamic environment.

### Acknowledgments

This work is sponsored by the National Council of Science and Technology of Taiwan under grants NSTC 111-2221-E-A49-067-MY3, NSTC 112-2811-E-A49-545-MY2, and MOST 110-2221-E-A49-042-MY3. The work of C.-H. Liu was supported in part by the National Science Foundation under Awards CNS-2006453 and ECCS-2210106.

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

## A    Appendix

The following lemma is needed for deriving Lemma 2.

**Lemma 6.** *(Soltanolkotabi & Candès, 2012, Extracted from the proof of Lemma 7.5) Let $\mathbf{a} \in \mathbb{R}^{d_1}$ be distributed uniformly on $\{\mathbf{x} \mid \mathbf{x} \in \mathbb{R}^{d_1}, \|\mathbf{x}\|_2 = 1\}$ and $\mathbf{B} \in \mathbb{R}^{d_2 \times d_1}$. Then for $\alpha > 0$ we have*

$$Pr\left\{ \|\mathbf{Ba}\|_2 > \alpha \frac{\|\mathbf{B}\|_F}{\sqrt{d_2}} \right\} \leq 2e^{-\alpha^2/2}. \tag{92}$$

$\square$

Assume $\mathbf{x}_i \in \mathcal{S}_l$, $l \neq L$. Then we have $\mathbf{x}_i = \mathbf{U}_l \mathbf{b}$, in which $\mathbf{b}$ is uniformly distributed over the unit sphere of $\mathbb{R}^{d_l}$ by Assumption 1. Setting $\mathbf{C} = \mathbf{U}_L^T \mathbf{U}_l$ and $\alpha = 4 \log N / \sqrt{d_L}$, Lemma 6 implies

$$\Pr\left\{ \|\mathbf{U}_L^T \mathbf{x}_i\|_2 > 4 \log N \frac{\|\mathbf{U}_L^T \mathbf{U}_l\|_F}{\sqrt{d_L d_l}} \right\} \leq 2 e^{-8(\log N)^2/d_L} = 2 \left( e^{\log N} \right)^{-8 \log N / d_L} = \frac{2}{N^{(8 \log N)/d_L}}. \tag{93}$$

Since $\mathbf{r}_{m,\|}^{(N)} \in \mathcal{S}_L$, we have $\mathbf{r}_{m,\|}^{(N)}/\|\mathbf{r}_{m,\|}^{(N)}\|_2 = \mathbf{U}_L \mathbf{z}$ with $\|\mathbf{z}\|_2 = 1$. The Cauchy-Schwartz inequality implies

$$\left| \left\langle \mathbf{r}_{m,\|}^{(N)}/\|\mathbf{r}_{m,\|}^{(N)}\|_2, \mathbf{x}_i \right\rangle \right| = |\mathbf{z}^T \mathbf{U}_L^T \mathbf{x}_i| \leq \|\mathbf{z}\|_2 \|\mathbf{U}_L^T \mathbf{x}_i\|_2 = \|\mathbf{U}_L^T \mathbf{x}_i\|_2, \tag{94}$$

and therefore

$$\left\{ \left| \left\langle \mathbf{x}_i, \mathbf{r}_{m,\|}^{(N)}/\|\mathbf{r}_{m,\|}^{(N)}\|_2 \right\rangle \right| > 4 \log(N) \frac{\|\mathbf{U}_l^T \mathbf{U}_L\|_F}{\sqrt{d_L d_l}} \right\} \subset \left\{ \|\mathbf{U}_L^T \mathbf{x}_i\|_2 > 4 \log(N) \frac{\|\mathbf{U}_l^T \mathbf{U}_L\|_F}{\sqrt{d_L d_l}} \right\}. \tag{95}$$

With (93) and (95), it follows immediately

$$\Pr\left\{ \left| \left\langle \mathbf{x}_i, \mathbf{r}_{m,\|}^{(N)}/\|\mathbf{r}_{m,\|}^{(N)}\|_2 \right\rangle \right| > 4 \log(N) \frac{\|\mathbf{U}_l^T \mathbf{U}_L\|_F}{\sqrt{d_l d_L}} \right\} \leq \frac{2}{N^{(8 \log N)/d_L}}. \tag{96}$$

$$\square$$

## Appendix B

Let

$$\mathcal{D}(\mathbf{v}) \triangleq \left\{ (\mathbf{y}_1, \mathbf{y}_2, ..., \mathbf{y}_N) : \mathbf{r}_{m,\perp}^{(N)}/\|\mathbf{r}_{m,\perp}^{(N)}\|_2 = \mathbf{v} \right\}, \tag{97}$$

$f : \mathbb{R}^n \times ... \times \mathbb{R}^n \to \mathbb{R}$ be the probability density function of $(\mathbf{y}_1, \mathbf{y}_2, ..., \mathbf{y}_N)$, and $\mathbb{S}^{n-1}$ be the unit-sphere of $\mathbb{R}^n$. Then for $\mathbf{z} \in \mathcal{S}_L^\perp \cap \mathbb{S}^{n-1}$, we have

$$\begin{aligned}
\Pr\left\{ |\langle \mathbf{z}, \mathbf{r}_{m,\perp}^{(N)}/\|\mathbf{r}_{m,\perp}^{(N)}\|_2 \rangle| > \epsilon \right\} &= \int\limits_{(\mathcal{S}_L^\perp \cap \mathbb{S}^{n-1}) \cap \{\mathbf{r}:|\langle \mathbf{r}, \mathbf{z} \rangle| > \epsilon\}} \left[ \int\limits_{\mathcal{D}(\mathbf{v})} f(\mathbf{y}_1, \mathbf{y}_2, ..., \mathbf{y}_N) d\mathbf{y}_1 d\mathbf{y}_2 ... d\mathbf{y}_N \right] d\mathbf{v} \\
&\leq \int\limits_{(\mathcal{S}_L^\perp \cap \mathbb{S}^{n-1}) \cap \{\mathbf{r}:|\langle \mathbf{r}, \mathbf{z} \rangle| \geq \epsilon\}} \left[ \int\limits_{\mathcal{K} \cap \mathcal{D}(\mathbf{v})} f(\mathbf{y}_1, \mathbf{y}_2, ..., \mathbf{y}_N) d\mathbf{y}_1 d\mathbf{y}_2 ... d\mathbf{y}_N \right. \\
&\qquad\qquad \left. + \int\limits_{\mathcal{K}^c \cap \mathcal{D}(\mathbf{v})} f(\mathbf{y}_1, \mathbf{y}_2, ..., \mathbf{y}_N) d\mathbf{y}_1 d\mathbf{y}_2 ... d\mathbf{y}_N \right] d\mathbf{v},
\end{aligned} \tag{98}$$

where the last inequality holds by defining

$$\mathcal{K} \triangleq \{\mathbf{x} + \mathbf{e} : \mathbf{x} \in \mathcal{S}_{k_1} \cap \mathbb{S}^{n-1}, \|\mathbf{e}\|_2 \leq 3\sigma/2\} \times ... \times \{\mathbf{x} + \mathbf{e} : \mathbf{x} \in \mathcal{S}_{k_N} \cap \mathbb{S}^{n-1}, \|\mathbf{e}\|_2 \leq 3\sigma/2\}, \tag{99}$$

in which we assume that $\mathbf{y}_i \in \mathcal{Y}_{k_i}$, hence $\mathbf{x}_i \in \mathcal{S}_{k_i}$, for $1 \leq i \leq N$. Since the maximum function is continuous and the composition of continuous functions is also continuous, $\mathbf{r}_{m,\perp}^{(N)}/\|\mathbf{r}_{m,\perp}^{(N)}\|_2$ is a continuous function of $(\mathbf{y}_1, \mathbf{y}_2, ..., \mathbf{y}_N)$. Consequently, since $(\mathcal{S}_L^\perp \cap \mathbb{S}^{n-1}) \cap \{\mathbf{r} : |\langle \mathbf{r}, \mathbf{z} \rangle| \geq \epsilon\}$ is closed, the inverse image of $\mathcal{D}((\mathcal{S}_L^\perp \cap \mathbb{S}^{n-1}) \cap \{\mathbf{r} : |\langle \mathbf{r}, \mathbf{z} \rangle| \geq \epsilon\})$ is also closed (Marsden & Hoffman, 1993). Notice that $\mathcal{K} \cap \mathcal{D}((\mathcal{S}_L^\perp \cap \mathbb{S}^{n-1}) \cap \{\mathbf{r} : |\langle \mathbf{r}, \mathbf{z} \rangle| \geq \epsilon\})$ is closed and bounded in the Euclidean space, and is therefore compact by the Heine-Borel theorem (Marsden & Hoffman, 1993). Since $f(\mathbf{y}_1, \mathbf{y}_2, ..., \mathbf{y}_N)$ is continuous on the compact set $\mathcal{K} \cap \mathcal{D}((\mathcal{S}_L^\perp \cap \mathbb{S}^{n-1}) \cap \{\mathbf{r} : |\langle \mathbf{r}, \mathbf{z} \rangle| \geq \epsilon\})$, by the extreme value theorem (Marsden & Hoffman, 1993) there exists a constant $\delta$ such that

$$f(\mathbf{y}_1, \mathbf{y}_2, ..., \mathbf{y}_N) \leq \delta, \ \forall (\mathbf{y}_1, \mathbf{y}_2, ..., \mathbf{y}_N) \in \mathcal{K} \cap \mathcal{D}((\mathcal{S}_L^\perp \cap \mathbb{S}^{n-1}) \cap \{\mathbf{r} : |\langle \mathbf{r}, \mathbf{z} \rangle| \geq \epsilon\}), \tag{100}$$

which together with (98) implies

$$\Pr\Big\{\Big|\Big\langle \mathbf{z}, \frac{\mathbf{r}_{m,\perp}^{(N)}}{\|\mathbf{r}_{m,\perp}^{(N)}\|_2}\Big\rangle\Big| > \epsilon\Big\} = \int\limits_{(\mathcal{S}_L^\perp \cap \mathbb{S}^{n-1}) \cap \{\mathbf{r}:|\langle \mathbf{r},\mathbf{z}\rangle| \geq \epsilon\}} \Big[\delta|\mathcal{K}| + \int\limits_{\mathcal{K}^c \cap \mathcal{D}(\mathbf{v})} f(\mathbf{y}_1,\mathbf{y}_2,...,\mathbf{y}_N)d\mathbf{y}_1 d\mathbf{y}_2...d\mathbf{y}_N\Big]d\mathbf{v}$$

$$\overset{(a)}{\leq} \int\limits_{(\mathcal{S}_L^\perp \cap \mathbb{S}^{n-1}) \cap \{\mathbf{r}:|\langle \mathbf{r},\mathbf{z}\rangle| \geq \epsilon\}} \Big[\delta|\mathcal{K}| + Ne^{-n/8}\Big]d\mathbf{v}$$

$$\overset{(b)}{=} \int\limits_{(\mathcal{S}_L^\perp \cap \mathbb{S}^{n-1}) \cap \{\mathbf{r}:|\langle \mathbf{r},\mathbf{z}\rangle| \geq \epsilon\}} \Big[c/A(\mathcal{S}_L^\perp \cap \mathbb{S}^{n-1})\Big]d\mathbf{v} \overset{(c)}{\leq} 2ce^{-(n-d_L)\epsilon^2/2}, \tag{101}$$

where (a) holds by lemma 4 with $|\mathcal{K}|$ here denoting the Lebesgue measure of $\mathcal{K}$, (b) is true as we define $A(\mathcal{S}_L^\perp \cap \mathbb{S}^{n-1})$ to be the surface area of $\mathcal{S}_L^\perp \cap \mathbb{S}^{n-1}$ and the constant $c$ is defined as $c \triangleq (\delta|\mathcal{K}| + Ne^{-n/8})A(\mathcal{S}_L^\perp \cap \mathbb{S}^{n-1})$, and (c) follows from (28). With (101), we reach

$$\Pr\Big\{\Big|\Big\langle \mathbf{x}_i, \mathbf{r}_{m,\perp}^{(N)}/\|\mathbf{r}_{m,\perp}^{(N)}\|_2\Big\rangle\Big| > \epsilon\Big\} = \int\limits_{\mathbb{R}^n} \Pr\Big\{\Big|\Big\langle \mathbf{x}_i, \mathbf{r}_{m,\perp}^{(N)}/\|\mathbf{r}_{m,\perp}^{(N)}\|_2\Big\rangle\Big| > \epsilon\Big|\mathbf{x}_i = \mathbf{v}\Big\}f_{\mathbf{x}_i}(\mathbf{v})d\mathbf{v}$$

$$\overset{(a)}{\leq} \int\limits_{\mathbb{R}^n} 2en^{-(n-d_L)\epsilon^2/2}f_{\mathbf{x}_i}(\mathbf{v})d\mathbf{v} = 2en^{-(n-d_L)\epsilon^2/2}, \tag{102}$$

where (a) holds by (101), in which $f_{\mathbf{x}_i}$ is the probability density function of $\mathbf{x}_i$. Let $\epsilon = \sqrt{6\log N/(n-d_L)}$. Then (102) gives

$$\Pr\Big\{\Big|\Big\langle \mathbf{x}_i, \mathbf{r}_{m,\perp}^{(N)}/\|\mathbf{r}_{m,\perp}^{(N)}\|_2\Big\rangle\Big| > \sqrt{\frac{6\log N}{n-d_L}}\Big\} \leq 2cN^{-3}. \tag{103}$$

$\square$

## Appendix C

In this appendix, we prove the Schur-convexity of $\widetilde{J}_0$. Notably, straightforward manipulations show

$$(k_m - k_q)\Big(\frac{\partial J_0}{\partial k_m} - \frac{\partial J_0}{\partial k_q}\Big) = (k_m - k_q)(J'(k_m) - J'(k_q)) \geq 0, \ \forall k_m, k_q \in \{1,2,...,p\}. \tag{104}$$

It then follows

$$(k_m - k_q)\Big(\frac{\partial \widetilde{J}_0}{\partial k_m} - \frac{\partial \widetilde{J}_0}{\partial k_q}\Big) = (k_m - k_q)(\widetilde{J}'(k_m) - \widetilde{J}'(k_q)) \overset{(a)}{=} (k_m - k_q)(J'(k_m) - J'(k_q)),$$

$$\forall \ k_m, k_q \in \{1,2,...,p\}. \tag{105}$$

where (a) holds by (74) and (b) is due to (104). When evaluated around $k_m \in \{1,2,...,p\}$ and $k_q = 0$, the resultant partial derivative reads

$$(k_m - k_q)\Big(\frac{\partial \widetilde{J}_0}{\partial k_m} - \frac{\partial \widetilde{J}_0}{\partial k_q}\Big) = k_m(\widetilde{J}'(k_m) - \widetilde{J}'(0)) \overset{(a)}{=} k_m(J'(k_m) - \widetilde{J}'(0))$$

$$\overset{(b)}{=} k_m(J'(k_m) - J'(1)) \overset{(c)}{\geq} 0, \tag{106}$$

where (a) holds by (74), (b) follows from (75), and (c) is true due to (104). Clearly, the function $\widetilde{J}_0$ is permutation invariant, together with (105) and (106) ensures that $\widetilde{J}_0$ satisfies the conditions of Lemma 5, and thus is Schur-convex. $\square$

