# OpenReview forum: "Greedier is Better: Selecting Multiple Neighbors per Iteration for Sparse Subspace Clustering"
_TMLR — Accepted by TMLR_

### Review · Reviewer_9QED · 2023-03-30

**Summary Of Contributions:**

In this paper, a novel SSC scheme is presented that utilizes generalized OMP (GOMP), an enhanced version of OMP that identifies multiple neighbors in each iteration. The proposed method also introduces a new stopping rule that only requires knowledge of the signal dimension and the number of neighbors identified per iteration. Compared to conventional OMP, GOMP requires fewer iterations and has lower algorithmic complexity. Furthermore, the proposed stopping rule does not require an offline estimation of the subspace dimension or noise strength. Then, analytic performance guarantees are provided under the semi-random model. Finally, numerical experiments are conducted to verify the clustering performance of the proposed method.

**Audience:**

Yes

**Broader Impact Concerns:**

None.

**Claims And Evidence:**

No

**Requested Changes:**

Please refer to the weaknesses enumerated above.

**Strengths And Weaknesses:**

Strengths:
1. Compared with OMP, the proposed GOMP costs less time for fast neighbor identification.
2. The proposed GOMP employs a new stopping rule geared toward fulfilling the dimension-aware property for greedy-based neighbor selection.
3. The paper also provides analytic performance guarantees for the proposed method.

Weaknesses:
1. This paper is relatively difficult to understand, especially for those who are not very familiar with SSC. In the abstract, the authors do not introduce what shortcomings exist in existing work and what the motivation is for doing the current work. After introducing the motivation and related work, the author did not list specific objective functions and explain the problems that exist in existing work. Instead, they directly begin to introduce their work, which makes it difficult for readers to understand the advantages of the proposed method. The author should address the above issues to make the paper easier to read.
2. Many mathematical symbols lack explanation, especially in Algorithm 1, such as $\Lambda_m$, $\mathbf{Y}$, $\mathbf{Y}_{\Lambda_m}$. The author should provide corresponding explanations when symbols first appear.
3. It is very good to provide theoretical guarantees. However, in the proof process in the paper, the authors heavily relied on the proofs of others, which greatly increase the difficulty of reading the paper. Usually, it is best to only cite other people's theorems or lemmas, and to write the entire theorem in the paper for readers' convenience, rather than citing the proof process.

Overall, I think that this paper needs significant adjustments in order to reach the level required for publication.

---

> ### Author Response · Authors · 2023-06-27
> **Authors’ responses to Reviewer 9QED’s comments**
>
> We would like to thank the reviewer 9QED for his/her insightful review. We have revised the manuscript accordingly, wherein the main changes are highlighted in blue for ease of reviewer’s identification. Below are our point-to-point responses to all the raised concerns.
>
> $\newline$
>
> $\textcolor{Salmon}{\bf{Weakness \text{ }1}}$: This paper is relatively difficult to understand, especially for those who are not very familiar with SSC. In the abstract, the authors do not introduce what shortcomings exist in existing work and what the motivation is for doing the current work. After introducing the motivation and related work, the author did not list specific objective functions and explain the problems that exist in existing work. Instead, they directly begin to introduce their work, which makes it difficult for readers to understand the advantages of the proposed method. The author should address the above issues to make the paper easier to read.
>
> $\textcolor{NavyBlue}{\bf{[Reply]}}$: We thank the reviewer’s suggestions for improving the presentation and technical writing of this paper. In the abstract, several sentences are added to introduce the shortcomings existing in previous works and explain our motivation (see the 3rd to 5th lines and the 7th line of the revised abstract on page 1). In Section 1.1, which is the beginning of the paper, a new paragraph is added to give an overview of SSC so that readers can quickly know the specific objective function (optimizing problem) that considered in SSC, as well as the shortcomings that exist in existing works (see page 1 and 2 of the revised manuscript). Hopefully the revisions done can better clarify the contributions/advantages of the proposed method, and improve readability of this paper.
>
> $\newline$
>
> $\textcolor{Salmon}{\bf{Weakness \text{ }2}}$: Many mathematical symbols lack explanation, especially in Algorithm 1, such as  $\Lambda_\{m\}$,  $\mathbf{Y}$,  $\mathbf{Y}\_\{\Lambda_\{m\}\}$. The author should provide corresponding explanations when symbols first appear
>
> $\textcolor{NavyBlue}{\bf{[Reply]}}$: We thank the reviewer for this comment. We have accordingly checked over the whole manuscript to make sure every mathematical symbol is explained when the symbol first appears. In particular, $\mathbf{Y}$  is introduced in the 11th line below equation (1) on page 1 of the revised manuscript and  $\Lambda_\{m\}$  ( $\mathbf{Y}\_\{\Lambda_\{m\}\}$, respectively) is introduced in the 12th (13th, respectively) line below equation (2) on page 2 of the revised manuscript.
>
> $\newline$
>
> $\textcolor{Salmon}{\bf{Weakness \text{ }3}}$: It is very good to provide theoretical guarantees. However, in the proof process in the paper, the authors heavily relied on the proofs of others, which greatly increase the difficulty of reading the paper. Usually, it is best to only cite other people's theorems or lemmas, and to write the entire theorem in the paper for readers' convenience, rather than citing the proof process.
>
> $\textcolor{NavyBlue}{\bf{[Reply]}}$:We thank the reviewer for this comment. Following the reviewer’s suggestion, we cite the entire theorems and lemmas in the referenced papers in place of the proof process. Specific revision made to take care of this comment include:
> 1. On page 24 of the revised manuscript, we cite Lemma 10 of (Tschannen & Bolcskei, 2018), appearing as Lemma 4 at the beginning of Section 5, in place of the proof process given in the 1st line below equation (52) on page 25 and the 1st line below equation (96) on page 34 in the old manuscript.
> 2. On page 37 of the revised manuscript, we cite Lemma 7.5 of (Soltanolkotabi & Candes, 2012), appearing as Lemma 6, to replace the proof process given in the 2nd line below equation (88) on page 33 in the old manuscript.
>
> We hope the readability of the revised manuscript has been accordingly improved.

---

### Review · Reviewer_ZEAg · 2023-05-06

**Summary Of Contributions:**

The paper proposes a generalized orthogonal matching pursuit (GOMP) algorithm where multiple neighbors are identified at each iteration together with a new stopping criterion. Compared to the classical OMP, GOMP requires fewer iterations and then reduces computational complexity. However, it is not clear whether GOMP takes more running time for each iteration than OMP and thus the claimed comparison fairness needs to be addressed. Recovery rate under certain probability is also theoretically analyzed for the proposed GOMP. Numerical results on synthetic and realistic data sets are provided.

**Audience:**

Yes

**Broader Impact Concerns:**

None.

**Claims And Evidence:**

No

**Requested Changes:**

Substantial revision on language and careful check of notional consistency are desired.
1. In (1), replace the comma by the set union sign.
2. In p.2, it is not clear about, 'errorless neighbor identification alone ... pessimistic condition."
3. In Algorithm 1, it should be "m<- m+1" in 1) and "m-1" -> "m" in 4). It is not clear how the output is related to step 8).
4. In p.11, "MNST" -> "MNIST"?
5. In p.12, "gen-erated" -> "generated", "beginequation"?
...

**Strengths And Weaknesses:**

strengths: The proposed algorithm seems novel.
weakness: The paper is hard to follow with some typos, English and notational issues.

---

> ### Author Response · Authors · 2023-06-27
> **Authors’ responses to Reviewer ZEAg’s comments**
>
> We would like to thank the reviewer ZEAg for his/her insightful review. We have revised the manuscript accordingly, wherein the main changes are marked in blue for ease of reviewer’s identification. Below are our point-to-point responses to all the raised concerns.
>
> $\newline$
>
> $\textcolor{Salmon}{\bf{Weakness \text{ }1}}$: The paper proposes a generalized orthogonal matching pursuit (GOMP) algorithm where multiple neighbors are identified at each iteration together with a new stopping criterion. Compared to the classical OMP, GOMP requires fewer iterations and then reduces computational complexity. However, it is not clear whether GOMP takes more running time for each iteration than OMP and thus the claimed comparison fairness needs to be addressed. Recovery rate under certain probability is also theoretically analyzed for the proposed GOMP. Numerical results on synthetic and realistic data sets are provided.
>
> $\textcolor{NavyBlue}{\bf{[Reply]}}$: We thank the reviewer for this comment. If a total number of $pM$ neighbors are to be recovered and GOMP identifies $p>1$ neighbors per iteration, then it just calls for $M$ iterations, while OMP requires $pM$  ones. For  $1\leq m\leq M$, the running time of the $m$th iteration of GOMP is slightly higher than that of the $pm$th iteration of OMP. The reason behind is that, while OMP computes the maximal absolute inner product, GOMP seeks the first $p$ largest ones: this entails additional sorting efforts and therefore larger running time. Despite this, GOMP still benefits from fewer iterations ($(p-1)M$  less than OMP) and results in shorter running time. To further clarify this point, Section 2.3 regarding algorithmic complexity is expanded to discuss and compare the running time of GOMP and OMP (see page 10 of the revised manuscript). Using the data set for generating Fig. 1, the newly added Table I lists the consumed running time of the two methods. The table shows that throughout all iterations the running time of GOMP is about 37.5% of that of OMP.
>
> $\newline$
>
> $\textcolor{Salmon}{\bf{Weakness \text{ }2}}$: Substantial revision on language and careful check of notional consistency are desired.
> 1. In (1), replace the comma by the set union sign.
> 2. In p.2, it is not clear about, 'errorless neighbor identification alone ... pessimistic condition."
> 3. In Algorithm 1, it should be "m<- m+1" in 1) and "m-1" -> "m" in 4). It is not clear how the output is related to step 8).
> 4. In p.11, "MNST" -> "MNIST"?
> 5. In p.12, "gen-erated" -> "generated", "beginequation"? ...
>
> $\textcolor{NavyBlue}{\bf{[Reply]}}$: We thank the reviewer for this comment. We have carefully gone through the manuscript again to revise language usage, in particular, to fix the typos pointed out by the reviewer (in questions 1, 4, and 5) and check notational consistency. In addition, the following changes are made to clarify the statements about SDP and Algorithm 1.
>
> 1. To take care of question 2, we first revise the definition of SDP (see the last two lines on page 2 of the revised manuscript); then, the sentence pointed out by the reviewer is rewritten to avoid confusion (see the 3rd line on page 3 of the revised manuscript).
>
> 2. As for question 3 concerning statements in Algorithm 1 (now appears as Algorithm 4 on page 9 of the revised manuscript), the output of step 8) is corrected to “  $\bf{Output}$: Partition  $\mathcal{Y}=\widehat{\mathcal{Y}}\_1\cup\cdots\cup\widehat{\mathcal{Y}}\_\widehat{L}$  ”. Meanwhile, the notation in step 1) is modified to $m\gets m+1$  as suggested by the reviewer. Regarding the statement in step 4), we first note that the symbol $m$  stands for the running index of iterations, but not the total number of conducted iterations. To clarify this point, in step 4) we therefore assume the algorithm is terminated with $M^{(i)}$  iterations, and replace the symbol "$\Lambda\_\{m-1\}$" by “ $\Lambda\_\{M^{(i)}-1\}$ ”. Here we reserve “$\Lambda\_\{M^{(i)}-1\}$ ” instead of “$\Lambda\_\{M^{(i)}\}$ ”, mainly because the data points recovered in the  $M^{(i)}$th iteration, namely,  $\mathbf{y}\_j$,  $j\in\mathcal{T}\_\{M^{(i)}\}$, are unlikely to be neighbors of  $\mathbf{y}\_i$ and thus do not participate in the computation of the least squares solution in step 4). At the end of Section 2.2, we add a new paragraph to elaborate more upon this point (see page 9 of the revised manuscript).

---

### Review · Reviewer_dwHk · 2023-07-07

**Summary Of Contributions:**

This paper explores the use of generalized orthogonal matching pursuit (GOMP) as a replacement for OMP in computing the sparse coefficients in sparse subspace clustering (SSC) -- the main insight being that the strength of the noise subspaces increases as more subspace components are recovered by OMP incrementally such that selecting many components in each iteration may lead to better recovery. The paper also proposes a new stopping criteria for GOMP iterations that depend only on the ambient signal dimension and the number of neighbors selected at every GOMP iteration. The recovery rate analysis is provided under the semi-random noise model. Experiments are provided on synthetic and real datasets and demonstrate promising results.

**Audience:**

Yes

**Broader Impact Concerns:**

The broader impact is missing, and as I see it, it is unclear. I do not see any ethical issues with this paper.

**Claims And Evidence:**

Yes

**Requested Changes:**

Please see above.

Overall, I think the idea is interesting, however, the claimed contributions appear less rigorous and unconvincing. The theoretical results are too complex to have any takeaway insights. The experiments show promise. Overall, I am slightly aligning towards acceptance, although the paper needs to be revised to account for the problems I listed.

**Strengths And Weaknesses:**

Strengths:
The paper provides an interesting direction of using GOMP for SSC. I think this thought is interesting and novel, and the provided recovery rate analysis is perhaps useful for future research (although I am not an expert in analysis to comment on the usefulness of those results). Further,
1. The paper is relatively well written and easy to read.
2. There is sufficient background provided that makes the context of the contributions very clear.
. The experiments are quite elaborate and show promise against OMP; however the performances do not appear better than other approaches to clustering (although I am not too worried about that).

Weaknesses:
1. The motivation for using GOMP, while intuitive, is less convincing. The key idea to use GOMP is that the angle of deviation (AoD) between the residual strength in the ground truth subspace and its orthogonal complement increases as more subspaces are found incrementally using OMP. However, when using GOMP, won't this problem persist if one does not know the dimensionality d_k of the  subspace? And in case, if one knows the subspace dimensionality d_k, then why not use p=d_k?

2. Further, when working in a sparse subspace clustering setting, there may be several data points y_i that belong to the subspace S_k. Let's say two points y_i and y_j \in S_k are very similar that selecting them both in a GOMP iteration does not add any benefit to AoD as such. Won't you need a degree of incoherence between the p selected neighbors in GOMP to make them useful for the subspace selection  This latter point also affects the claimed algorithmic complexity benefits as, in such settings, one will expend more compute for finding the p nearest neighbors while in the worst case ending up with the same number of iterations as in OMP for finding the subspaces in the SSC setting. In this sense, I think there appears to be a disconnect in the paper from considering GOMP as a means of replacing OMP in signal recovery against using GOMP for subspace selection in SSC; in the latter, the matrix Y of data points may contain many redundant data points and using p selected neighbors destroys the intended "sparsity" desired in SSC? I would think it may be important to include additional assumptions on Y to account for such issues.

3. A second contribution of this paper is the proposal of a halting rule based on the subspace dimensionality as against the know-how of the number of iterations M or the strength of the noise \tau used in prior methods. Considering Eq. (6), it is not clear to me why this stopping rule is significant or how can it work in a practical setting? More specifically, I presume p << d_k, and is assumed n-d_k ~ n, then p/n is also close to zero? For example, in the real experiments reported in the paper, p is about 6 and n is in thousands. If so, ||r_{m+1}|| / ||r_m|| is very close to 1? if n is large, this stopping criteria (||r_{m+1}||/||r_m|| \leq 1 - \sqrt(p/n)) is very close to 1, and in that case, how can one even use this criteria? With this sensitivity on p/n, how is this criteria better than using a thresholded stopping rule as in \tau used in previous works? A second issue with (6) is that it is derived by assuming that in (4) the signal component \Pi P_l x_i is close to zero after a few iterations. In reality, we do not necessarily know when this component is close to zero and it is unclear to me if this stopping criteria makes sense when the signal component is non-zero.

5. The recovery rate analysis appears quite complicated, and there are very strong assumptions that need to be made to make it comprehensible, such as the noise variance \sigma is small (14). The experiments also do not adhere to the insights in the recovery rates. While, the paper provides various situations when the recovery rates could be understood, it does not appear to be quite aligned with the experimental results.

---

> ### Author Response · Authors · 2023-08-03
> **Authors’ responses to Reviewer dwHk’s comments (1/6)**
>
> We would like to thank the reviewer dwHk for his/her insightful review. We have revised the manuscript accordingly, wherein the main changes in the context are marked in $\textcolor{purple}{\text{purple}}$ for ease of reviewer’s identification. Below are our point-to-point responses to all the raised concerns.
>
> $\newline$
>
> $\textcolor{Salmon}{\bf{Weakness \text{ }1}}$: The motivation for using GOMP, while intuitive, is less convincing. The key idea to use GOMP is that the angle of deviation (AoD) between the residual strength in the ground truth subspace and its orthogonal complement increases as more subspaces are found incrementally using OMP. However, when using GOMP, won't this problem persist if one does not know the dimensionality $d\_k$ of the subspace? And in case, if one knows the subspace dimensionality $d\_k$, then why not use $p=d\_k$?
>
> $\textcolor{NavyBlue}{\bf{[Reply]}}$: We thank the reviewer for this comment. We would like to clarify that AoD of the residual vector increases as the algorithm iterates, $\textit{irrespective of OMP or GOMP}$. Yet, multi-neighbor recovery achieved by GOMP can reduce the number of iterations so as to prevent large deviation of AoD caused by noise; this is believed to be a generic phenomenon $\textit{no matter whether the subspace dimension } d\_k  \textit{ is known or not}$, as confirmed by our experimental results in Fig. 1 wherein the implementation of GOMP does not assume knowledge of $d\_k$. In case that $d\_k$  is exactly known, we can choose to set $p=d\_k$ so that the proposed GOMP executes just one iteration; the resultant performance is expected to be reasonably good provided that each data point has at least $d\_k$ correct neighbors in close proximity. At the end of Section 4.1, new computer simulations are conducted and a new paragraph is further added to investigate the performance of GOMP using $p=d\_k$ (please see page 18 of the revised manuscript).

---

> > ### Author Response · Authors · 2023-08-03
> > **Authors’ responses to Reviewer dwHk’s comments (2/6)**
> >
> > $\textcolor{Salmon}{\bf{Weakness \text{ }2}}$: Further, when working in a sparse subspace clustering setting, there may be several data points $\mathbf{y}\_i$ that belong to the subspace $\mathcal{S}\_k$. Let's say two points $\mathbf{y}\_i$ and $\mathbf{y}\_j\in\mathcal{S}\_k$ are very similar that selecting them both in a GOMP iteration does not add any benefit to AoD as such. Won't you need a degree of incoherence between the $p$ selected neighbors in GOMP to make them useful for the subspace selection? This latter point also affects the claimed algorithmic complexity benefits as, in such settings, one will expend more compute for finding the $p$ nearest neighbors while in the worst case ending up with the same number of iterations as in OMP for finding the subspaces in the SSC setting. In this sense, I think there appears to be a disconnect in the paper from considering GOMP as a means of replacing OMP in signal recovery against using GOMP for subspace selection in SSC; in the latter, the matrix $\mathbf{Y}$ of data points may contain many redundant data points and using $p$ selected neighbors destroys the intended "sparsity" desired in SSC? I would think it may be important to include additional assumptions on $\mathbf{Y}$ to account for such issues.
> >
> > $\textcolor{NavyBlue}{\bf{[Reply]}}$: We thank the reviewer for this comment. We would like to clarify that data coherence as mentioned by the reviewer can otherwise add benefit to AoD. To see this, assume that two data points $\mathbf{y}\_i$ and $\mathbf{y}\_j\in\mathcal{S}\_k$ are coherent (aligned toward the same direction) and both are selected by GOMP during the same iteration. As such, the subspace identified throughout this iteration is of dimension $p-1$ rather than $p$. To obtain the updated residual vector $\mathbf{r}\_m\^{(i)}$, the current residual $\mathbf{r}\_\{m-1\}\^{(i)}$ is projected onto the orthogonal complement of the span of “already-selected neighbors”, which is of a higher dimension (one more) than the inherent case. As such, the resultant signal component  $\mathbf{r}\^{(i)}\_\{{m,\|\|\}}$ is better retained, leading to a potentially larger $\|\|\mathbf{r}\^{(i)}\_\{{m,\|\|\}}\|\|\_2$ and consequently a smaller AoD$(=\tan\^{-1}(\|\|\mathbf{r}\^{(i)}\_\{{m,\bot\}}\|\|\_2/\|\|\mathbf{r}\^{(i)}\_\{{m,\|\|\}}\|\|\_2))$, since the strength $\|\|\mathbf{r}\^{(i)}\_\{{m,\bot\}}\|\|\_2$ of the projected misidentified neighbors (if any) plus noise onto $\mathcal{S}\_k\^{\bot}$ is roughly the same in both cases. At the end of Section 2.1, a new remark along with a new experimental study (designed to specifically take care of the situation mentioned by the reviewer) are added to elaborate more on this point; please see page 7 of the revised manuscript.
> >
> >   In view of the above discussions, additional criteria or mechanisms for ensuring incoherence among the $p$ selected neighbors does not seem necessary. We would like to mention that, for practical noisy datasets (our study begins with the assumption that the dataset $\mathcal{Y}$ in (1) is noisy), almost coherent data points seldom occur, and the above-mentioned situation is very rare. As far as we know, mathematical conditions guaranteeing data incoherence, such as the restricted-isometry-property (RIP) of the data matrix $\mathbf{Y}$, has been adopted in the study of performance guarantees of sparse subspace clustering (see Soltanolkotabi et al., 2014; Tschannen & Bölcskei, 2018).

---

> > > ### Author Response · Authors · 2023-08-03
> > > **Authors’ responses to Reviewer dwHk’s comments (3/6)**
> > >
> > > $\textcolor{Salmon}{\bf{Weakness \text{ }3}}$: A second contribution of this paper is the proposal of a halting rule based on the subspace dimensionality as against the know-how of the number of iterations $M$ or the strength of the noise $\tau$ used in prior methods. Considering Eq. (6), it is not clear to me why this stopping rule is significant or how can it work in a practical setting? More specifically, I presume $p\ll d\_k$, and is assumed $n-d\_k\approx n$, then $p/n$ is also close to zero? For example, in the real experiments reported in the paper, $p$ is about 6 and $n$ is in thousands. If so, $\|\|\mathbf{r}^{(i)}\_\{{m+1\}}\|\|\_2/\|\|\mathbf{r}^{(i)}\_\{{m\}}\|\|\_2$ is very close to 1? If $n$ is large, this stopping criteria $\|\|\mathbf{r}^{(i)}\_\{{m+1\}}\|\|\_2/\|\|\mathbf{r}^{(i)}\_\{{m\}}\|\|\_2\leq 1-\sqrt{p/n}$ is very close to 1, and in that case, how can one even use this criteria? With this sensitivity on $p/n$, how is this criteria better than using a thresholded stopping rule as in $\tau$ used in previous works? A second issue with (6) is that it is derived by assuming that in (4) the signal component  $\prod^m\_\{l=1\}\mathbf{P}\_l\mathbf{x}\_i$ is close to zero after a few iterations. In reality, we do not necessarily know when this component is close to zero and it is unclear to me if this stopping criteria makes sense when the signal component is non-zero.
> > >
> > > $\textcolor{NavyBlue}{\bf{[Reply]}}$: We thank the reviewer for this comment. Firstly, we would like to clarify the significance of the proposed stopping rule (6). The development of (6) is based on the premise that the residual $\mathbf{r}\_m^{(i)}$ is noise-only (the case with non-zero projected signal is addressed in our reply to the 2nd raised issue) so that $\|\|\mathbf{r}^{(i)}\_\{{m+1\}}\|\|\_2\approx \|\|\mathbf{r}^{(i)}\_\{{m\}}\|\|\_2$, and therefore the ratio $\|\|\mathbf{r}^{(i)}\_\{{m+1\}}\|\|\_2/\|\|\mathbf{r}^{(i)}\_\{{m\}}\|\|\_2$ is close to one. The threshold $1-\sqrt{p/n}$ on the right-hand-side of (6) is obtained by exploiting the Gaussian noise assumption, as argued in Section 2.2. We would like to note that the left-hand-side of (6), namely, the residual norm ratio $\|\|\mathbf{r}^{(i)}\_\{{m+1\}}\|\|\_2/\|\|\mathbf{r}^{(i)}\_\{{m\}}\|\|\_2$, is a random variable, so that there’s no way of ensuring the inequality always holds. Instead, under Gaussian noise assumption it has been shown in the proof of Theorem 2 (see Section 5.3) that, when $\lceil d\_k/p \rceil$, inequality (6) holds with a probability higher than $1-2pe^{-\sqrt{n/p}}$ (for ease of reviewer’s reference, we highlight the proof in Section 5.3 in the Appendix placed at the end of our response). As a result, as $p/n$ is close to zero (thus, $n/p$ is very large), the proposed stopping rule (6) can highly likely be triggered. In view of the above fact, our proposed stopping rule (6) can work in a practical setting irrespective of the sensitivity pointed out by the reviewer. We also note that the above probabilistic interpretation of stopping criteria is not uncommon in the SSC literature. Indeed, the widely considered thresholding-based criterion $\|\|\mathbf{r}^{(i)}\_\{{m\}}\|\|\_2\leq\tau$ (see Dyer et al., 2013 ;Tschannen & Bölcskei, 2018) cannot be guaranteed to meet deterministically, as $\|\|\mathbf{r}^{(i)}\_\{{m\}}\|\|\_2$ is a random variable. The second issue will be answered in the next cell.

---

> > > > ### Author Response · Authors · 2023-08-03
> > > > **Authors’ responses to Reviewer dwHk’s comments (4/6)**
> > > >
> > > >   To answer the 2nd issue, we agree with the reviewer that we cannot directly examine if the projected signal component $\prod^m\_\{l=1\}\mathbf{P}\_l\mathbf{x}\_i$ is zero. Yet, we would like to clarify that, as long as sufficiently many true neighbors are recovered, the term $\prod^m\_\{l=1\}\mathbf{P}\_l\mathbf{x}\_i$ actually acts as a noise. Hence the residual is indeed “noise-only” irrespective of whether $\prod^m\_\{l=1\}\mathbf{P}\_l\mathbf{x}\_i$ is zero or not, and our arguments of deriving (6) remains valid. To see this, let us stack the “already-selected” neighbors up to the $m$th iteration as a matrix $\widetilde{\mathbf{Y}}^{(i)}\_m\in\mathbb{R}^{n\times pm}$, which admits the form $\widetilde{\mathbf{Y}}^{(i)}\_m=\widetilde{\mathbf{X}}^{(i)}\_m+\widetilde{\mathbf{E}}^{(i)}\_m$, where $\widetilde{\mathbf{X}}^{(i)}\_m$ and $\widetilde{\mathbf{E}}^{(i)}\_m$ are the signal point and noise matrices, respectively. In case there are sufficiently many correct neighbors recovered, we have $\mathbf{x}\_i=\widetilde{\mathbf{X}}^{(i)}\_m\mathbf{c}$ for some $\mathbf{c}$. In this way, the projected signal component reads
> > > >
> > > >
> > > > $\prod^m\_\{l=1\}\mathbf{P}\_l\mathbf{x}\_i=\prod^{m-1}\_\{l=1\}\mathbf{P}\_l\underbrace{\Big(\mathbf{I}-\widetilde{\mathbf{Y}}^{(i)}\_m(\widetilde{\mathbf{Y}}^{(i)}\_m\text{}^T\widetilde{\mathbf{Y}}^{(i)}\_m)^{-1}\Big)\widetilde{\mathbf{Y}}^{(i)}\_m\text{}^T}\_\{\mathbf{P}\_m\}\mathbf{x}\_i$
> > > >
> > > > $=\prod^{m-1}\_\{l=1\}\mathbf{P}\_l\Big(\mathbf{I}-\widetilde{\mathbf{Y}}^{(i)}\_m(\widetilde{\mathbf{Y}}^{(i)}\_m\text{}^T\widetilde{\mathbf{Y}}^{(i)}\_m)^{-1}\Big)\widetilde{\mathbf{Y}}^{(i)}\_m\text{}^T(\mathbf{x}\_i-\widetilde{\mathbf{Y}}^{(i)}\_m\mathbf{c})$
> > > >
> > > > $=\prod^{m-1}\_\{l=1\}\mathbf{P}\_l\Big(\mathbf{I}-\widetilde{\mathbf{Y}}^{(i)}\_m(\widetilde{\mathbf{Y}}^{(i)}\_m\text{}^T\widetilde{\mathbf{Y}}^{(i)}\_m)^{-1}\Big)\widetilde{\mathbf{Y}}^{(i)}\_m\text{}^T(\mathbf{x}\_i-\widetilde{\mathbf{X}}^{(i)}\_m\mathbf{c}-\widetilde{\mathbf{E}}^{(i)}\_m\mathbf{c})$
> > > >
> > > > $=\prod^{m-1}\_\{l=1\}\mathbf{P}\_l\Big(\mathbf{I}-\widetilde{\mathbf{Y}}^{(i)}\_m(\widetilde{\mathbf{Y}}^{(i)}\_m\text{}^T\widetilde{\mathbf{Y}}^{(i)}\_m)^{-1}\Big)\widetilde{\mathbf{Y}}^{(i)}\_m\text{}^T(-\widetilde{\mathbf{E}}^{(i)}\_m\mathbf{c})=\prod^m\_\{l=1\}\mathbf{P}\_l(-\widetilde{\mathbf{E}}^{(i)}\_m\mathbf{c}).$
> > > >
> > > > The residual vector accordingly becomes $\mathbf{r}\_m^{(i)}=\prod^m\_\{l=1\}\mathbf{P}\_l(-\widetilde{\mathbf{E}}^{(i)}\_m\mathbf{c})+\prod^m\_\{l=1\}\mathbf{P}\_l\mathbf{e}\_i=\prod^m\_\{l=1\}\mathbf{P}\_l(-\widetilde{\mathbf{E}}^{(i)}\_m\mathbf{c}+\mathbf{e}\_i)$, which is again a projected noise. Hence, the proposed stopping rule (6) makes sense and still works even when $\prod^m\_\{l=1\}\mathbf{P}\_l\mathbf{x}\_i$ is non-zero.
> > > >
> > > >   Considering all the above facts, the revisions done to take care of this comment include:
> > > >
> > > > 1. The paragraph below (6) is expanded to comment on the probabilistic interpretation of inequality (6). Meanwhile, a footnote is newly added to mention similar interpretation has been seen in the previous studies that employs the residual norm thresholding $\|\|\mathbf{r}^{(i)}\_\{{m\}}\|\|\_2\leq\tau$ (see page 8 of the revised manuscript).
> > > > 2. A remark is newly added to the end of Section 2.2 to explain why the proposed stopping rule (6) remains valid in case that the projected signal component $\prod^m\_\{l=1\}\mathbf{P}\_l\mathbf{x}\_i$ is nonzero (see page 9 of the revised manuscript).

---

> > > > > ### Author Response · Authors · 2023-08-03
> > > > > **Authors’ responses to Reviewer dwHk’s comments (5/6)**
> > > > >
> > > > > $\textcolor{Salmon}{\bf{Weakness \text{ }4}}$: The recovery rate analysis appears quite complicated, and there are very strong assumptions that need to be made to make it comprehensible, such as the noise variance $\sigma$  is small (14). The experiments also do not adhere to the insights in the recovery rates. While, the paper provides various situations when the recovery rates could be understood, it does not appear to be quite aligned with the experimental results.
> > > > >
> > > > > $\textcolor{NavyBlue}{\bf{[Reply]}}$: We thank the reviewer for this comment. First of all, we would like to mention that the development of analytic performance guarantees for machine learning systems, and in particular for sparse subspace clustering, is in general a difficult task. While the derived recovery rate formulae based on the semi-random model are quite complicated, oftentimes it is necessary to stick to certain assumptions, such as low subspace affinity and small noise, to make the analytical results comprehensible (like small noise variance in (14) as pointed out by the reviewer). We emphasize again that our theoretical study is built on the standard semi-random model (uniform source data and Gaussian noise), which is widely considered in the SSC literature (Soltanolkotabi & Candès, 2012; Tschannen & Bölcskei, 2018). It is not unexpected that the obtained analytic results do not appear to be always aligned with the experimental results using real datasets, which typically do not obey the semi-random model assumption. As far as we know, our proposed approach based on the semi-random model (and the likes) could be by far the only solution to analytic guarantees for SSC. Extension of current analyses to more general model assumptions remains an open problem (indeed a big challenge) in the study of SSC, and is one of our chief future works.

---

> > > > > > ### Author Response · Authors · 2023-08-03
> > > > > > **Authors’ responses to Reviewer dwHk’s comments (6/6)**
> > > > > >
> > > > > > $\bf{\large \text{Appendix: Highlight of Proof in Section 5.3}}$
> > > > > >
> > > > > >
> > > > > >   Our exposure in the main context of recovery rate analysis (Section 3) considers neighbor identification for the last data point, namely, $\mathbf{y}\_N$, whose ground truth subspace is $\mathcal{S}\_L$ (see the paragraph before Theorem 1, on page 11 of the revised manuscript). Throughout the proof in Section 5.3 the subscript of the residual vector in the $m$th iteration is $N$ rather than $i$. Hence, we set $i=N$ in inequality (6) since $\mathbf{y}\_N$ is concerned.
> > > > > >
> > > > > >   Using the Pythagorean theorem, an equivalent form of (6) is first given in (81) on page 32 of the revised manuscript, and our purpose is to show that, with $m=\lceil d\_L/p \rceil$, inequality (81) (or inequality (83)) holds with an overwhelming probability. By definition (80), $\widetilde{\mathbf{r}}^{(N)}\_\{\lceil d\_L/p \rceil+1\}$ represents the difference between the residuals $\mathbf{r}^{(N)}\_\{\lceil d\_L/p \rceil\}$ and $\mathbf{r}^{(N)}\_\{\lceil d\_L/p \rceil+1\}$, normalized with respect to $\|\|\mathbf{r}^{(N)}\_\{\lceil d\_L/p \rceil\}\|\|\_2$. Recall from step 3) of the GOMP in Table IV that
> > > > > >
> > > > > > $\newline$
> > > > > >
> > > > > >  $\mathbf{r}^{(N)}\_\{\lceil d\_L/p \rceil+1\}=\Big(\mathbf{I}-\mathbf{Y}\_\{\Lambda\_\{\lceil d\_L/p \rceil+1 \}\}(\mathbf{Y}\_\{\Lambda\_\{\lceil d\_L/p \rceil+1 \}\}^T\mathbf{Y}\_\{\Lambda\_\{\lceil d\_L/p \rceil+1 \}\})^{-1}\mathbf{Y}\_\{\Lambda\_\{\lceil d\_L/p \rceil+1 \}\}^T\Big)\mathbf{r}^{(N)}\_\{\lceil d\_L/p \rceil\},$
> > > > > >
> > > > > > $\newline$
> > > > > >
> > > > > > where $\mathbf{Y}\_\{\Lambda\_\{\lceil d\_L/p \rceil+1 \}\}$ is the matrix consists of all the selected neighbors up to the $(\lceil d\_L/p \rceil+1)$th iteration, implying $\mathbf{r}^{(N)}\_\{\lceil d\_L/p \rceil\}-\mathbf{r}^{(N)}\_\{\lceil d\_L/p \rceil+1\}=\Big(\mathbf{Y}\_\{\Lambda\_\{\lceil d\_L/p \rceil+1 \}\}(\mathbf{Y}\_\{\Lambda\_\{\lceil d\_L/p \rceil+1 \}\}^T\mathbf{Y}\_\{\Lambda\_\{\lceil d\_L/p \rceil+1 \}\})^{-1}\mathbf{Y}\_\{\Lambda\_\{\lceil d\_L/p \rceil+1 \}\}^T\Big)\mathbf{r}^{(N)}\_\{\lceil d\_L/p \rceil\}$ and, therefore,
> > > > > >
> > > > > > $\newline$
> > > > > >
> > > > > >  $\widetilde{\mathbf{r}}^{(N)}\_\{\lceil d\_L/p \rceil+1\}=\Big(\mathbf{Y}\_\{\Lambda\_\{\lceil d\_L/p \rceil+1 \}\}(\mathbf{Y}\_\{\Lambda\_\{\lceil d\_L/p \rceil+1 \}\}^T\mathbf{Y}\_\{\Lambda\_\{\lceil d\_L/p \rceil+1 \}\})^{-1}\mathbf{Y}\_\{\Lambda\_\{\lceil d\_L/p \rceil+1 \}\}^T\Big)\mathbf{r}^{(N)}\_\{\lceil d\_L/p \rceil\}/\|\|\mathbf{r}^{(N)}\_\{\lceil d\_L/p \rceil\}\|\|\_2,$
> > > > > >
> > > > > > $\newline$
> > > > > >
> > > > > > which is exactly equation (87). Using the fact that $\mathbf{r}^{(N)}\_\{\lceil d\_L/p \rceil\}$ is orthogonal to all the selected neighbors up to the $\lceil d\_L/p \rceil$th iteration, thus $\widetilde{\mathbf{r}}^{(N)}\_\{\lceil d\_L/p \rceil+1\}$ lying in a subspace (spanned by just those recovered during the $(\lceil d\_L/p \rceil+1)$th iteration) of dimension $p$, along with the union bound, (88) derived a probability lower bound for the event that the strength of $\mathbf{r}^{(N)}\_\{\lceil d\_L/p \rceil\}/\|\|\mathbf{r}^{(N)}\_\{\lceil d\_L/p \rceil\}\|\|\_2$ per dimension is less than $\sqrt{2\sqrt{p/n}-p/n}/\sqrt{p}$. With the aid of the Gaussian noise assumption and conditioned on that all the revered neighbors are correct, we go through (89)~(91) to show that the mentioned event occurs with a probability higher than $1-2e^{-\sqrt{n/p}}$.

---

### Decision · Action_Editors · 2023-10-03

**Recommendation:** Accept with minor revision

**Comment:**

Based on the strength of reviewer dwHk's recommendation, I am pleased to accept the paper. The reviewer raised several issues, but the authors had good responses that satisfied both the reviewer and myself.  The paper has some theory and also shows some practical benefits, and also has good discussion of motivation for GOMP over OMP.  The paper and figures are professional.

The other two reviewers, while less extensive in their comments, did not raise any issues or objections.

Before accepting the final version of the paper, I request that the authors do a final copy-edit themselves. For example, the title of Section 3 is "Theorectical results" (rather than "Theoretical"), so please at least run a spell-check.

**Audience:**

Yes, some of TMLR audience would be interested. Sparse subspace clustering is still not yet a "workhorse" algorithm in machine learning (for example, it is not built in to scikit-learn), but it's received quite a bit of attention the past 10 years, and is of sufficient interest.

**Claims And Evidence:**

The claims are supported by theoretical proofs and by numerical experiments. The reviewers did not raise any concerns here.

---

> ### Author Response · Authors · 2023-10-06
> **Official Comment by Authors**
>
> Dear Prof. Becker:
>
> Thank you very much for the kind decision. We will have the paper finalized through a careful copy-edit.
>
> Best regards,
>
> Jwo-Yuh Wu